# Contextual Slate GLM Bandits with Limited Adaptivity

**Tanmay Goyal** [1]   **Sukruta Prakash Midigeshi** [1]   **Gaurav Sinha** [1]

## Abstract

We investigate the contextual slate bandit problem with generalized linear rewards under limited adaptivity. At each round, the learner is presented with $N$ sets of items, where each item is represented by a $d$-dimensional feature vector. The learner then constructs a slate by selecting one item per set; the resulting slate yields a scalar reward sampled from a Generalized Linear Model (GLM). We propose algorithms under two limited-adaptivity settings: (a) Batched and (b) Rarely-Switching. For the batched setting, we introduce `B-SlateGLinCB`, which partitions the time horizon into $\mathcal{O}(\log\log T)$ batches such that each batch's policy relies only on data from previous batches. For the rarely-switching setting, we propose `RS-SlateGLinCB`, which adaptively performs only $\mathcal{O}(Nd\log T)$ parameter updates. Under a diversity assumption on the item sequences, we prove that `B-SlateGLinCB` and `RS-SlateGLinCB` achieve regret bounds of $\mathcal{O}(Nd^{3/2}\sqrt{T})$ and $\mathcal{O}(Nd\sqrt{T})$, respectively. Notably, both bounds are independent of the nonlinearity parameter $\kappa$ that is typically found to scale the regret of GLM bandit algorithms. Our algorithms are computationally efficient, requiring only poly$(N)$ time per round despite $2^{\Omega(N)}$ possible slates. Simulations show our algorithms outperform existing baselines with limited adaptivity and remain competitive with `Slate-GLM-OFU`, a fully adaptive state-of-the-art algorithm. Notably, a slightly modified `B-SlateGLinCB` empirically matches this baseline. Finally, we demonstrate strong performance in a practical in-context example selection task for language models.

[1]Microsoft Research India, Bengaluru. Correspondence to: <gauravsinha@microsoft.com>.

*Proceedings of the 43rd International Conference on Machine Learning*, Seoul, South Korea. PMLR 306, 2026. Copyright 2026 by the author(s).

## 1. Introduction

The online slate bandit framework models sequential decision-making where a learner must select a slate of items in each round. A slate is formed by choosing one item for each of several slots, with each slot having a distinct and potentially dynamic pool of candidate items. Following the selection, the learner observes a single reward for the entire slate (bandit feedback). The learner aims to design a selection policy that maximizes cumulative reward, or equivalently, minimizes cumulative regret over a horizon of $T$ rounds. This framework is well-suited for many real-world applications such as landing page optimization, where page components are selected to maximize conversions (Hill et al., 2017), and dynamic ad creative optimization, where advertisements are automatically assembled from various elements (Chen et al., 2021).

Such broad applications of online slate bandits have led to extensive research across various settings. When the item sets for each slot remain constant throughout the horizon and semi-bandit feedback (individual item level rewards within a slate) is provided, efficient and low-regret algorithms are well-known (Kale et al.; Rhuggenaath et al., 2020). Recently, Dimakopoulou et al. (2019) devised an efficient Thompson Sampling method for the fixed-item-set scenario that accommodates bandit feedback by attributing the same slate-level reward to all items within the chosen slate. Subsequently, Goyal & Sinha (2025) explored the stochastic contextual setting, characterized by item sets varying stochastically over time, and utilized bandit feedback from a logistic model. Their proposed algorithm, `Slate-GLM-OFU`, efficiently navigates an exponentially large space of candidate slates through an optimistic selection process for each slot. It achieves optimal regret, provided a "diversity" assumption holds for the sequence of chosen items, thus achieving strong theoretical and empirical performance for the contextual logistic slate bandit problem under bandit feedback.

Despite these algorithmic advances, critical challenges remain for deploying these methods in practice. Web-scale applications, such as online advertising and real-time recommendations, require bandit algorithms to operate with limited adaptivity. Two popular limited adaptivity settings studied in literature are: (a) **Batched** - The algorithm must partition the horizon $\{1, \ldots, T\}$ into very few intervals (batches)

and its policy during a batch should only depend on observations (slates selected and rewards received) from the previous batches, and (b) **Rarely-Switching** - The algorithm adaptively (and rarely) decides when to update its estimate of the reward parameters. While both the settings clearly offer practical efficiency by reducing the number of parameter estimations, the batched setting also enables parallelization, i.e., rounds within a batch can be executed independently of each other, significantly improving throughput.

Motivated by these challenges, we tackle the online contextual slate bandit problem in both these settings of limited adaptivity. Further, we assume that the environment provides a single reward for the selected slate (bandit feedback), generated by a Generalized Linear Model (GLM) with unknown parameters. We summarize our contributions below.

### 1.1. Our Contributions

First, in Section 3, we present `B-SlateGLinCB` (Algorithm 1), a batched algorithm for the Contextual Slate Bandit problem with GLM rewards, that operates over $\mathcal{O}(\log \log T)$ batches. We prove that, if the sets of items are chosen stochastically, then, at the end of $T$ rounds, under a popular diversity assumption (Assumption 2.1), `B-SlateGLinCB` incurs $\tilde{\mathcal{O}}(Nd^{3/2}\sqrt{T})$ regret, where each item is represented by a $d$-dimensional feature vector. In Algorithm 3, Appendix B, we also show an alternate approach using Distributional Optimal Designs (Ruan et al., 2021), and obtain a regret guarantee of $\tilde{\mathcal{O}}(Nd\sqrt{T}\min\{\sqrt{d}, \sqrt{N}\})$.

Next, in Section 4, we present `RS-SlateGLinCB` (Algorithm 2), a rarely-switching algorithm for the Contextual Slate Bandit problem with GLM rewards, that estimates reward parameters only $\mathcal{O}(Nd \log T)$ times. We prove that, at the end of $T$ rounds, for adversarially chosen item sets, under the same diversity assumption as above, `RS-SlateGLinCB` incurs $\mathcal{O}(Nd\sqrt{T})$ regret, matching the regret bound of `Slate-GLM-OFU` (Algorithm 1, Goyal & Sinha (2025)) which estimates parameters at all $T$ rounds, i.e., is not constrained by limited adaptivity.

A key feature of both our algorithms is per-round efficiency. They exhibit poly($N$) per round time complexity by selecting the items for the slots independently of each other. By doing so, they avoid iterating over the $2^{\Omega(N)}$ set of possible slates, making them practically feasible when $N$ is large.

Finally, in Section 5, under diverse experimental settings, we empirically demonstrate that both our algorithms achieve sublinear regret, significantly outperform other limited adaptivity baselines, and that `RS-SlateGLinCB` is quite competitive with `Slate-GLM-OFU`, a fully adaptive algorithm. We also propose `B-SlateGLinCB+`, a batched algorithm with slight modifications to `B-SlateGLinCB`, and show that its regret matches that of `Slate-GLM-OFU`. Using

`B-SlateGLinCB+`, we implement prompt tuning on language models with exemplar selection. We demonstrate strong performance in binary classification tasks and show that our performance matches that of `Slate-GLM-OFU`.

### 1.2. Related Work

**Slate Bandits**: Due to their practical relevance in real-world applications such as recommendation systems and advertising, slate bandits have recently attracted considerable attention (Chen et al., 2021; Hill et al., 2017). However, many of these works lack rigorous theoretical foundations, which have been explored in a separate line of research (Wang et al., 2017; Kale et al.; Rhuggenaath et al., 2020; Lagrée et al., 2016; Dimakopoulou et al., 2019; Goyal & Sinha, 2025). While several of these works (Wang et al., 2017; Lagrée et al., 2016; Rhuggenaath et al., 2020; Kale et al.) assume semi-bandit feedback (a reward for for each item chosen in the slate), more recent efforts such as Dimakopoulou et al. (2019) and Goyal & Sinha (2025) address the challenging slate-level bandit feedback scenario. In particular, Dimakopoulou et al. (2019) use a heuristic-based method to attribute the bandit feedback to each of the slots, while Goyal & Sinha (2025) decompose the selection rule into a slot-level selection rule, allowing their algorithms to avoid iterating over the exponential sized set of candidate slates while still obtaining optimal regret guarantees. However, these algorithms update their parameters at each round, and hence, are not easily adaptable to limited adaptivity settings.

**Limited Adaptivity**: Recently, there has been considerable interest in the batched and rarely-switching limited adaptivity settings. In the multi-armed bandit setting, several works have studied the advantages of batching (Cesa-Bianchi et al., 2013; Perchet et al., 2015; Gao et al., 2019), Subsequently, Ruan et al. (2021) proposed batched algorithms for contextual linear bandits, by introducing distributional optimal designs, and using them to guide and determine policy updates. Building on these ideas, recent work has explored batched algorithms for more complex reward models, including GLMs (Sawarni et al., 2024) and multinomial logit (MNL) models (Midigeshi et al., 2025). The rarely-switching setting for contextual linear bandits was introduced by Abbasi-Yadkori et al. (2011) and since, has been studied for other reward models as well (Sawarni et al., 2024; Midigeshi et al., 2025). However, these algorithms do not easily extend to the slate bandit setting, where combinatorial action spaces and structured feedback introduce unique challenges not addressed in prior batched bandit literature.

## 2. Notations and Problem Setup

In this section, we define some general notations and describe the problem setup in complete detail.

We represent the sets $\{1, \ldots, N\}$ and $\{m, \ldots, N\}$ as $[N]$ and $[m, N]$ respectively. Unless otherwise specified, all vectors, matrices, and sets are represented using bold lower case, bold upper case, and calligraphic upper case letters respectively. A matrix $\boldsymbol{A}$ is said to be positive semi-definite (p.s.d), denoted $\boldsymbol{A} \succeq 0$, if all the eigenvalues of $\boldsymbol{A}$ are non-negative. We define the norm of a vector $\boldsymbol{x}$ with respect to a p.s.d matrix $\boldsymbol{A}$ as $\|\boldsymbol{x}\|_{\boldsymbol{A}} = \sqrt{\boldsymbol{x}^\top \boldsymbol{A} \boldsymbol{x}}$. We use $\mathbb{P}$ and $\mathbb{E}$ to denote the probability and expectation of a quantity respectively. For any vector $\boldsymbol{x} = (x_1^1, \ldots, x_d^1, \ldots, x_1^N, \ldots x_d^N) \in \mathbb{R}^{Nd}$, $\boldsymbol{x}^i = (x_1^i, \ldots, x_d^i) \in \mathbb{R}^d$ denotes the $i^{th}$ block of $\boldsymbol{x}$. Finally, we use $\tilde{\mathcal{O}}(.)$ to suppress polylogarithmic factors.

## 2.1. Contextual Slate Bandits

Let $T \in \mathbb{N}$ denote the total number of rounds of interaction between a learner and an environment. In the contextual slate bandit problem, at each round $t \in [T]$, the learner is presented with $N$ sets of items $\mathcal{X}_t^1, \ldots \mathcal{X}_t^N \subset \mathbb{R}^{Nd}$. For each $i \in [N]$, the learner selects an item $\boldsymbol{x}_t^i \in \mathcal{X}_t^i$, thereby constructing a slate $\boldsymbol{x} = (\boldsymbol{x}_t^1, \ldots, \boldsymbol{x}_t^N) \in \mathcal{X}_t := \mathcal{X}_t^1 \times \ldots \times \mathcal{X}_t^N$. We say item $\boldsymbol{x}_t^i$ is used in the $i^{th}$ "slot" on the slate. The environment then reveals to her a single scalar $r_t(\boldsymbol{x}_t)$. The goal of the learner is to minimize her cumulative regret $R(T)$, defined as,

$$R(T) = \mathbb{E}\left[\sum\nolimits_{t \in [T]} \max_{\boldsymbol{x} \in \mathcal{X}_t} r_t(\boldsymbol{x}) - r_t(\boldsymbol{x}_t)\right], \quad (1)$$

where the expectation is over the randomness in the rewards.

## 2.2. Generalized Linear Models (GLMs)

We follow the definition of GLMs provided in Definition 2.1, Sawarni et al. (2024). Let $r \in \mathbb{R}$ be a random variable and $\boldsymbol{x} \in \mathbb{R}^{Nd}$ be a random vector in the Euclidean space. We say that $r(\boldsymbol{x})$ is sampled from a GLM, if, the conditional random variable $r \mid \boldsymbol{x}$ is distributed as per an exponential distribution, i.e.,

$$\mathbb{P}_{\boldsymbol{\theta}^\star}(r \mid \boldsymbol{x}) = \exp\left(r(\boldsymbol{x}^\top \boldsymbol{\theta}^\star) - b(\boldsymbol{x}^\top \boldsymbol{\theta}^\star) + c(r)\right).$$

Here $\boldsymbol{\theta}^\star \in \mathbb{R}^{Nd}$ parametrizes the density function. Further, following Sawarni et al. (2024), we assume that $b$ is twice-differentiable, $\dot{b}$ is assumed to be monotonic, and $r \in [0, R]$ almost surely, for some known $R \in \mathbb{R}$.

We define the link function $\mu$ as $\mu(\boldsymbol{x}_t^\top \boldsymbol{\theta}^\star) = \mathbb{E}[r_t \mid \boldsymbol{x}_t] = \dot{b}(\boldsymbol{x}_t^\top \boldsymbol{\theta}^\star)$. Thus, $\mu$ is monotonic, and further, following Filippi et al. (2010), we assume it to be $L_\mu$-Lipschitz. A significant property of GLMs is the *self-concordance* property (Sawarni et al., 2024), i.e, for GLMs supported on $[0, R]$, $|\ddot{\mu}(z)| \leq R\dot{\mu}(z) \; \forall \; z \in \mathbb{R}$.

## 2.3. Contextual Slate Bandits with Limited Adaptivity

In the limited adaptivity setting, the learner is constrained to make $M$ policy updates. Our goal is to solve the contex-

tual bandit problem with GLM rewards parametrized by an unknown parameter vector $\boldsymbol{\theta}^\star$ in both the prevalent limited adaptivity settings: *batched* and *rarely-switching*. Next, we formally describe these settings.

**Batched Slate GLM Bandits :** The learner is required to partition the horizon $[T]$ into $M$ disjoint batches $\mathcal{T}_1, \ldots, \mathcal{T}_M$ *apriori*, and the policy of selecting slates can only be updated between two consecutive batches. Therefore, during round $t \in \mathcal{T}_m$, the policy can only utilize the set of observations from previous batches $\{\mathcal{T}_i\}_{i=1}^{m-1}$ and the present set of items $\{\mathcal{X}_t^i\}_{i \in [N]}$, allowing for parallelization within a batch. It is known that when $M = \Omega(\log \log T)$, $\Theta(\log \log T)$ batches suffice to obtain optimal regret in $T$ (Cesa-Bianchi et al., 2013; Gao et al., 2019). Hence, we develop algorithms that make $\mathcal{O}(\log \log T)$ updates[1]. Further, we assume that in each slot $i \in [N]$, the set of items $\mathcal{X}_t^i \; \forall t \in [T]$ are sampled independently from a distribution $\mathcal{D}^i$ supported on $\mathbb{R}^d$. The goal of the learner is to minimize the expected cumulative regret defined in (1), where the expectation also incorporates the randomness in all the item sets $\{\mathcal{X}_t^i\}_{t \in [T], i \in [N]}$.

**Rarely-Switching Slate GLM Bandits:** Here, while the learner is constrained to estimate $\boldsymbol{\theta}^\star$ only $M$ times, she can adaptively decide when to make these estimates. We present an algorithm that makes $M = \mathcal{O}(\log T)$ policy updates, matching the lower bound in Ruan et al. (2021) up to polylog factors. We do not assume any stochasticity in the item sets, i.e., they can be adversarial. Our goal is to minimize the expected cumulative regret as defined in (1).

## 2.4. Non-Linearity Parameter $\kappa$

An important quantity that often arises while dealing with GLMs is an instance-dependent non-linear parameter $\kappa$, defined as

$$\kappa = \sup_{\boldsymbol{x} \in \mathcal{X}} \sup_{\boldsymbol{\theta}:\|\boldsymbol{\theta}\| \leq S} \frac{1}{\dot{\mu}(\boldsymbol{x}^\top \boldsymbol{\theta})}, \quad (2)$$

where $\mathcal{X}$ is the set of all actions (slates, in our case) across all rounds, and $S$ is an upper bound on $\|\boldsymbol{\theta}^\star\|_2$. Intuitively, $\kappa$ quantifies the deviation of the reward model from linearity, and can be exponential in $\|\boldsymbol{\theta}^\star\|_2$ (Faury et al., 2020). As a result, several works utilizing non-linear reward models in batched as well as non-batched settings (Faury et al., 2022; Sawarni et al., 2024; Zhang & Sugiyama, 2023; Midigeshi et al., 2025; Goyal & Sinha, 2025) have focused on achieving $\kappa$-free regret bounds (in the leading term).

## 2.5. Additional Assumptions

Following the works of several other GLM bandit papers, we assume that the norm of the hidden reward parameter

---

[1]When $M = o(\log \log T)$, our algorithms easily extend to the generic schedule presented in Sawarni et al. (2024).

$\|\boldsymbol{\theta}^\star\|_2 \leq S$, with $S$ being known. Also, following Goyal & Sinha (2025), for all rounds $t \in [T]$ and all slots $i \in [N]$, for any $\boldsymbol{z} \in \mathcal{X}_t^i$, we have $\|\boldsymbol{z}\|_2 \leq \frac{1}{\sqrt{N}}$. While these assumptions are somewhat standard in the bandit literature, we make an additional "diversity" assumption described below.

**Assumption 2.1** (Diversity Assumption). We assume that our algorithm ensures that the sequence of items selected are "diverse" enough, i.e. for all slots $i \in [N]$ and some $\rho > 0$

$$\mathbb{E}[\boldsymbol{x}_t^i \mid \mathcal{F}_{t-1}] = \boldsymbol{0} \quad \text{and} \quad \mathbb{E}[\boldsymbol{x}_t^i \boldsymbol{x}_t^{i\top} \mid \mathcal{F}_{t-1}] \succeq \rho \boldsymbol{I},$$

where $\boldsymbol{0}$ and $\boldsymbol{I}$ represent the zero vector and the identity matrix, while the filtration $\mathcal{F}_{t-1} = \sigma(\boldsymbol{x}_1, r_1, \ldots, \boldsymbol{x}_{t-1}, r_{t-1})$ encodes all the information up till time $t$.

Note that Goyal & Sinha (2025) assume that the eigenvalues of the expected design matrix grow as $\Omega(\kappa)^2$. We show that it suffices to assume that the eigenvalues grow as $\Omega(1)$, hence, matching the diversity assumptions made in relevant linear bandit literature (Das & Sinha, 2024; Chatterji et al., 2020; Bastani et al., 2021; Kannan et al., 2018; Raghavan et al., 2018; Papini et al., 2021). Thus, our assumption is strictly weaker than the one in Goyal & Sinha (2025).

Intuitively, these conditions ensure that the items chosen in each slot span the entire space in a way that the associated design matrices have eigenvalues sufficiently bounded away from zero. Such a diversity assumption has been used in several prior works (Goyal & Sinha, 2025; Das & Sinha, 2024; Papini et al., 2021; Chatterji et al., 2020) to obtain strong regret bounds. Similar to Goyal & Sinha (2025), we use this assumption to prove that the eigenvalues of the design matrices used in our algorithms grow (sufficiently) linearly. We refer the reader to Section 2.1 of Goyal & Sinha (2025) for a thorough discussion of the diversity assumption. Also, in Appendix H, we empirically validate the linear growth of eigenvalues of the design matrices for our algorithms.

### 2.6. GLM-MLE Loss

Let $\boldsymbol{x}_s \subset \mathbb{R}^{Nd}$ be the slate selected at round $s \in [t-1]$ and $r_s$ be the corresponding reward. The maximum likelihood estimator (MLE), $\widehat{\boldsymbol{\theta}}_t$, based on these observations, is the maximizer of the function,

$$\sum_{s=1}^{t-1} \log \mathbb{P}_{\boldsymbol{\theta}}(r_s \mid \boldsymbol{x}_s) = \sum_{s=1}^{t-1} r_s \cdot \boldsymbol{x}_s^\top \boldsymbol{\theta} - \mu(\boldsymbol{x}_s^\top \boldsymbol{\theta}). \quad (3)$$

We refer the readers to Sections 2 and 3 of Filippi et al. (2010) for more details. Note that (3) is an unconstrained optimization problem. If the MLE $\widehat{\boldsymbol{\theta}}_t$ lies outside the set of

admissible parameters $\Theta = \{\boldsymbol{\theta} : \|\boldsymbol{\theta}\|_2 \leq S\}$, we project $\widehat{\boldsymbol{\theta}}_t$ back on to $\Theta$, worsening the regret by $poly(R, S)$ (see Appendix E of Sawarni et al. (2024) for a more detailed explanation). Henceforth, for the sake of exposition, we assume $\widehat{\boldsymbol{\theta}}_t \in \Theta, \forall t \in [T]$. However, all results easily extend to include the projection described above.

### 2.7. G-Optimal Design

Let $\mathcal{X} \subset \mathbb{R}^d$. The G-Optimal design $\pi_G(\mathcal{X})$ is a probability distribution on $\mathcal{X}$ defined as

$$\pi_G(\mathcal{X}) = \underset{\pi \in \Delta(\mathcal{X})}{\arg\min} \max_{\boldsymbol{x} \in \mathcal{X}} \|\boldsymbol{x}\|_{\boldsymbol{V}^{-1}}^2 \text{ where } \boldsymbol{V} = \mathbb{E}_\pi[\boldsymbol{x}\boldsymbol{x}^\top].$$

Here, $\Delta(\mathcal{X})$ is the set of all probability distributions over $\mathcal{X}$. The Keifer-Wolfowitz theorem (Kiefer & Wolfowitz, 1959) states that $\max_{\boldsymbol{x} \in \mathcal{X}} \|\boldsymbol{x}\|_{\boldsymbol{V}(\pi_G(\mathcal{X}))^{-1}}^2 \leq d$. While computing an exact G-optimal design is known to be NP-hard (Grötschel et al., 2012; Summa et al., 2014), there exist efficient[3] algorithms to compute approximate optimal designs (see Chapter 21, Lattimore & Szepesvari (2017)).

## 3. B-SlateGLinCB

In this section, we present a batched algorithm for Contextual Slate GLM Bandits, which we refer to as B-SlateGLinCB (Algorithm 1). First, we provide a detailed explanation of the algorithm, highlighting the non-trivialities involved in a multi-slot batched algorithm. Next, in Section 3.2, we provide a regret guarantee for it. Finally, in Section 3.3, we make some additional remarks.

### 3.1. Algorithmic Description

B-SlateGLinCB takes as inputs the number of slots $N$, the number of batches $M$, the length of the horizon $T$, an upper bound $S$ on the $\ell_2$-norm of the true reward parameter $\boldsymbol{\theta}^\star$, the probability of failure $\delta$, and the instance-dependent nonlinearity parameter $\kappa$.[4] As discussed in Section 2.3, when the number of batches is $\Omega(\log \log T)$, we only require $M = \Theta(\log \log T)$ batches. Hence, without loss of generality, we develop our algorithm for $M = \mathcal{O}(\log \log T)$ batches. First, in *Step 3*, we define the $M + 1$ batches $\mathcal{T}_0, \ldots, \mathcal{T}_M$. and define the batches to be consecutive disjoint subsets of $[T]$ with lengths given as,

$$|\mathcal{T}_0| = \lfloor \sqrt{T} \rfloor \quad , \quad |\mathcal{T}_m| = \lfloor T^{1-2^{-m}} \rfloor \quad \forall m \in [M]. \quad (4)$$

**Warm-up Batch**: We begin with a warm-up batch $\mathcal{T}_0$ (*Steps 4-11*). At each round $t \in \mathcal{T}_0$, for each slot $i \in [N]$, we receive the set of items $\mathcal{X}_t^i$. Then, for each slot $i \in [N]$, the algorithm samples an item $\boldsymbol{x}_t^i$ from a G-optimal design

---

[2]i.e, Goyal & Sinha (2025) assume $\mathbb{E}[\boldsymbol{x}_t^i \boldsymbol{x}_t^{i\top} \mid \mathcal{F}_{t-1}] \succeq \rho\kappa\boldsymbol{I}$ for some $\rho > 0$.

[3]polynomial in $|\mathcal{X}|$ and $d$.

[4]An upper bound on $\kappa$ and $S$ suffices (Sawarni et al., 2024).

**Algorithm 1** `B-SlateGLinCB`

1: **Inputs:** Number of Slots $N$, Number of batches $M$, Horizon $T$, Parameter norm bound $S$, Failure Level $\delta$, and non-linearity $\kappa$.

2: Initialize $\boldsymbol{\theta}_0 = \mathbf{0}_{Nd}$ and $\lambda = \mathcal{O}(NdR^2 \log T/\delta)$.

3: Define batches $\mathcal{T}_0, \mathcal{T}_1, \ldots, \mathcal{T}_M$ as per (4).

4: {`Warmup Batch`}

5: **for** $t \in \mathcal{T}_0$ **do**

6:    Receive the set of items $\mathcal{X}_t^i$ for all slots $i \in [N]$.

7:    **for** $i \in [N]$ **do**

8:       Obtain $\boldsymbol{x}_t^i \sim \pi_G(\mathcal{X}_t^i)$ (defined in Section (2.7)).

9:    **end for**

10:    Play the slate $\boldsymbol{x}_t = (\boldsymbol{x}_t^1, \ldots, \boldsymbol{x}_t^N)$ and obtain $r_t$.

11: **end for**

12: Compute $\widehat{\boldsymbol{\theta}}_0 = \arg\min_{\boldsymbol{\theta}} \sum_{t \in \mathcal{T}_0} \ell(\boldsymbol{x}_t, r_t, \boldsymbol{\theta})$ as per (3).

13: Define $\boldsymbol{V}_0^i = \lambda \boldsymbol{I}_d + \sum_{t \in \mathcal{T}_0} \boldsymbol{x}_t^i \boldsymbol{x}_t^{i\top}$ for all slots $i \in [N]$.
   //`Other batches`

14: **for** $m \in [M]$ **do**

15:    **for** $t \in \mathcal{T}_m$ **do**

16:       Receive the set of items $\mathcal{X}_t^i$ for all slots $i \in [N]$.

17:       **for** $i \in [N]$ **do**

18:          **for** $l \in [0, m-1]$ **do**

19:             {`Perform elimination`}

20:             $\mathcal{X}_t^i \leftarrow \{\boldsymbol{z} \in \mathcal{X}_t^i : \text{UCB}^{i,k}(\boldsymbol{z}) \geq \max_{\boldsymbol{y} \in \mathcal{X}_t^i} \text{LCB}^{i,k}(\boldsymbol{y})\}$.

21:          **end for**

22:          Obtain $\boldsymbol{x}_t^i \sim \pi_G(\mathcal{X}_t^i)$ (defined in Section (2.7)) and $\boldsymbol{b}_t^i = \arg\max_{\boldsymbol{y} \in \mathcal{X}_t^i} \|\boldsymbol{y}\|_{(\boldsymbol{V}_0^i)^{-1}}$.

23:       **end for**

24:       Play the slate $\boldsymbol{x}_t = (\boldsymbol{x}_t^1, \ldots, \boldsymbol{x}_t^N)$ and obtain $r_t$.

25:       Construct the slate $\boldsymbol{b}_t = (\boldsymbol{b}_t^1, \ldots, \boldsymbol{b}_t^N)$.

26:    **end for**

27:    $\widehat{\boldsymbol{\theta}}_m = \arg\min_{\boldsymbol{\theta}} \sum_{t \in \mathcal{T}_m} \ell(\boldsymbol{x}_t, r_t, \boldsymbol{\theta})$ as per (3).

28:    $\boldsymbol{H}_m^i = \lambda \boldsymbol{I}_d + \sum_{t \in \mathcal{T}_m} \frac{\dot{\mu}(\boldsymbol{b}_t^\top \widehat{\boldsymbol{\theta}}_0)}{\beta_t} \boldsymbol{x}_t^i \boldsymbol{x}_t^{i\top} \ \forall i \in [N]$ (9).

29: **end for**

---

as per (3) computed over $\mathcal{X}_t^i$ and plays the resultant slate $\boldsymbol{x}_t = (\boldsymbol{x}_t^1, \ldots, \boldsymbol{x}_t^N)$, receiving feedback $r_t$. At the end of this batch, in *Step 12*, we compute an estimate $\widehat{\boldsymbol{\theta}}_0$ of the reward parameters $\boldsymbol{\theta}^\star$, by minimizing the GLM-MLE loss as per (3) over the set $\{(\boldsymbol{x}_t, r_t)\}_{t \in \mathcal{T}_0}$. Then, in *Step 13*, for all slots $i \in [N]$, we compute the design matrices $\boldsymbol{V}_0^i = \lambda \boldsymbol{I}_d + \sum_{t \in \mathcal{T}_0} \boldsymbol{x}_t^i \boldsymbol{x}_t^{i\top}$ for all the items chosen in the $i^{th}$ slot in batch $\mathcal{T}_0$. Here, $\lambda = \mathcal{O}(NdR^2 \log(T/\delta))$. The estimate $\widehat{\boldsymbol{\theta}}_0$ and matrices $\boldsymbol{V}_0^i$ are utilized in the subsequent batches to control regret.

**Batches** $m \in [M]$: In *Steps 15-28*, we execute the $m^{th}$ batch ($m \in [M]$). For each round $t \in \mathcal{T}_m$, we receive the $N$ sets of items $\{\mathcal{X}_t^i\}_{i \in [N]}$. For each slot $i \in [N]$, we prune $\mathcal{X}_t^i$ (*Steps 17-21*) based on a criterion we discuss next. For

any slot $i \in [N]$, item $\boldsymbol{z} \in \mathcal{X}_t^i$ and a prior batch $l \in [0, m-1]$, define the scores $\text{UCB}^{i,l}(\boldsymbol{z})$ and $\text{LCB}^{i,l}(\boldsymbol{z})$ (upper and lower confidence bounds respectively) as follows:

$$\text{UCB}^{i,l}(\boldsymbol{z}) = \begin{cases} \boldsymbol{z}^\top \widehat{\boldsymbol{\theta}}_0^i + 2\sqrt{\kappa}\gamma \|\boldsymbol{z}\|_{(\boldsymbol{V}_0^i)^{-1}} & l = 0, \\ \boldsymbol{z}^\top \widehat{\boldsymbol{\theta}}_l^i + 2\gamma \|\boldsymbol{z}\|_{(\boldsymbol{H}_l^i)^{-1}} & l \neq 0. \end{cases} \quad (5)$$

$$\text{LCB}^{i,l}(\boldsymbol{z}) = \begin{cases} \boldsymbol{z}^\top \widehat{\boldsymbol{\theta}}_0^i - 2\sqrt{\kappa}\gamma \|\boldsymbol{z}\|_{(\boldsymbol{V}_0^i)^{-1}} & l = 0, \\ \boldsymbol{z}^\top \widehat{\boldsymbol{\theta}}_l^i - 2\gamma \|\boldsymbol{z}\|_{(\boldsymbol{H}_l^i)^{-1}} & l \neq 0. \end{cases} \quad (6)$$

Here, $\gamma = \mathcal{O}(SR\sqrt{Nd\log(T/\delta)})$. In the definitions above (for $l \neq 0$), the first term (i.e. $\boldsymbol{z}^\top \widehat{\boldsymbol{\theta}}_l^i$) utilizes an estimate $\widehat{\boldsymbol{\theta}}_l = (\widehat{\boldsymbol{\theta}}_l^1, \ldots, \widehat{\boldsymbol{\theta}}_l^N)$ of the true reward parameters $\boldsymbol{\theta}^\star$. This estimate is calculated in *Step 27* by minimizing the GLM-MLE loss as per (3) over the set $\{(\boldsymbol{x}_t, r_t)\}_{t \in \mathcal{T}_l}$ at the end of the $l^{th}$ batch. For $l \neq 0$, the second term in (5) and (6) (i.e., $\pm 2\gamma \|\boldsymbol{z}\|_{(\boldsymbol{H}_l^i)^{-1}}$) utilizes a slot-level scaled design matrix

$$\boldsymbol{H}_l^i = \lambda \boldsymbol{I}_d + \sum_{t \in \mathcal{T}_l} \frac{\dot{\mu}(\boldsymbol{b}_t^\top \widehat{\boldsymbol{\theta}}_0)}{\beta_t} \boldsymbol{x}_t^i \boldsymbol{x}_t^{i\top}, \quad (7)$$

Here, $\boldsymbol{b}_t = (\boldsymbol{b}_t^1, \ldots, \boldsymbol{b}_t^N)$ is a slate (called *scaling-slate*) computed at round $t$, with its $i^{th}$ item defined as

$$\boldsymbol{b}_t^i = \arg\max_{\boldsymbol{z} \in \mathcal{X}_t^i} \|\boldsymbol{z}\|_{(\boldsymbol{V}_0^i)^{-1}}. \quad (8)$$

This *scaling-slate* $\boldsymbol{b}_t$ is used to construct the scaling term $\dot{\mu}(\boldsymbol{b}_t^\top \widehat{\boldsymbol{\theta}}_0)$ in $\boldsymbol{H}_l^i$, which is then normalized using

$$\beta_t = \exp\left(\min\left\{2S, 6\sqrt{\kappa}\gamma \sum_{i \in [N]} \|\boldsymbol{b}_t^i\|_{(\boldsymbol{V}_0^i)^{-1}}\right\}\right). \quad (9)$$

We provide more details on the choice of this scaled design matrices $\boldsymbol{H}_l^i$ in Section 3.1.1. Finally, using the scores defined in (5) and (6), we eliminate all items $\boldsymbol{z} \in \mathcal{X}_t^i$ for which $\text{UCB}^{i,l}(\boldsymbol{z}) < \text{LCB}^{i,l}(\boldsymbol{y})$ for some previous batch $l \in [0, m-1]$ and some $\boldsymbol{y} \in \mathcal{X}_t^i$. Then, in *Step 22*, we construct a G-optimal design on the remaining items in $\mathcal{X}_t^i$ and sample an item $\boldsymbol{x}_t^i$ from it. After completing this procedure for all slots, we play the constructed slate $\boldsymbol{x}_t = (\boldsymbol{x}_t^1, \ldots, \boldsymbol{x}_t^N)$ and receive reward $r_t$ for it. We then compute the scaled design matrices $\boldsymbol{H}_m^i$ (*Step 28*), which will ultimately be used in eliminations performed during batches $l \in [m+1, M]$.

### 3.1.1. SCALED MATRICES $\boldsymbol{H}_m^i$

As described earlier, the scores $\text{UCB}^{i,l}$ and $\text{LCB}^{i,l}$ for slot $i \in [N]$ and batch $l \in [M]$, used during elimination (*Step 19-20* in Algorithm 1) utilize a slot-level scaled design matrix $\boldsymbol{H}_l^i$ defined in (7). Moreover, the slate is also constructed by sampling items (*Step 22*) for each slot separately. This ensures that the per-round time complexity grows as $\text{poly}(K, N)$. We now explain why it also helps us obtain a

$\kappa$-free optimal regret guarantee. First, using Assumption 2.1 along with a recent technique from Goyal & Sinha (2025), in Lemma A.13 (Appendix A) we show that the block diagonal matrix $diag(\boldsymbol{H}_l^1, \ldots, \boldsymbol{H}_l^N)$ is multiplicatively equivalent to the matrix

$$\tilde{\boldsymbol{H}}_l = \lambda \boldsymbol{I}_{Nd} + \sum_{t \in \mathcal{T}_l} \left( \dot{\mu}(\boldsymbol{b}_t^\top \widehat{\boldsymbol{\theta}}_0)/\beta_t \right) \boldsymbol{x}_t \boldsymbol{x}_t^T,$$

where $\{\boldsymbol{x}_t\}_{t \in \mathcal{T}_l}$ is the sequence of actions played during the $l^{th}$ batch and $\beta_t$ is the normalization term defined in (9). Then, we show that $\tilde{\boldsymbol{H}}_l \preccurlyeq \boldsymbol{H}_l^\star$, where $\boldsymbol{H}_l^\star$ is the hessian of the GLM-MLE loss (Section 2.6) computed on $\{(\boldsymbol{x}_t, r_t)\}_{t \in \mathcal{T}_l}$, i.e.,

$$\boldsymbol{H}_l^\star = \lambda \boldsymbol{I}_{Nd} + \sum_{t \in \mathcal{T}_l} \dot{\mu}(\boldsymbol{x}_t^\top \boldsymbol{\theta}^\star) \boldsymbol{x}_t \boldsymbol{x}_t^\top.$$

It is well known from recent literature (Sawarni et al., 2024; Faury et al., 2022) that $\boldsymbol{H}_m^\star$ can be used to construct confidence sets for the MLE estimator $\widehat{\boldsymbol{\theta}}_l$ obtained by minimizing the GLM-MLE loss on $\{(\boldsymbol{x}_t, r_t)\}_{t \in \mathcal{T}_l}$. To be precise, for any $\delta \in (0, 1)$, we can show that

$$\mathbb{P}\left[ \|\widehat{\boldsymbol{\theta}}_l - \boldsymbol{\theta}^\star\|_{\boldsymbol{H}_l^\star} \leq \mathcal{O}(\sqrt{Nd \log(T/\delta)}) \right] \geq 1 - \delta. \quad (10)$$

Since at round $t \in \mathcal{T}_l$, we selected the item $\boldsymbol{x}_t^i$ for slot $i$ by sampling from a G-optimal design constructed on $\mathcal{X}_t^i$ (post elimination), we get the optimal design bound (Lemma A.10, Appendix A) as:

$$\mathbb{E}\left[ \left\| \boldsymbol{x}_t^i \sqrt{\dot{\mu}(\boldsymbol{b}_t^\top \widehat{\boldsymbol{\theta}}_0)/\beta_t} \right\|_{(\boldsymbol{H}_l^i)^{-1}} \right] = \mathcal{O}\left( \frac{d}{\sqrt{|\mathcal{T}_l|}} \right).$$

Multiplicative equivalence between $diag(\boldsymbol{H}_l^1, \ldots, \boldsymbol{H}_l^N)$ and $\tilde{\boldsymbol{H}}_l$, then yields that $\left\| \boldsymbol{x}_t \sqrt{\dot{\mu}(\boldsymbol{b}_t^\top \widehat{\boldsymbol{\theta}}_0)/\beta_t} \right\|_{(\tilde{\boldsymbol{H}}_l)^{-1}}$ and $\sum_{i \in [N]} \left\| \boldsymbol{x}_t^i \sqrt{\dot{\mu}(\boldsymbol{b}_t^\top \widehat{\boldsymbol{\theta}}_0)/\beta_t} \right\|_{(\boldsymbol{H}_l^i)^{-1}}$ are also multiplicatively equivalent, implying that

$$\mathbb{E}\left[ \left\| \boldsymbol{x}_t \sqrt{\dot{\mu}(\boldsymbol{b}_t^\top \widehat{\boldsymbol{\theta}}_0)/\beta_t} \right\|_{(\tilde{\boldsymbol{H}}_l)^{-1}} \right] = \mathcal{O}\left( \frac{Nd}{\sqrt{|\mathcal{T}_l|}} \right). \quad (11)$$

These bounds in (10) and (11) are key to proving $\kappa$-independent optimal (in $T$) regret guarantees.

### 3.2. Regret Guarantee for B-SlateGLinCB

In Theorem 3.1 we present our regret guarantee for B-SlateGLinCB and provide the proof in Appendix A.

**Theorem 3.1.** *At the end of $T$ rounds, B-SlateGLinCB (Algorithm 1) incurs a regret $R(T)$ which can be bounded as*

$$R(T) = \tilde{\mathcal{O}}\left( RSNd^{3/2} \sqrt{T \cdot \mathbb{E}_{\{\mathcal{X}^i \sim \mathcal{D}^i\}_{i \in [N]}} \dot{\mu}(\boldsymbol{x}_\star^\top \boldsymbol{\theta}^\star)} \right)$$

*where $\boldsymbol{x}_\star = \arg\max_{\boldsymbol{x} \in \mathcal{X}} \boldsymbol{x}^\top \boldsymbol{\theta}^\star$ and $\mathcal{X} = \mathcal{X}^1 \times \ldots \times \mathcal{X}^N$.*

### 3.3. Additional Remarks

*Remark* 3.2. Replacing the G-Optimal design by a Distributional Optimal design (Ruan et al., 2021) results in an algorithm (Algorithm 3, Appendix B), that incurs regret $R(T)$ bounded as

$$R(T) = \tilde{\mathcal{O}}\left( RSNd\sqrt{T} \cdot \sqrt{\min\left\{ \frac{d}{\kappa_1}, \frac{N}{\kappa_2} \right\}} \right),$$

where

$$\frac{1}{\kappa_1} = \mathbb{E}_{\{\mathcal{X}^i \sim \mathcal{D}^i\}_{i=1}^N} \dot{\mu}(\boldsymbol{x}_\star^\top \boldsymbol{\theta}^\star), \quad \frac{1}{\kappa_2} = \max_{\{\mathcal{X}^i \sim \mathcal{D}^i\}_{i=1}^N} \dot{\mu}(\boldsymbol{x}_\star^\top \boldsymbol{\theta}^\star),$$

and $\boldsymbol{x}_\star = \arg\max_{\boldsymbol{x} \in \mathcal{X}} \boldsymbol{x}^\top \boldsymbol{\theta}^\star$ and $\mathcal{X} = \mathcal{X}^1 \times \ldots \times \mathcal{X}^N$. Thus, an improved dependence on $d$ simultaneously worsens the dependence on $N$ and the reward sensitivity of the optimal slates. We provide a more detailed explanation along with the proof in Appendix B.

*Remark* 3.3 (Novelty of *scaling-slate* $\boldsymbol{b}_t$). A natural extension of B-GLinCB (Algorithm 1, Sawarni et al. (2024)) to our setting is to scale each item $\boldsymbol{x}^i$ with $\dot{\mu}((\boldsymbol{x}^i)^\top \widehat{\boldsymbol{\theta}}_0^i)$. However, this results in terms of the form $\dot{\mu}((\boldsymbol{x}_t^i)^\top \widehat{\boldsymbol{\theta}}_0^i)/\dot{\mu}(\boldsymbol{x}_t^\top \widehat{\boldsymbol{\theta}}_0)$ which can grow linearly with $\kappa$ when $N > 1$. By choosing the scaling-slate as $\boldsymbol{b}_t$, we show that $\dot{\mu}(\boldsymbol{b}_t^\top \widehat{\boldsymbol{\theta}}_0)/\dot{\mu}(\boldsymbol{x}_t^\top \widehat{\boldsymbol{\theta}}_0) = \mathcal{O}(1)$, helping avoid a potential $\kappa$-dependency.

## 4. RS-SlateGLinCB

In this section, we describe a rarely-switching algorithm for Slate GLM Bandits, which we refer to as RS-SlateGLinCB (Algorithm 2). RS-SlateGLinCB employs a switching condition to adaptively determine policy updates. We first describe the algorithm, and subsequently, in Section 4.2, we provide provide certain guarantees for RS-SlateGLinCB and make some remarks.

### 4.1. Algorithmic Description

RS-SlateGLinCB takes as inputs the number of slots $N$, the length of the horizon $T$, an upper bound $S$ on the $\ell_2$-norm of the true reward parameters $\boldsymbol{\theta}^\star$, the probability of failure $\delta$, and the instance-dependent non-linearity $\kappa$[5].

**Warm-up Batch**: The algorithm begins with a warm-up batch $\mathcal{T}_0$ (*Steps 3-7*) comprising $\lfloor \sqrt{T} \rfloor$ rounds. At each round $t \in \mathcal{T}_0$, in *Steps 4-6*, the algorithm observes the $N$ different item-sets $\mathcal{X}_t^i$, and for each slot $i \in [N]$, selects $\boldsymbol{x}_t^i = \arg\max_{\boldsymbol{z} \in \mathcal{X}_t^i} \|\boldsymbol{z}\|_{(\boldsymbol{V}_t^i)^{-1}}$, where $\boldsymbol{V}_t^i$ is defined via

$$\boldsymbol{V}_t^i = \boldsymbol{V}_{t-1}^i + \boldsymbol{x}_t^i \boldsymbol{x}_t^{i\top}, \boldsymbol{V}_0^i = \lambda \boldsymbol{I}$$

and $\lambda = \mathcal{O}(NdR^2 S^{-1} \log(T/\delta))$. The algorithm plays the resulting slate $\boldsymbol{x}_t = (\boldsymbol{x}_t^1, \ldots, \boldsymbol{x}_t^N)$ and receives the

---

[5]An upper bound on $\kappa$ and $S$ suffices.

**Algorithm 2** RS-SlateGLinCB

1: **Inputs:** Number of Slots $N$, Horizon $T$, Parameter norm bound $S$, Failure Level $\delta$, non-linearity $\kappa$.
2: Initialize the warm-up batch $\mathcal{T}_0 = \lfloor \sqrt{T} \rfloor$ and $\boldsymbol{V}^i = \lambda \boldsymbol{I}$, for all $i \in [N]$ where $\lambda = \mathcal{O}\left(NdR^2S^{-1}\log(T\delta^{-1})\right)$.
   //Warmup Batch
3: **for** $t \in [\mathcal{T}_0]$ **do**
4:    Receive the set of items $\mathcal{X}_t^i$ for all slots $i \in [N]$.
5:    Select $\boldsymbol{x}_t^i = \max_{\boldsymbol{x} \in \mathcal{X}_t^i} \|\boldsymbol{x}\|_{(\boldsymbol{V}_t^i)^{-1}}$, play the slate $\boldsymbol{x}_t = (\boldsymbol{x}_t^1, \dots, \boldsymbol{x}_t^N)$ and obtain reward $r_t$.
6:    For all $i \in [N]$, update $\boldsymbol{V}^i \leftarrow \boldsymbol{V}^i + \boldsymbol{x}_t^i \boldsymbol{x}_t^{i\top}$.
7: **end for**
8: Update $\widehat{\boldsymbol{\theta}}_0 = \arg\min_{\boldsymbol{\theta}} \sum_{t \in [\mathcal{T}_w]} \ell(\boldsymbol{x}_t, r_t, \boldsymbol{\theta})$.
9: Set $s = |\mathcal{T}_0|$, $\boldsymbol{H}_s = \lambda \boldsymbol{I}_{Nd}$, $\boldsymbol{H}_s^i = \lambda \boldsymbol{I}_d \; \forall i \in [N]$.
10: **for** $t \in [s+1, T]$ **do**
11:    Receive the set of items $\mathcal{X}_t^i$ for all slots $i \in [N]$.
       //Determinant condition
12:    **if** $\det(\boldsymbol{H}_t) \geq 2 \det(\boldsymbol{H}_s)$ **then**
13:       Compute $\widehat{\boldsymbol{\theta}}_s = \arg\min_{\boldsymbol{\theta}} \sum_{t'=s+1}^{t-1} \ell(\boldsymbol{x}_{t'}, r_{t'}, \boldsymbol{\theta})$ and update $s \leftarrow t$.
14:    **end if**
15:    **for** each slot $i \in [N]$ **do**
16:       $\mathcal{X}_t^i \leftarrow \{\boldsymbol{z} \in \mathcal{X}_t^i : \text{UCB}^{i,0}(\boldsymbol{z}) \geq \max_{\boldsymbol{y} \in \mathcal{X}_t^i} \text{LCB}^{i,0}(\boldsymbol{y})\}$.
17:       Select $\boldsymbol{x}_t^i = \arg\max_{\boldsymbol{z} \in \mathcal{X}_t^i} \{\boldsymbol{z}^\top \widehat{\boldsymbol{\theta}}_s^i + 2\sqrt{2}\beta \|\boldsymbol{z}\|_{(\boldsymbol{H}_t^i)^{-1}}\}$
18:    **end for**
19:    Play slate $\boldsymbol{x}_t = (\boldsymbol{x}_t^1, \dots, \boldsymbol{x}_t^N)$ and obtain reward $r_t$.
20:    Update the matrix $\boldsymbol{H}_{t+1} \leftarrow \boldsymbol{H}_t + \dot{\mu}(\boldsymbol{x}_t^\top \widehat{\boldsymbol{\theta}}_0) e^{-1} \boldsymbol{x}_t \boldsymbol{x}_t^\top$ and $\boldsymbol{H}_{t+1}^i \leftarrow \boldsymbol{H}_t^i + \dot{\mu}(\boldsymbol{x}_t^\top \widehat{\boldsymbol{\theta}}_0) e^{-1} \boldsymbol{x}_t^i \boldsymbol{x}_t^{i\top} \; \forall i \in [N]$.
21: **end for**

---

reward $r_t$ corresponding to it. At the end of this warm up batch, in *Step 8* we compute $\widehat{\boldsymbol{\theta}}_0$ by minimizing the GLM-MLE loss as per (3) over the set $\{(\boldsymbol{x}_t, r_t)\}_{t \in \mathcal{T}_0}$. Similar to B-SlateGLinCB, $\widehat{\boldsymbol{\theta}}_0$ is used to define scaled design matrices which help us in obtaining $\kappa$-free regret which we discuss next.

**Rarely-Switching Algorithm**: In *Steps 9-20*, we execute the rest of the rarely-switching algorithm. For each round $t \in [|\mathcal{T}_0| + 1, T]$ and each slot $i \in [N]$, we define a slot level scaled design matrix

$$\boldsymbol{H}_t^i = \lambda \boldsymbol{I}_d + \sum_{s=|\mathcal{T}_0|+1}^{t-1} \left(\dot{\mu}(\boldsymbol{x}_s^\top \widehat{\boldsymbol{\theta}}_0)/e\right) \boldsymbol{x}_s^i \boldsymbol{x}_s^{i\top}.$$

We also define a slate level scaled design matrix

$$\boldsymbol{H}_t = \lambda \boldsymbol{I}_{Nd} + \sum_{s=|\mathcal{T}_0|+1}^{t-1} \left(\dot{\mu}(\boldsymbol{x}_s^\top \widehat{\boldsymbol{\theta}}_0)/e\right) \boldsymbol{x}_s \boldsymbol{x}_s^\top.$$

Similar to our discussion in Section 3.1.1, we can show that $\boldsymbol{H}_t \preccurlyeq \boldsymbol{H}_t^\star$, where $\boldsymbol{H}_t^\star$ is the Hessian of the GLM-MLE loss

(Section 2.6) computed on the pairs $\{(\boldsymbol{x}_s, r_s)\}_{s=|\mathcal{T}_0|+1}^{t-1}$, and is defined as

$$\boldsymbol{H}_t^\star = \lambda \boldsymbol{I}_{Nd} + \sum_{s=|\mathcal{T}_0|+1}^{t-1} \dot{\mu}(\boldsymbol{x}_t^\top \boldsymbol{\theta}^\star) \boldsymbol{x}_s \boldsymbol{x}_s^\top.$$

After receiving the set of items $\mathcal{X}_t^i$ for all slots $i \in [N]$, in *Step 13*, we check for a *determinant condition* indicating whether the true parameter $\boldsymbol{\theta}^\star$ needs to be re-estimated. In particular, we check if the determinant of $\boldsymbol{H}_t$ is more than double the determinant of $\boldsymbol{H}_s$, where $s < t$ is the last round at which an estimate $\widehat{\boldsymbol{\theta}}_s$ of $\boldsymbol{\theta}^\star$ was computed. If true, we compute $\widehat{\boldsymbol{\theta}}_t$ by minimizing the GLM-MLE loss over all rounds $t \in [|\mathcal{T}_0| + 1, t - 1]$, and update $s = t$. Then, regardless of whether the determinant condition was true, in *Steps 16-17*, for each slot $i \in [N]$, we eliminate all items $\boldsymbol{z} \in \mathcal{X}_t^i$ that satisfy $\text{UCB}^{i,0}(\boldsymbol{z}) < \text{LCB}^{i,0}(\boldsymbol{y})$ for some item $\boldsymbol{y} \in \mathcal{X}_t^i$. Here, $\text{UCB}^{i,k}$ and $\text{LCB}^{i,k}$ are scores defined in (5) and (6) respectively. Finally, in Step *17*, from the remaining items in $\mathcal{X}_t^i$, we select $\boldsymbol{x}_t^i$ such that,

$$\boldsymbol{x}_t^i = \arg\max_{\boldsymbol{z} \in \mathcal{X}_t^i} \{\boldsymbol{z}^\top \widehat{\boldsymbol{\theta}}_s^i + 2\sqrt{2}\beta \|\boldsymbol{z}\|_{(\boldsymbol{H}_t^i)^{-1}}\}$$

where $\beta = \mathcal{O}(R\sqrt{NdS \log(T/\delta)})$ and $s \leq t$ was the last round at which an estimate $\widehat{\boldsymbol{\theta}}_s = (\widehat{\boldsymbol{\theta}}_s^1, \dots, \widehat{\boldsymbol{\theta}}_s^N)$ of $\boldsymbol{\theta}^\star$ was computed. We play the slate $\boldsymbol{x}_t = (\boldsymbol{x}_t^1, \dots, \boldsymbol{x}_t^N)$ and receive reward $r_t$. We then update $\boldsymbol{H}_t^i$ ($i \in [N]$) and $\boldsymbol{H}_t$.

### 4.2. Guarantees and Remarks for RS-SlateGLinCB

In Theorem 4.1 and Lemma 4.2, we present the regret guarantee for RS-SlateGLinCB and a bound on the number of parameter updates made by it respectively. We provide the proofs in Appendix C.

**Theorem 4.1.** *At the end of $T$ rounds, RS-SlateGLinCB (Algorithm 2) incurs a regret $R(T)$ which can be bounded as*

$$R(T) = \tilde{\mathcal{O}}\left(R\sqrt{T} + RS^{1/2}Nd\sqrt{\sum_{t \in [T]} \dot{\mu}(\boldsymbol{x}_{t,\star}^\top \boldsymbol{\theta}^\star)}\right).$$

**Lemma 4.2.** *During $T$ rounds, RS-SlateGLinCB (Algorithm 2) updates its policy at most $\mathcal{O}(Nd \log T)$ times.*

*Remark* 4.3 (Improvement in number of updates over Sawarni et al. (2024)). Our warm-up ensures that Algorithm 2 makes $\mathcal{O}(Nd \log T)$ updates. When $N = 1$, this matches other rarely-switching algorithms (Abbasi-Yadkori et al., 2011; Ruan et al., 2021; Midigeshi et al., 2025), and significantly improves upon RS-GLinCB (Algorithm 2, Sawarni et al. (2024)), which makes $\tilde{\mathcal{O}}(\kappa d^2 \log^2 T)$ updates.

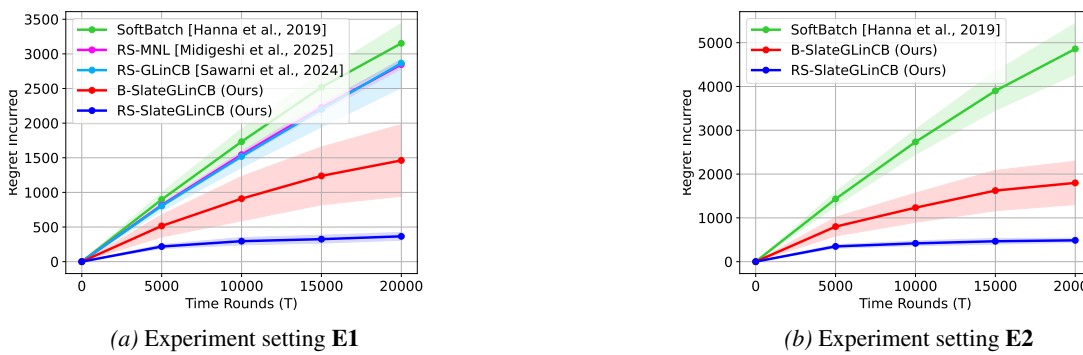

*(a)* Experiment setting **E1**  *(b)* Experiment setting **E2**

*Figure 1.* Comparison with limited adaptivity algorithms, `SoftBatch`, `RS-MNL` and `RS-GLinCB`

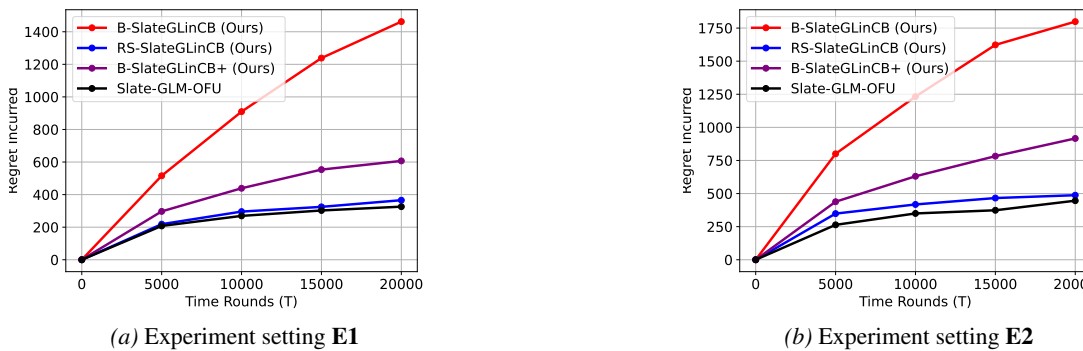

*(a)* Experiment setting **E1**  *(b)* Experiment setting **E2**

*Figure 2.* Comparison with the sequentially adaptive slate bandit algorithm `Slate-GLM-OFU`

## 5. Experiments

### 5.1. Synthetic Experiments

In this section, first, we empirically compare our algorithms, `B-SlateGLinCB` (Algorithm 1) and `RS-SlateGLinCB` (Algorithm 2) to other baseline algorithms that accommodate limited adaptivity[6]. These include `RS-GLinCB` (Algorithm 2, Sawarni et al. (2024)), `RS-MNL` (Algorithm 3, Midigeshi et al. (2025)), and a modified version of `SoftBatch` (Algorithm 5, Hanna et al. (2023)). We are not aware of any limited adaptivity algorithm specifically designed for the slate setting. Then, we also compare the regret of our algorithm to `Slate-GLM-OFU` (Algorithm 1, Goyal & Sinha (2025)), which is fully adaptive, i.e., parameters are updated at all rounds. To the best of our knowledge, this is the only contextual slate bandit algorithm designed for GLM rewards. In Appendix E, we detail the implementation details for all algorithms and showcase additional experiments.

**Experimental Design**: At each $t \in [T]$, for each slot $i \in [N]$, the set of items $\mathcal{X}_t^i \subset \mathbb{R}^d$, is chosen such that $|\mathcal{X}_t^i| = K = 5$ and $d = 5$. Each item in $\mathcal{X}_t^i$ is sampled from $[-1, 1]^5$ and is normalized to have $\ell_2$-norm $1/\sqrt{N}$ where

$N$ varies depending on the experiment setting. We randomly select $\boldsymbol{\theta}^\star$ from $[-1, 1]^{Nd}$ and normalize it to have $\ell_2$-norm $S$. For our algorithms, we set $\delta = 1/N^2$ which puts it in the range $[0.004, 0.04]$ for the values of $N$ used. For the baselines, we use the default values of $\delta$[7] provided in the corresponding implementation. We vary $S, N$ to create two different experiment settings capturing low and high regimes of $\kappa$ and the number of slates $K^N$: **E1**: $(S, N) = (2, 5)$, resulting in $K^N = 3125$ slates with dimension $Nd = 25$ and $\kappa \approx 7.38$. **E2**: $(S, N) = (5, 10)$, resulting in $K^N = 9765725$ slates with dimension $Nd = 50$ and $\kappa \approx 150$. We run our experiments for $T \in \{5000 * m : m \in [4]\}$ rounds and average over 25 different seeds for sampling rewards.

#### 5.1.1. RESULTS

**Comparison with limited adaptivity algorithms:** We see in Figures 1a and 1b that our algorithms `B-SlateGLinCB` and `RS-SlateGLinCB` achieve sublinear regret, and significantly outperform the limited adaptivity baselines in both the settings **E1** and **E2** respectively. These results also provide strong empirical support for our $\kappa$-free regret guarantees in Theorems 3.1 and 4.1. We also observe that the regret of `RS-SlateGLinCB` is better than `B-SlateGLinCB`

---

[6]We use a logistic reward model for all our experiments.

[7]which are of the same order as ours.

in both regimes, which can possibly be attributed to better constants as well as the $\sqrt{d}$ gap between the bounds provided in our theorems.

**Comparison with a fully adaptive algorithm**: In Figures 2a and 2b we compare the regret of our algorithms with that of the fully adaptive slate bandit algorithm `Slate-GLM-OFU`. Since `Slate-GLM-OFU` is not constrained by limited adaptivity, its parameters are updated at all rounds. Hence, we expect its regret to be better than that of our algorithms. However, we observe that for both settings **E1** and **E2**, the gap between `Slate-GLM-OFU` and `RS-SlateGLinCB` is quite small. In Figures 2a and 2b, we also include a slight modification of `B-SlateGLinCB`, which we refer to as `B-SlateGlinCB+`, and notice that its regret is extremely close to that of `Slate-GLM-OFU`. Similar to `B-SlateGLinCB`, `B-SlateGLinCB+` is also a batched algorithm with only $\mathcal{O}(\log \log T)$ parameter updates; however, it modifies *Step 18* of `B-SlateGLinCB` to perform fewer eliminations, i.e., instead of iterating over all previous batches $l \in [0, m-1]$, it only checks the elimination condition in *Step 20* for $l = m - 1$. While this clearly reduces the per-round time complexity, empirically, we observe that it also incurs much lower regret. It would be interesting to study the constraints under which one can prove strong regret bounds for such heuristics. In Appendix F, we provide additional insights and experiments for `B-SlateGLinCB+`.

## 5.2. Real World Experiments: Prompt Tuning

Next, we employ `B-SlateGLinCB+` to perform prompt tuning for language models through exemplar selection.

**Experimental Design**: All experiments are conducted using RoBERTa-large (Liu et al., 2021) as the base model and Nomic-Embed-Text-v1.5 (Nussbaum et al., 2024) on a binary sentiment classification task, namely, the SST-2 dataset (Socher et al., 2013). The instruction prompt is fixed *apriori*, and is designed in the form of a slate, where each of the $N$ *slots* correspond to an exemplar. At each time round, the algorithm is presented with a query and $N$ (different) pools consisting of $K$ candidate examples each. The algorithm is then required to select an exemplar (*item*) from each of the candidate pools to construct the prompt (*slate*). We choose $N = 6$ and $K = 9$.

At each time round, we construct the arm-sets as follows: the feature vector for each candidate example is a concatenation of three different components; a joint embedding between the query presented in the particular time round and the candidate example, the true label for the candidate example, and a pair of scores that measure the similarity between the query and the candidate example. We describe the experimental setup in complete detail in Appendix G.

**Baselines and Results**: We choose the following algorithms to be our baselines: (i) the base language model, without making use of any exemplars, (ii) the base language model, where the exemplars are chosen randomly (and hence, there is no learning) at each round, and (iii) the fully adaptive `Slate-GLM-OFU`, which updates its policy at each round. In Figure 3, we report the average cumulative accuracy of our algorithm `B-SlateGLinCB+` against that obtained by the other baselines over an augmented test set consisting of 4870 queries. We see that `B-SlateGLinCB+` achieves substantially higher accuracy than baselines (i) and (ii). Also, even though it performs only $\mathcal{O}(\log \log T)$ updates, it is incredibly competitive with `Slate-GLM-OFU`, showcasing its utility in practical scenarios.

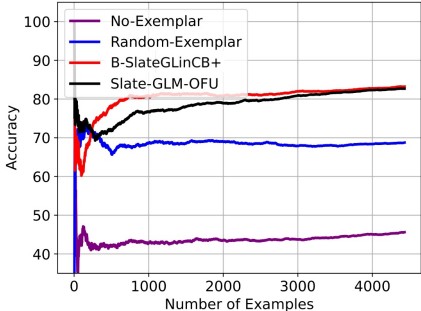

*Figure 3.* Prompt Tuning on SST-2

## 6. Conclusions

We present a batched algorithm `B-SlateGLinCB` and a rarely switching algorithm `RS-SlateGLinCB` for slate GLM bandits with bandit feedback. Under Assumption 2.1, we prove that `B-SlateGLinCB` and `RS-SlateGLinCB` incur $\mathcal{O}(Nd^{3/2}\sqrt{T})$ and $\mathcal{O}(Nd\sqrt{T})$ regret respectively, while having $poly(N)$ per round time complexity. Empirically, we show that our algorithms outperform all baseline limited adaptivity algorithms. At the same time, `RS-SlateGLinCB` is quite competitive with the fully adaptive `Slate-GLM-OFU` (Algorithm 1, (Goyal & Sinha, 2025)) algorithm. We also show that the regret of a modified algorithm `B-SlateGLinCB+` matches that of `Slate-GLM-OFU`. Finally, we implement prompt tuning using `B-SlateGLinCB+` on language models with exemplar selection and demonstrate strong performance in binary classification tasks. In fact, our performance matches that of the fully adaptive `Slate-GLM-OFU` algorithm. Developing batched algorithms with provably optimal regret guarantees and empirical performance matching `Slate-GLM-OFU` remains an important future direction.

## Impact Statement

This paper presents work whose goal is to advance the field of Machine Learning. There are many potential societal consequences of our work, none which we feel must be specifically highlighted here.

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

# A. Regret Analysis for `B-SlateGLinCB`

In this section, we state and prove the regret bound for `B-SlateGLinCB` (Algorithm 1).

## A.1. Notations

We first define the following scalar quantities: $\gamma = \mathcal{O}\left(SR\sqrt{Nd}\log(T\delta^{-1})\right)$ and $\lambda = \mathcal{O}\left(NdR^2\log(T\delta^{-1})\right)$.

We now define $\tilde{x}^i = x^i \otimes e_i$, where $\otimes$ represents the Kronecker product and $e_i$ is the $i^{th}$ standard basis vector. Note that this definition of $\tilde{x}^i$ is the same as the definition of *lift* of $x^i$ given in Goyal & Sinha (2025). Hence, all the properties shown in Goyal & Sinha (2025) continue to hold, and hence, for a slate $x = (x^1, \ldots, x^N)$, we have that

$$x = \sum_{i=1}^{N} \tilde{x}^i.$$

We define the slate-level warm-up design matrix $V_0$ as well as the slot-level design warm-up matrix $V_0^i$ for all $i \in [N]$ as follows:

1. $V_0 = \lambda I_{Nd} + \sum_{t \in \mathcal{T}_0} x_t x_t^\top.$

2. $V_0^i = \lambda I_d + \sum_{t \in \mathcal{T}_0} x_t^i {x_t^i}^\top.$

Now, recall the definition of the slate $b_t$ from Section 3: $b_t = (b_t^1, \ldots, b_t^N)$ where

$$b_t^i = \arg\max_{z \in \mathcal{X}_t^i} \|z\|_{(V_0^i)^{-1}}.$$

For batch $m$, we define the Hessian of the GLM-MLE loss as

$$H_m^\star = \lambda I_{Nd} + \sum_{t \in \mathcal{T}_m} \dot{\mu}(x_t^\top \theta^\star) x_t x_t^\top.$$

Since $\theta^\star$ is unknown, we estimate the Hessian using a scaled design matrix $H_m$, defined as

$$H_m = \lambda I_{Nd} + \sum_{t \in \mathcal{T}_m} \dot{\mu}(b_t^\top \widehat{\theta}_0)\beta_t^{-1} x_t x_t^\top,$$

where $\beta_t$ is a normalization factor obtained from the self-concordance relation, and is given by:

$$\beta_t = \exp\left(\min\left\{2S, 6\sqrt{\kappa}\gamma \sum_{i=1}^{N} \|b_t^i\|_{(V_0^i)^{-1}}\right\}\right).$$

Finally, for all $i \in [N]$, we define the slot-level scaled matrices $H_m^i$ for batch $m$ as

$$H_m^i = \lambda I_d + \sum_{t \in \mathcal{T}_m} \dot{\mu}(b_t^\top \widehat{\theta}_0)\beta_t^{-1} x_t^i {x_t^i}^\top.$$

For any slot $i \in [N]$, item $z \in \mathcal{X}_t^i$ and a prior batch $l \in [M]$, we define the scores $\text{UCB}^{i,l}(z)$ and $\text{LCB}^{i,l}(z)$ as

$$\text{UCB}^{i,l}(z) = \begin{cases} z^\top \widehat{\theta}_0^i + 2\sqrt{\kappa}\gamma \|z\|_{(V_0^i)^{-1}} & l = 0, \\ z^\top \widehat{\theta}_l^i + 2\gamma \|z\|_{(H_l^i)^{-1}} & l \neq 0. \end{cases}$$

$$\mathrm{LCB}^{i,l}(\boldsymbol{z}) = \begin{cases} \boldsymbol{z}^\top \widehat{\boldsymbol{\theta}}_0^i - 2\sqrt{\kappa}\gamma \|\boldsymbol{z}\|_{(\boldsymbol{V}_0^i)^{-1}} & l = 0, \\ \boldsymbol{z}^\top \widehat{\boldsymbol{\theta}}_l^i - 2\gamma \|\boldsymbol{z}\|_{(\boldsymbol{H}_l^i)^{-1}} & l \neq 0. \end{cases}$$

At round $t$, for all slots $i \in [N]$, the item-set $\mathcal{X}_t^i$ is sampled from distribution $\mathcal{D}^i$. In batch $m$, after pruning the arm-set $\mathcal{X}_t^i$ with respect to $\widehat{\boldsymbol{\theta}}_k^i$ for $0 \leq k \leq m - 1$, we obtain an item-set sampled from the distribution $\mathcal{D}_k^i$. Thus, pruning with respect to $\widehat{\boldsymbol{\theta}}_0^i, \widehat{\boldsymbol{\theta}}_1^i, \ldots, \widehat{\boldsymbol{\theta}}_{m-1}^i$ results in a sequence of set of items whose distributions are denoted by $\mathcal{D}_0^i, \mathcal{D}_1^i, \ldots, \mathcal{D}_{m-1}^i$.

Finally, define the following quantities:

$$T(\boldsymbol{H}) := \frac{(48L_\mu^2 + 8L_\mu N\rho)(N-1)^2}{3\rho^2} \log\left(\frac{2dNT}{\delta}\right) \quad \text{and} \quad T(\boldsymbol{V}) := \frac{(48 + 8N\rho)(N-1)^2}{3\rho^2} \log\left(\frac{2dNT}{\delta}\right).$$

Unless otherwise mentioned, without loss of generality, we assume that all constants such as $S, T, R, N, d, \kappa$ and $L_\mu$ are greater than 1.

### A.2. Regret Guarantee for `B-SlateGLinCB`

Now, we restate the regret guarantee for `B-SlateGLinCB`, given in Theorem 3.1), and provide a proof for the same.

**Theorem A.1.** *Let $R(T)$ denote the regret of* `B-SlateGLinCB` *(Algorithm 1). If*

$$\sqrt{\frac{2dN}{\delta}} \frac{(48 + 8N\rho)(N-1)^2}{3\rho^2} \geq \frac{e}{2}$$

*and*

$$T \geq T_0 := \frac{\delta}{2dN} \exp\left(-2W_{-1}\left(\frac{-3\rho^2\sqrt{\delta}}{2\sqrt{2dN}(48L_\mu^2 + 8L_\mu N\rho)(N-1)^2}\right)\right),$$

*where $W_{-1}$ represents the decreasing branch of the Lambert W function (see Lemma A.15), then,*

$$R(T) = \tilde{\mathcal{O}}\left(RSNd^{3/2}\sqrt{T \cdot \mathop{\mathbb{E}}_{\{\mathcal{X}^i \sim \mathcal{D}^i\}_{i=1}^N} \dot{\mu}(\boldsymbol{x}_\star^\top \boldsymbol{\theta}^\star)}\right),$$

*where $\boldsymbol{x}_\star = \arg\max_{\boldsymbol{x} \in \mathcal{X}} \boldsymbol{x}^\top \boldsymbol{\theta}^\star$ and $\mathcal{X} = \mathcal{X}^1 \times \ldots \times \mathcal{X}^N$.*

*Proof.* At round $t \in \mathcal{T}_m$, let $\boldsymbol{x}_{t,\star}$ be the optimal slate, i.e,

$$\boldsymbol{x}_{t,\star} = \arg\max_{\boldsymbol{x} \in \mathcal{X}_t} \boldsymbol{x}^\top \boldsymbol{\theta}^\star.$$

Then, the expected regret for Algorithm 1, $R(T)$ can be written as

$$R(T) \leq \mathop{\mathbb{E}}_{\{\mathcal{X}_t^i \sim \mathcal{D}^i\}_{i=1}^N} \left[\sum_{m \in [M]} \sum_{t \in \mathcal{T}_m} \left|\mu(\boldsymbol{x}_{t,\star}^\top \boldsymbol{\theta}^\star) - \mu(\boldsymbol{x}_t^\top \boldsymbol{\theta}^\star)\right|\right].$$

We choose our batch lengths as follows:

$$|\mathcal{T}_0| = \lfloor\sqrt{T}\rfloor \quad \text{and} \quad |\mathcal{T}_m| = \lfloor T^{1-2^{-m}}\rfloor, m \geq 1.$$

We now make a few observations regarding these batch lengths. First, we obtain a total of $M = \mathcal{O}(\log\log T)$ batches. We also obtain the following inequalities:

$$\frac{|\mathcal{T}_m|}{\sqrt{|\mathcal{T}_{m-1}|}} \leq \frac{T^{1-2^{-m}}}{\sqrt{\lfloor T^{1-2^{1-m}}\rfloor}} = \frac{\sqrt{T} \cdot T^{\frac{1-2^{1-m}}{2}}}{\sqrt{\lfloor T^{1-2^{1-m}}\rfloor}} \leq \sqrt{T}, \quad \frac{|\mathcal{T}_m|^2}{|\mathcal{T}_{m-1}|} \leq \frac{T^{2-2^{1-m}}}{\lfloor T^{1-2^{1-m}}\rfloor} = \frac{T \cdot T^{1-2^{1-m}}}{\lfloor T^{1-2^{1-m}}\rfloor} \leq T.$$

Now, to use Lemma A.10, we require that $|\mathcal{T}_0| \geq T(\boldsymbol{V})$ and $|\mathcal{T}_k| \geq T(\boldsymbol{H})$ for all $k \in [M]$. Using the definitions of $T(\boldsymbol{V})$ and $T(\boldsymbol{H})$ from Section A.1 and the chosen batch lengths, we get:

$$\sqrt{T} \geq \frac{(48 + 8N\rho)(N-1)^2}{3\rho^2} \log\left(\frac{2dNT}{\delta}\right) \quad \text{and} \quad \sqrt{T} \geq \frac{(48L_\mu^2 + 8L_\mu N\rho)(N-1)^2}{3\rho^2} \log\left(\frac{2dNT}{\delta}\right).$$

Now, since $L_\mu \geq 1$, the assumptions

$$\sqrt{\frac{2dN}{\delta}} \frac{(48 + 8N\rho)(N-1)^2}{3\rho^2} \geq \frac{e}{2}$$

and

$$T \geq 2 \geq \frac{\delta}{2dN} \exp\left(\frac{1}{2}\right)$$

ensures that we can use Lemma A.15 to satisfy the conditions for Lemma A.10 giving us

$$T \geq \frac{\delta}{2dN} \exp\left(\max\left\{-2W_{-1}\left(\frac{-3\rho^2\sqrt{\delta}}{2\sqrt{2dN}(48 + 8N\rho)(N-1)^2}\right), -2W_{-1}\left(\frac{-3\rho^2\sqrt{\delta}}{2\sqrt{2dN}(48L_\mu^2 + 8L_\mu N\rho)(N-1)^2}\right)\right\}\right)$$

$$= T_0 := \frac{\delta}{2dN} \exp\left(-2W_{-1}\left(\frac{-3\rho^2\sqrt{\delta}}{2\sqrt{2dN}(48L_\mu^2 + 8L_\mu N\rho)(N-1)^2}\right)\right)$$

where we use the fact that $\exp(-x)$ is decreasing, $W_{-1}(x)$ is decreasing on $(-e^{-1}, 0)$ (Lemma A.15) and $L_\mu \geq 1$.

Thus, assuming $T \geq T_0$, using Lemma A.10 for $m \geq 2$, as well as, a trivial regret bound of $R$ for each round $t \in \mathcal{T}_0 \cup \mathcal{T}_1$, we get that

$$R(T) \leq R(|\mathcal{T}_0| + |\mathcal{T}_1|) + \sum_{m \in [2,M]} \sum_{t \in \mathcal{T}_m} \mathbb{E}_{\{\mathcal{X}_t^i \sim \mathcal{D}_m^i\}_{i=1}^N} \left|\mu(\boldsymbol{x}_{t,\star}^\top \boldsymbol{\theta}^\star) - \mu(\boldsymbol{x}_t^\top \boldsymbol{\theta}^\star)\right|.$$

$$\leq R(|\mathcal{T}_0| + |\mathcal{T}_1|) + \sum_{m=2}^M \left[\frac{320e^{3S}\gamma^2 N^2 d^2 \sqrt{\kappa L_\mu}}{S} \frac{|\mathcal{T}_m|}{\sqrt{|\mathcal{T}_{m-1}||\mathcal{T}_0|}} + 8\gamma \sqrt{\mathbb{E}_{\{\mathcal{X}^i \sim \mathcal{D}^i\}_{i=1}^N} \dot\mu(\boldsymbol{x}_\star^\top \boldsymbol{\theta}^\star)} \sqrt{\frac{8d^2 N |\mathcal{T}_m|^2}{|\mathcal{T}_{m-1}|} + \frac{4L_\mu N |\mathcal{T}_m|}{\rho}}\right].$$

Substituting the values of $|\mathcal{T}_m|, \frac{|\mathcal{T}_m|}{\sqrt{|\mathcal{T}_{m-1}|}}, \frac{|\mathcal{T}_m|^2}{|\mathcal{T}_{m-1}|}$, and using the facts that $|\mathcal{T}_m| \leq T$ and $|\mathcal{T}_0| = \lfloor\sqrt{T}\rfloor \geq \sqrt{T}/2$, we get

$$R(T) \leq \left(8\gamma \sqrt{\mathbb{E}_{\{\mathcal{X}^i \sim \mathcal{D}^i\}_{i=1}^N} \dot\mu(\boldsymbol{x}_\star^\top \boldsymbol{\theta}^\star)} \sqrt{8d^2 N + 4L_\mu N\rho^{-1}} \log\log T + 2R\right) \sqrt{T} + \tilde{\mathcal{O}}\left(e^{3S}\sqrt{\kappa} T^{1/4}\right).$$

Substituting $\gamma = \mathcal{O}\left(SR\sqrt{Nd\log(T\delta^{-1})}\right)$ from Lemma A.2 gives us

$$R(T) = \tilde{\mathcal{O}}\left(RSNd^{3/2} \sqrt{T \cdot \mathbb{E}_{\{\mathcal{X}^i \sim \mathcal{D}^i\}_{i=1}^N} \dot\mu(\boldsymbol{x}_\star^\top \boldsymbol{\theta}^\star)}\right).$$

$\square$

## A.3. Supporting Lemmas for Theorem A.1

**Lemma A.2.** *(Lemma A.2, Sawarni et al. (2024)) Let $\{\boldsymbol{x}_1, \ldots, \boldsymbol{x}_t\} \subset \mathbb{R}^d$ be independent arm pulls and $\{r_1, \ldots, r_t\}$ be the corresponding rewards associated with them. Define the matrix $\boldsymbol{H}_t^\star$ as follows:*

$$\boldsymbol{H}_t^\star = \lambda \boldsymbol{I} + \sum_{s \in [t]} \dot\mu(\boldsymbol{x}_s^\top \boldsymbol{\theta}^\star) \boldsymbol{x}_s \boldsymbol{x}_s^\top.$$

*Also, let $\widehat{\boldsymbol{\theta}}_t$ be the maximum likelihood estimator of $\boldsymbol{\theta}^\star$. Then, for $\lambda = \mathcal{O}\left(dR^2 \log(T\delta^{-1})\right)$, with high probability,*

$$\|\boldsymbol{\theta}^\star - \widehat{\boldsymbol{\theta}}_t\|_{\boldsymbol{H}_t^\star} \leq \gamma := \mathcal{O}\left(SR\sqrt{d\log(T\delta^{-1})}\right).$$

**Lemma A.3.** *Let $\boldsymbol{x} = (\boldsymbol{x}^1, \ldots, \boldsymbol{x}^N)$ be some slate. Then, for $|\mathcal{T}_0| \geq T(\boldsymbol{V})$, we have that*

$$\boldsymbol{x}^\top(\boldsymbol{\theta}^\star - \widehat{\boldsymbol{\theta}}_0) \leq 2\sqrt{\kappa}\gamma \sum_{i=1}^N \|\boldsymbol{x}^i\|_{(\boldsymbol{V}_0^i)^{-1}}.$$

*Proof.* For the sake of this proof, define

$$\boldsymbol{H}_0^\star = \lambda \boldsymbol{I} + \sum_{t \in \mathcal{T}_0} \dot{\mu}(\boldsymbol{x}_t^\top \boldsymbol{\theta}^\star) \boldsymbol{x}_t \boldsymbol{x}_t^\top.$$

Then, using the definition of $\kappa$, we have that $\dot{\mu}(\boldsymbol{x}_t^\top \boldsymbol{\theta}^\star) \geq \kappa^{-1}$. Thus,

$$\boldsymbol{H}_0^\star \succeq \lambda \boldsymbol{I} + \kappa^{-1} \sum_{t \in \mathcal{T}_0} \boldsymbol{x}_t \boldsymbol{x}_t^\top$$

$$= \kappa^{-1} \left( \kappa \lambda \boldsymbol{I} + \sum_{t \in \mathcal{T}_0} \boldsymbol{x}_t \boldsymbol{x}_t^\top \right)$$

$$\succeq \kappa^{-1} \boldsymbol{V}_0$$

where the last inequality uses the fact that $\lambda \geq 1$ and $\kappa \geq 1$. Hence, we can write that

$$\boldsymbol{x}^\top(\boldsymbol{\theta}^\star - \widehat{\boldsymbol{\theta}}_0) \leq \|\boldsymbol{\theta}^\star - \widehat{\boldsymbol{\theta}}_0\|_{\boldsymbol{H}_0^\star} \|\boldsymbol{x}\|_{\boldsymbol{H}_0^{\star-1}}$$

$$\leq \sqrt{\kappa}\gamma \|\boldsymbol{x}\|_{\boldsymbol{V}_0^{-1}}$$

$$\leq 2\sqrt{\kappa}\gamma \sum_{i=1}^N \|\boldsymbol{x}^i\|_{(\boldsymbol{V}_0^i)^{-1}}.$$

where the second inequality follows from Lemma A.2 and the final inequality follows from Lemma A.14. $\square$

**Lemma A.4.** *Let $t \in \mathcal{T}_m$. Let $\boldsymbol{x}, \boldsymbol{y} \in \mathcal{X}_t$ be two slates which do not get eliminated. Then*

$$\left| (\boldsymbol{x} - \boldsymbol{y})^\top \widehat{\boldsymbol{\theta}}_0 \right| \leq 4\sqrt{\kappa}\gamma \sum_{i=1}^N \|\boldsymbol{b}^i\|_{(\boldsymbol{V}_0^i)^{-1}}.$$

*Proof.* Using the triangle inequality, we can write

$$\left| (\boldsymbol{x} - \boldsymbol{y})^\top \widehat{\boldsymbol{\theta}}_0 \right| \leq \sum_{i=1}^N \left| (\boldsymbol{x}^i - \boldsymbol{y}^i)^\top \widehat{\boldsymbol{\theta}}_0^i \right|.$$

Now, since both $\boldsymbol{x}$ and $\boldsymbol{y}$ survive the elimination, their respective components $\boldsymbol{x}^i$ and $\boldsymbol{y}^i$ for all $i \in [N]$ also do not get eliminated. Thus, for a fixed $i$, we have

$$\text{UCB}^{i,0}(\boldsymbol{x}^i) \geq \max_{\boldsymbol{z} \in \mathcal{X}_t^i} \text{LCB}^{i,0}(\boldsymbol{z}^i) \geq \text{LCB}^{i,0}(\boldsymbol{y}^i).$$

Using the definitions of $\text{UCB}^{i,0}(\boldsymbol{z})$ and $\text{LCB}^{i,0}(\boldsymbol{z})$ (Section A.1), we get

$$(\boldsymbol{y}^i - \boldsymbol{x}^i)^\top \widehat{\boldsymbol{\theta}}_0^i \leq 2\sqrt{\kappa}\gamma \|\boldsymbol{x}^i\|_{(\boldsymbol{V}_0^i)^{-1}} + 2\sqrt{\kappa}\gamma \|\boldsymbol{y}^i\|_{(\boldsymbol{V}_0^i)^{-1}}.$$

A symmetric argument gives us

$$(\boldsymbol{x}^i - \boldsymbol{y}^i)^\top \widehat{\boldsymbol{\theta}}_0^i \leq 2\sqrt{\kappa}\gamma \|\boldsymbol{x}^i\|_{(\boldsymbol{V}_0^i)^{-1}} + 2\sqrt{\kappa}\gamma \|\boldsymbol{y}^i\|_{(\boldsymbol{V}_0^i)^{-1}}.$$

Thus, combining both the inequalities, we get

$$\left|(\boldsymbol{x}^i - \boldsymbol{y}^i)^\top \widehat{\boldsymbol{\theta}}_0^i\right| \leq 2\sqrt{\kappa}\gamma\|\boldsymbol{x}^i\|_{(\boldsymbol{V}_0^i)^{-1}} + 2\sqrt{\kappa}\gamma\|\boldsymbol{y}^i\|_{(\boldsymbol{V}_0^i)^{-1}}$$
$$\leq 4\sqrt{\kappa}\gamma \max_{\boldsymbol{z} \in \mathcal{X}_t^i}\|\boldsymbol{z}\|_{(\boldsymbol{V}_0^i)^{-1}}$$

Substituting this back and using the definition of $\boldsymbol{b}_t^i$ gives us

$$\left|(\boldsymbol{x} - \boldsymbol{y})^\top \widehat{\boldsymbol{\theta}}_0\right| \leq 4\sqrt{\kappa}\gamma \sum_{i=1}^{N}\|\boldsymbol{b}^i\|_{(\boldsymbol{V}_0^i)^{-1}}.$$

$\square$

**Lemma A.5.** *Let $t \in \mathcal{T}_m$. Also, let $\boldsymbol{H}_m^\star$ and $\boldsymbol{H}_m$ be as defined in Section A.1. Then, we have that*

$$\boldsymbol{H}_m^\star \succeq \boldsymbol{H}_m.$$

*Also, for any slate $\boldsymbol{x} \in \mathcal{X}_t$ that survives the elimination, if $|\mathcal{T}_0| \geq T(\boldsymbol{V})$ and $|\mathcal{T}_m| \geq T(\boldsymbol{H})$, then, we have that*

$$\boldsymbol{x}^\top(\boldsymbol{\theta}^\star - \widehat{\boldsymbol{\theta}}_m) \leq 2\gamma \sum_{i=1}^{N}\|\boldsymbol{x}^i\|_{(\boldsymbol{H}_m^i)^{-1}}.$$

*Proof.* Recall the definition of $\boldsymbol{b}_t$ from Section A.1. From the self-concordance property of GLMs, we have

$$\dot{\mu}(\boldsymbol{b}_t^\top\widehat{\boldsymbol{\theta}}_0) \leq \dot{\mu}(\boldsymbol{x}^\top\boldsymbol{\theta}^\star)\exp(|\boldsymbol{b}_t^\top\widehat{\boldsymbol{\theta}}_0 - \boldsymbol{x}^\top\boldsymbol{\theta}^\star|).$$

We can bound $|\boldsymbol{b}_t^\top\widehat{\boldsymbol{\theta}}_0 - \boldsymbol{x}^\top\boldsymbol{\theta}^\star|$ as follows:

$$|\boldsymbol{b}_t^\top\widehat{\boldsymbol{\theta}}_0 - \boldsymbol{x}^\top\boldsymbol{\theta}^\star| \leq |\boldsymbol{b}_t^\top\widehat{\boldsymbol{\theta}}_0| + |\boldsymbol{x}^\top\boldsymbol{\theta}^\star| \leq 2S.$$

Also, using Lemma A.3 and Lemma A.4, we have

$$|\boldsymbol{b}^\top\widehat{\boldsymbol{\theta}}_0 - \boldsymbol{x}^\top\boldsymbol{\theta}^\star| \leq |\boldsymbol{x}^\top\boldsymbol{\theta}^\star - \boldsymbol{x}^\top\widehat{\boldsymbol{\theta}}_0| + |\boldsymbol{b}^\top\widehat{\boldsymbol{\theta}}_0 - \boldsymbol{x}^\top\widehat{\boldsymbol{\theta}}_0|$$
$$\leq 6\sqrt{\kappa}\gamma \sum_{i=1}^{N}\|\boldsymbol{b}^i\|_{(\boldsymbol{V}_0^i)^{-1}}.$$

Thus, combining both the inequalities results in

$$\dot{\mu}(\boldsymbol{b}^\top\widehat{\boldsymbol{\theta}}_0) \leq \dot{\mu}(\boldsymbol{x}^\top\boldsymbol{\theta}^\star)\exp\left(\min\left\{2S, 6\sqrt{\kappa}\gamma \sum_{i=1}^{N}\|\boldsymbol{b}^i\|_{(\boldsymbol{V}_0^i)^{-1}}\right\}\right).$$

Noting that the multiplicative factor on the right side is precisely $\beta_t$ (refer Section A.1), we get that:

$$\boldsymbol{H}_m^\star = \lambda\boldsymbol{I}_{Nd} + \sum_{t \in \mathcal{T}_m} \dot{\mu}(\boldsymbol{x}_t^\top\boldsymbol{\theta}^\star)\boldsymbol{x}_t\boldsymbol{x}_t^\top$$
$$\succeq \lambda\boldsymbol{I}_{Nd} + \sum_{t \in \mathcal{T}_m} \dot{\mu}(\boldsymbol{b}^\top\widehat{\boldsymbol{\theta}}_0)\beta_t^{-1}\boldsymbol{x}_t\boldsymbol{x}_t^\top = \boldsymbol{H}_m.$$

This completes the proof for the first part. For the second part, we have

$$\boldsymbol{x}^\top(\boldsymbol{\theta}^\star - \widehat{\boldsymbol{\theta}}_m) \leq \|\boldsymbol{\theta}^\star - \widehat{\boldsymbol{\theta}}_m\|_{\boldsymbol{H}_m^\star}\|\boldsymbol{x}\|_{\boldsymbol{H}_m^{\star-1}}$$
$$\leq 2\gamma \sum_{i=1}^{N}\left\|\boldsymbol{x}^i\right\|_{(\boldsymbol{H}_m^i)^{-1}}.$$

where the final inequality follows from Lemma A.2 and Lemma A.13.

$\square$

**Lemma A.6.** *Let $t \in \mathcal{T}_m$. Let $\boldsymbol{x}, \boldsymbol{y} \in \mathcal{X}_t$ be two slates which do not get eliminated. Then, for all $1 \leq k \leq m-1$, we have*

$$\left| (\boldsymbol{x} - \boldsymbol{y})^\top \widehat{\boldsymbol{\theta}}_k \right| \leq 4\gamma \sum_{i=1}^{N} \max_{\boldsymbol{z} \in \mathcal{X}_t^i} \|\boldsymbol{z}\|_{(\boldsymbol{H}_k^i)^{-1}}.$$

*Proof.* Using the triangle inequality, we can write

$$\left| (\boldsymbol{x} - \boldsymbol{y})^\top \widehat{\boldsymbol{\theta}}_k \right| \leq \sum_{i=1}^{N} \left| (\boldsymbol{x}^i - \boldsymbol{y}^i)^\top \widehat{\boldsymbol{\theta}}_k^i \right|.$$

Now, since both $\boldsymbol{x}$ and $\boldsymbol{y}$ survive the elimination, for an arbitrary but fixed slot $i \in [N]$, their respective components $\boldsymbol{x}^i$ and $\boldsymbol{y}^i$ also do not get eliminated. Thus, we have

$$\text{UCB}^{i,k}(\boldsymbol{x}^i) \geq \max_{\boldsymbol{z} \in \mathcal{X}_t^i} \text{LCB}^{i,k}(\boldsymbol{z}^i) \geq \text{LCB}^{i,k}(\boldsymbol{y}^i).$$

Using the definitions of $\text{UCB}^{i,k}(\boldsymbol{z})$ and $\text{LCB}^{i,k}(\boldsymbol{z})$ (Section A.1), we get

$$(\boldsymbol{y}^i - \boldsymbol{x}^i)^\top \widehat{\boldsymbol{\theta}}_k^i \leq 2\gamma \|\boldsymbol{x}^i\|_{(\boldsymbol{H}_k^i)^{-1}} + 2\gamma \|\boldsymbol{y}^i\|_{(\boldsymbol{H}_k^i)^{-1}}.$$

A symmetric argument gives us

$$(\boldsymbol{x}^i - \boldsymbol{y}^i)^\top \widehat{\boldsymbol{\theta}}_k^i \leq 2\gamma \|\boldsymbol{x}^i\|_{(\boldsymbol{H}_k^i)^{-1}} + 2\gamma \|\boldsymbol{y}^i\|_{(\boldsymbol{H}_k^i)^{-1}}.$$

Thus, combining both the inequalities gives us

$$\begin{aligned}
\left| (\boldsymbol{x}^i - \boldsymbol{y}^i)^\top \widehat{\boldsymbol{\theta}}_k^i \right| &\leq 2\gamma \|\boldsymbol{x}^i\|_{(\boldsymbol{H}_k^i)^{-1}} + 2\gamma \|\boldsymbol{y}^i\|_{(\boldsymbol{H}_k^i)^{-1}} \\
&\leq 4\gamma \max_{\boldsymbol{z} \in \mathcal{X}_t^i} \|\boldsymbol{z}\|_{(\boldsymbol{H}_k^i)^{-1}}.
\end{aligned}$$

Summing over all slots $i \in [N]$ finishes the proof. $\qquad\square$

**Lemma A.7.** *For $t \in \mathcal{T}_m$, where $m > 0$, define the optimal slate $\boldsymbol{x}_{t,\star} = (\boldsymbol{x}_{t,\star}^1, \ldots, \boldsymbol{x}_{t,\star}^N)$ as follows:*

$$\boldsymbol{x}_{t,\star} = \arg\max_{\boldsymbol{x} \in \mathcal{X}_t} \boldsymbol{x}^\top \boldsymbol{\theta}^\star.$$

*If $|\mathcal{T}_0| \geq T(\boldsymbol{V})$ and $|\mathcal{T}_k| \geq T(\boldsymbol{H})$ for all $1 \leq k \leq m-1$, then, the optimal slate never gets eliminated.*

*Proof.* First, note that since $\boldsymbol{x}_{t,\star} = \arg\max_{\boldsymbol{x} \in \mathcal{X}_t} \boldsymbol{x}^\top \boldsymbol{\theta}^\star$, it is easy to see that $\boldsymbol{x}_{t,\star}^i = \arg\max_{\boldsymbol{z} \in \mathcal{X}_t^i} \boldsymbol{z}^\top \boldsymbol{\theta}^{\star i}$, or, in other words $\tilde{\boldsymbol{x}}_{t,\star}^i = \arg\max_{\boldsymbol{z} \in \mathcal{X}_t^i} \tilde{\boldsymbol{z}}^\top \boldsymbol{\theta}^\star$.

Fix $i \in [N]$. Then, for some arbitrary $\boldsymbol{z} \in \mathcal{X}_t^i$, we have that

$$\begin{aligned}
0 &\leq \left( \tilde{\boldsymbol{x}}_{t,\star}^i - \tilde{\boldsymbol{z}} \right)^\top \boldsymbol{\theta}^\star \\
&= \left( \tilde{\boldsymbol{x}}_{t,\star}^i - \tilde{\boldsymbol{z}} \right)^\top \left( \boldsymbol{\theta}^\star - \widehat{\boldsymbol{\theta}}_0 \right) + \left( \tilde{\boldsymbol{x}}_{t,\star}^i - \tilde{\boldsymbol{z}} \right)^\top \widehat{\boldsymbol{\theta}}_0 \\
&\leq 2\sqrt{\kappa}\gamma \|\boldsymbol{x}_{t,\star}^i\|_{(\boldsymbol{V}_0^i)^{-1}} + 2\sqrt{\kappa}\gamma \|\boldsymbol{z}\|_{(\boldsymbol{V}_0^i)^{-1}} + \left( \tilde{\boldsymbol{x}}_{t,\star}^i - \tilde{\boldsymbol{z}} \right)^\top \widehat{\boldsymbol{\theta}}_0 \\
&= \text{UCB}^{i,0}(\boldsymbol{x}_{t,\star}^i) - \text{LCB}^{i,0}(\boldsymbol{z}),
\end{aligned}$$

where the second inequality follows from Lemma A.3. Since this is true $\forall \boldsymbol{z} \in \mathcal{X}_t^i$, we get

$$\text{UCB}^{i,0}(\boldsymbol{x}_{t,\star}^i) \geq \max_{\boldsymbol{z} \in \mathcal{X}_t^i} \text{LCB}^{i,0}(\boldsymbol{z}),$$

or in other words,

$$\mathrm{UCB}^{i,0}(\boldsymbol{x}_{t,\star}^i) - \max_{\boldsymbol{z}\in\mathcal{X}_t^i}\mathrm{LCB}^{i,0}(\boldsymbol{z}) \geq 0.$$

Since this holds for a fixed but arbitrary $i \in [N]$, the above inequality holds for all $i \in [N]$.

Similarly, for all $k \in [1, m-1]$ and $i \in [N]$, we can show that

$$\mathrm{UCB}^{i,k}(\boldsymbol{x}_{t,\star}^i) - \max_{\boldsymbol{z}\in\mathcal{X}_t^i}\mathrm{LCB}^{i,k}(\boldsymbol{z}) \geq 0.$$

Thus, the components of the optimal slate, and hence, the optimal slate never gets eliminated. □

**Lemma A.8.** *For some slot $i \in [N]$ and batch $m \in [0, M]$, for all $j \leq m-1$, let $\mathcal{D}_j^i$ be defined as in Section A.1. Then, for any matrix $\boldsymbol{A} \succeq 0$, we have that*

$$\mathop{\mathbb{E}}_{\mathcal{X}^i\sim\mathcal{D}_m^i}\max_{\boldsymbol{x}^i\in\mathcal{X}^i}\|\boldsymbol{x}\|_{\boldsymbol{A}} \leq \mathop{\mathbb{E}}_{\mathcal{X}^i\sim\mathcal{D}_j^i}\max_{\boldsymbol{x}^i\in\mathcal{X}^i}\|\boldsymbol{x}\|_{\boldsymbol{A}} \ \forall j \in [m-1].$$

*Proof.* The proof only relies on the manner in which the sequence of distributions $\{\mathcal{D}_j^i\}_{j\in[M]}$ is constructed. In particular, the pruning step ensures that the set of items that survive the pruning with respect to $\widehat{\boldsymbol{\theta}}_m$ is always a smaller set than the set of items that survive the pruning with respect to $\widehat{\boldsymbol{\theta}}_j$ for $j \leq m-1$. Thus, for some slot $i \in [N]$, the pruning step gives rise to the following chain of subsets:

$$\mathcal{D}_m^i \subseteq \mathcal{D}_{m-1}^i \subseteq \ldots \subseteq \mathcal{D}_1^i \subseteq \mathcal{D}_0^i \subseteq \mathcal{D}^i.$$

Note that this result is a generalization of Claim A.11 from Sawarni et al. (2024) to the slate setting. Hence, the claim follows. □

**Lemma A.9.** *At round $t \in \mathcal{T}_m$, let $\boldsymbol{x}_{t,\star}$ be the optimal slate, i.e,*

$$\boldsymbol{x}_{t,\star} = \arg\max_{\boldsymbol{x}\in\mathcal{X}_t}\boldsymbol{x}^\top\boldsymbol{\theta}^\star.$$

*Define $\breve{\boldsymbol{x}} := \sqrt{\dot{\mu}(\boldsymbol{b}_t^\top\widehat{\boldsymbol{\theta}}_0)\beta_t^{-1}}\boldsymbol{x}$. If $|\mathcal{T}_0| \geq T(\boldsymbol{V})$ and $|\mathcal{T}_k| \geq T(\boldsymbol{H})$ for all $1 \leq k \leq m-1$, then, we have*

$$\left|\mu(\boldsymbol{x}_{t,\star}^\top\boldsymbol{\theta}^\star) - \mu(\boldsymbol{x}_t^\top\boldsymbol{\theta}^\star)\right| \leq 8\gamma\sqrt{\dot{\mu}(\boldsymbol{x}_{t,\star}^\top\boldsymbol{\theta}^\star)}\left(\sum_{i=1}^N\max_{\boldsymbol{z}\in\mathcal{X}_t^i}\|\breve{\boldsymbol{z}}\|_{(\boldsymbol{H}_{m-1}^i)^{-1}}\right)\left(\frac{5e^{3S}\sqrt{\kappa}\gamma}{S}\sum_{i=1}^N\|\boldsymbol{b}_t^i\|_{(\boldsymbol{V}_0^i)^{-1}} + 1\right).$$

*Proof.* Using a first-order Taylor series expansion, for some $z_t \in [\boldsymbol{x}_t^\top\boldsymbol{\theta}^\star, \boldsymbol{x}_{t,\star}^\top\boldsymbol{\theta}^\star]$, we have that

$$\left|\mu(\boldsymbol{x}_{t,\star}^\top\boldsymbol{\theta}^\star) - \mu(\boldsymbol{x}_t^\top\boldsymbol{\theta}^\star)\right| \leq \dot{\mu}(z_t)\left|\boldsymbol{x}_{t,\star}^\top\boldsymbol{\theta}^\star - \boldsymbol{x}_t^\top\boldsymbol{\theta}^\star\right|.$$

Further, using Lemma A.5 and Lemma A.6, we have

$$\dot{\mu}(z_t)\left|\boldsymbol{x}_{t,\star}^\top\boldsymbol{\theta}^\star - \boldsymbol{x}_t^\top\boldsymbol{\theta}^\star\right| \leq \dot{\mu}(z_t)|\boldsymbol{x}_{t,\star}^\top\boldsymbol{\theta}^\star - \boldsymbol{x}_{t,\star}^\top\widehat{\boldsymbol{\theta}}_{m-1}| + \dot{\mu}(z_t)|\boldsymbol{x}_t^\top\boldsymbol{\theta}^\star - \boldsymbol{x}_t^\top\widehat{\boldsymbol{\theta}}_{m-1}| + \dot{\mu}(z_t)|\boldsymbol{x}_{t,\star}^\top\widehat{\boldsymbol{\theta}}_{m-1} - \boldsymbol{x}_t^\top\widehat{\boldsymbol{\theta}}_{m-1}|$$

$$\leq 8\gamma\sum_{i=1}^N\dot{\mu}(z_t)\max_{\boldsymbol{u}\in\mathcal{X}_t^i}\|\boldsymbol{u}\|_{(\boldsymbol{H}_{m-1}^i)^{-1}}.$$

Define $\breve{\boldsymbol{x}} := \sqrt{\dot{\mu}(\boldsymbol{b}_t^\top\widehat{\boldsymbol{\theta}}_0)\beta_t^{-1}}\boldsymbol{x}$, then, we have

$$\left|\mu(\boldsymbol{x}_{t,\star}^\top\boldsymbol{\theta}^\star) - \mu(\boldsymbol{x}_t^\top\boldsymbol{\theta}^\star)\right| \leq 8\gamma\sum_{i=1}^N\sqrt{\frac{\dot{\mu}(z_t)^2\beta_t}{\dot{\mu}(\boldsymbol{b}_t^\top\widehat{\boldsymbol{\theta}}_0)}}\max_{\boldsymbol{u}\in\mathcal{X}_t^i}\|\breve{\boldsymbol{u}}\|_{(\boldsymbol{H}_{m-1}^i)^{-1}}.$$

Using the self-concordance property of GLMs, we have that

$$\frac{\dot{\mu}(z_t)}{\dot{\mu}(\boldsymbol{b}_t^\top \widehat{\boldsymbol{\theta}}_0)} \leq \exp\left(R\left|z_t - \boldsymbol{b}_t^\top \widehat{\boldsymbol{\theta}}_0\right|\right).$$

Since $z_t \in [\boldsymbol{x}_t^\top \boldsymbol{\theta}^\star, \boldsymbol{x}_{t,\star}^\top \boldsymbol{\theta}^\star]$, a trivial bound using $\|\boldsymbol{x}\| \leq 1$ and $\|\boldsymbol{\theta}^\star\| \leq S$ gives us

$$|z_t - \boldsymbol{b}_t^\top \widehat{\boldsymbol{\theta}}_0| \leq 2S.$$

Also, since the optimal slate $\boldsymbol{x}_{t,\star}$ never gets eliminated (Lemma A.7), using Lemma A.3 and Lemma A.4 gives us

$$|z_t - \boldsymbol{b}_t^\top \widehat{\boldsymbol{\theta}}_0| \leq |\boldsymbol{x}_t^\top \boldsymbol{\theta}^\star - \boldsymbol{x}_t^\top \widehat{\boldsymbol{\theta}}_0| + |\boldsymbol{x}_t^\top \widehat{\boldsymbol{\theta}}_0 - \boldsymbol{b}_t^\top \widehat{\boldsymbol{\theta}}_0| + |\boldsymbol{x}_t^\top \boldsymbol{\theta}^\star - z_t|$$

$$\leq 6\sqrt{\kappa}\gamma \sum_{i=1}^N \|\boldsymbol{b}_t^i\|_{(\boldsymbol{V}_0^i)^{-1}} + |\boldsymbol{x}_t^\top \boldsymbol{\theta}^\star - \boldsymbol{x}_{t,\star}^\top \boldsymbol{\theta}^\star|$$

and again, using Lemma A.3, Lemma A.4, and Lemma A.7, we have

$$\left|\boldsymbol{x}_{t,\star}^\top \boldsymbol{\theta}^\star - \boldsymbol{x}_t^\top \boldsymbol{\theta}^\star\right| \leq |\boldsymbol{x}_{t,\star}^\top \boldsymbol{\theta}^\star - \boldsymbol{x}_{t,\star}^\top \widehat{\boldsymbol{\theta}}_0| + |\boldsymbol{x}_t^\top \boldsymbol{\theta}^\star - \boldsymbol{x}_t^\top \widehat{\boldsymbol{\theta}}_0| + |\boldsymbol{x}_{t,\star}^\top \widehat{\boldsymbol{\theta}}_0 - \boldsymbol{x}_t^\top \widehat{\boldsymbol{\theta}}_0|$$

$$\leq 8\sqrt{\kappa}\gamma \sum_{i=1}^N \|\boldsymbol{b}_t^i\|_{(\boldsymbol{V}_0^i)^{-1}}.$$

Thus, we get

$$\frac{\dot{\mu}(z_t)}{\dot{\mu}(\boldsymbol{b}_t^\top \widehat{\boldsymbol{\theta}}_0)} \leq \exp\left(R\min\left\{2S, 14\sqrt{\kappa}\gamma \sum_{i=1}^N \|\boldsymbol{b}_t^i\|_{(\boldsymbol{V}_0^i)^{-1}}\right\}\right).$$

Similarly, we also have

$$\dot{\mu}(z_t) \leq \dot{\mu}(\boldsymbol{x}_{t,\star}^\top \boldsymbol{\theta}^\star)\exp\left(R|z_t - \boldsymbol{x}_{t,\star}^\top \boldsymbol{\theta}^\star|\right),$$

where the argument of the exponent can be bounded as

$$|z_t - \boldsymbol{x}_{t,\star}^\top \boldsymbol{\theta}^\star| \leq |\boldsymbol{x}_t^\top \boldsymbol{\theta}^\star - \boldsymbol{x}_{t,\star}^\top \boldsymbol{\theta}^\star| \leq 8\sqrt{\kappa}\gamma \sum_{i=1}^N \|\boldsymbol{b}_t^i\|_{(\boldsymbol{V}_0^i)^{-1}},$$

and hence,

$$\dot{\mu}(z_t) \leq \dot{\mu}(\boldsymbol{x}_{t,\star}^\top \boldsymbol{\theta}^\star)\exp\left(R\min\left\{2S, 8\sqrt{\kappa}\gamma \sum_{i=1}^N \|\boldsymbol{b}_t^i\|_{(\boldsymbol{V}_0^i)^{-1}}\right\}\right).$$

Using the definition of $\beta_t$ (Section A.1), we get

$$\sqrt{\frac{\dot{\mu}(z_t)^2 \beta_t}{\dot{\mu}(\boldsymbol{b}_t^\top \widehat{\boldsymbol{\theta}}_0)}} \leq \exp\left(R\min\left\{3S, 14\sqrt{\kappa}\gamma \sum_{i=1}^N \|\boldsymbol{b}_t^i\|_{(\boldsymbol{V}_0^i)^{-1}}\right\}\right)\sqrt{\dot{\mu}(\boldsymbol{x}_{t,\star}^\top \boldsymbol{\theta}^\star)}.$$

Finally, using Claim A.8 from Sawarni et al. (2024), we get

$$\sqrt{\frac{\dot{\mu}(z_t)^2 \beta_t}{\dot{\mu}(\boldsymbol{b}_t^\top \widehat{\boldsymbol{\theta}}_0)}} \leq \left(\frac{5e^{3S}\sqrt{\kappa}\gamma}{S} \sum_{i=1}^N \|\boldsymbol{b}_t^i\|_{(\boldsymbol{V}_0^i)^{-1}} + 1\right)\sqrt{\dot{\mu}(\boldsymbol{x}_{t,\star}^\top \boldsymbol{\theta}^\star)}.$$

Substituting this back, we get

$$\left|\mu(\boldsymbol{x}_{t,\star}^\top \boldsymbol{\theta}^\star) - \mu(\boldsymbol{x}_t^\top \boldsymbol{\theta}^\star)\right| \leq 8\gamma\sqrt{\dot{\mu}(\boldsymbol{x}_{t,\star}^\top \boldsymbol{\theta}^\star)}\left(\sum_{i=1}^N \max_{\boldsymbol{z}\in\mathcal{X}_t^i}\|\check{\boldsymbol{z}}\|_{(\boldsymbol{H}_{m-1}^i)^{-1}}\right)\left(\frac{5e^{3S}\sqrt{\kappa}\gamma}{S} \sum_{i=1}^N \|\boldsymbol{b}_t^i\|_{(\boldsymbol{V}_0^i)^{-1}} + 1\right).$$

$\square$

**Lemma A.10.** *At round $t \in \mathcal{T}_m$, let $\boldsymbol{x}_{t,\star}$ be the optimal slate,*

$$\boldsymbol{x}_{t,\star} = \arg\max_{\boldsymbol{x} \in \mathcal{X}_t} \boldsymbol{x}^\top \boldsymbol{\theta}^\star.$$

*If $|\mathcal{T}_0| \geq T(\boldsymbol{V})$ and $|\mathcal{T}_k| \geq T(\boldsymbol{H})$ for all $1 \leq k \leq m-1$, then, we have*

$$\sum_{t \in \mathcal{T}_m} \mathop{\mathbb{E}}_{\{\mathcal{X}_t^i \sim \mathcal{D}_m^i\}_{i=1}^N} \left| \mu(\boldsymbol{x}_{t,\star}^\top \boldsymbol{\theta}^\star) - \mu(\boldsymbol{x}_t^\top \boldsymbol{\theta}^\star) \right|$$

$$\leq \frac{320 e^{3S} \gamma^2 N^2 d^2 \sqrt{\kappa L_\mu}}{S} \frac{|\mathcal{T}_m|}{\sqrt{|\mathcal{T}_{m-1}||\mathcal{T}_0|}} + 8\gamma \sqrt{\mathop{\mathbb{E}}_{\{\mathcal{X}^i \sim \mathcal{D}^i\}_{i=1}^N} \dot{\mu}(\boldsymbol{x}_\star^\top \boldsymbol{\theta}^\star)} \sqrt{\frac{8 d^2 N |\mathcal{T}_m|^2}{|\mathcal{T}_{m-1}|} + \frac{4 L_\mu N |\mathcal{T}_m|}{\rho}}.$$

*where $\boldsymbol{x}_\star = \arg\max_{\boldsymbol{x} \in \mathcal{X}} \boldsymbol{x}^\top \boldsymbol{\theta}^\star$ and $\mathcal{X} = \mathcal{X}^1 \times \ldots \times \mathcal{X}^N$.*

*Proof.* Define $\breve{\boldsymbol{x}} := \sqrt{\dot{\mu}(\boldsymbol{b}_t^\top \widehat{\boldsymbol{\theta}}_0) \beta_t^{-1}} \boldsymbol{x}$ and $\mathop{\mathbb{E}}_{\{\mathcal{X}_t^i \sim \mathcal{D}_m^i\}_{i=1}^N}[.]$ as $\mathop{\mathbb{E}}_{t,m}[.]$. Then, using Lemma A.9, we wish to bound the following two terms (excluding constants):

$$1. \ \mathop{\mathbb{E}}_{t,m} \sqrt{\dot{\mu}(\boldsymbol{x}_{t,\star}^\top \boldsymbol{\theta}^\star)} \left( \sum_{i=1}^N \max_{\boldsymbol{z} \in \mathcal{X}_t^i} \|\breve{\boldsymbol{z}}\|_{(\boldsymbol{H}_{m-1}^i)^{-1}} \right) \left( \sum_{i=1}^N \|\boldsymbol{b}_t^i\|_{(\boldsymbol{V}_0^i)^{-1}} \right)$$

$$2. \ \mathop{\mathbb{E}}_{t,m} \sqrt{\dot{\mu}(\boldsymbol{x}_{t,\star}^\top \boldsymbol{\theta}^\star)} \left( \sum_{i=1}^N \max_{\boldsymbol{z} \in \mathcal{X}_t^i} \|\breve{\boldsymbol{z}}\|_{(\boldsymbol{H}_{m-1}^i)^{-1}} \right)$$

Note that we can write

$$\boldsymbol{H}_m^i = \lambda \boldsymbol{I} + \sum_{t \in \mathcal{T}_m} \breve{\boldsymbol{x}}_t^i (\breve{\boldsymbol{x}}_t^i)^\top,$$

and hence, using Lemma A.11 and Lemma A.12 in tandem, we get

$$\mathop{\mathbb{E}}_{\mathcal{X}^i \sim \mathcal{D}^i} \max_{\boldsymbol{z} \in \mathcal{X}^i} \|\boldsymbol{z}\|_{(\boldsymbol{V}_0^i)^{-1}}^2 \leq \frac{8 d^2}{|\mathcal{T}_0|},$$

$$\mathop{\mathbb{E}}_{\mathcal{X}^i \sim \mathcal{D}_m^i} \max_{\boldsymbol{z} \in \mathcal{X}^i} \|\breve{\boldsymbol{z}}\|_{(\boldsymbol{H}_m^i)^{-1}}^2 \leq \frac{8 d^2}{|\mathcal{T}_m|}.$$

Using these bounds as well as the trivial bound of $\sqrt{\dot{\mu}(\boldsymbol{x}_{t,\star}^\top \boldsymbol{\theta}^\star)} \leq \sqrt{L_\mu}$, we upper bound the first term as follows:

$$\mathop{\mathbb{E}}_{t,m} \sqrt{\dot{\mu}(\boldsymbol{x}_{t,\star}^\top \boldsymbol{\theta}^\star)} \left( \sum_{i=1}^N \max_{\boldsymbol{z} \in \mathcal{X}_t^i} \|\breve{\boldsymbol{z}}\|_{(\boldsymbol{H}_{m-1}^i)^{-1}} \sum_{i=1}^N \|\boldsymbol{b}_t^i\|_{(\boldsymbol{V}_0^i)^{-1}} \right) \leq \mathop{\mathbb{E}}_{t,m} \sqrt{L_\mu \cdot N^2 \left( \sum_{i=1}^N \max_{\boldsymbol{z} \in \mathcal{X}_t^i} \|\breve{\boldsymbol{z}}\|_{(\boldsymbol{H}_{m-1}^i)^{-1}}^2 \right) \left( \sum_{i=1}^N \|\boldsymbol{b}_t^i\|_{(\boldsymbol{V}_0^i)^{-1}}^2 \right)}$$

$$\leq \sqrt{L_\mu \cdot N^2 \sum_{i=1}^N \mathop{\mathbb{E}}_{\mathcal{X}_t^i \sim \mathcal{D}_m^i} \max_{\boldsymbol{z} \in \mathcal{X}_t^i} \|\breve{\boldsymbol{z}}\|_{(\boldsymbol{H}_{m-1}^i)^{-1}}^2 \sum_{i=1}^N \mathop{\mathbb{E}}_{\mathcal{X}_t^i \sim \mathcal{D}_m^i} \|\boldsymbol{b}_t^i\|_{(\boldsymbol{V}_0^i)^{-1}}^2}$$

$$\leq \sqrt{L_\mu \cdot N^2 \sum_{i=1}^N \mathop{\mathbb{E}}_{\mathcal{X}_t^i \sim \mathcal{D}_{m-1}^i} \max_{\boldsymbol{z} \in \mathcal{X}_t^i} \|\breve{\boldsymbol{z}}\|_{(\boldsymbol{H}_{m-1}^i)^{-1}}^2 \sum_{i=1}^N \mathop{\mathbb{E}}_{\mathcal{X}_t^i \sim \mathcal{D}^i} \|\boldsymbol{b}_t^i\|_{(\boldsymbol{V}_0^i)^{-1}}^2}$$

$$\leq \sqrt{L_\mu \cdot N^2 \cdot \sum_{i=1}^N \frac{8 d^2}{|\mathcal{T}_{m-1}|} \cdot \sum_{i=1}^N \frac{8 d^2}{|\mathcal{T}_0|}}$$

$$\leq \frac{8 N^2 d^2 \sqrt{L_\mu}}{\sqrt{|\mathcal{T}_{m-1}||\mathcal{T}_0|}}.$$

Here, the first inequality uses the Cauchy-Schwarz inequality, and the second inequality uses Jensen's inequality. The third inequality is a consequence of Lemma A.8 while the second-to-last inequality follows from the bounds we showed above.

Hence, summing over all $t \in \mathcal{T}_m$, we get that:

$$\sum_{t \in \mathcal{T}_m} \mathop{\mathbb{E}}_{t,m} \sqrt{\dot{\mu}(\boldsymbol{x}_{t,\star}^\top \boldsymbol{\theta}^\star)} \left( \sum_{i=1}^{N} \max_{\boldsymbol{z} \in \mathcal{X}_t^i} \|\check{\boldsymbol{z}}\|_{(\boldsymbol{H}_{m-1}^i)^{-1}} \sum_{i=1}^{N} \|\boldsymbol{b}_t^i\|_{(\boldsymbol{V}_0^i)^{-1}} \right) \leq \frac{8N^2 d^2 \sqrt{L_\mu} |\mathcal{T}_m|}{\sqrt{|\mathcal{T}_{m-1}||\mathcal{T}_0|}}.$$

We now upper bound the second term using the Cauchy-Schwarz inequality as:

$$\sum_{t \in \mathcal{T}_m} \mathop{\mathbb{E}}_{t,m} \sqrt{\dot{\mu}(\boldsymbol{x}_{t,\star}^\top \boldsymbol{\theta}^\star)} \sum_{i=1}^{N} \max_{\boldsymbol{z} \in \mathcal{X}_t^i} \|\check{\boldsymbol{z}}\|_{(\boldsymbol{H}_{m-1}^i)^{-1}} \leq \sum_{t \in \mathcal{T}_m} \sqrt{\left( \mathop{\mathbb{E}}_{t,m} \dot{\mu}(\boldsymbol{x}_{t,\star}^\top \boldsymbol{\theta}^\star) \right) \mathop{\mathbb{E}}_{t,m} \left( \sum_{i=1}^{N} \max_{\boldsymbol{z} \in \mathcal{X}_t^i} \|\check{\boldsymbol{z}}\|_{(\boldsymbol{H}_{m-1}^i)^{-1}} \right)^2}$$

Now, since the optimal slate never gets eliminated (Lemma A.7), we can write

$$\sqrt{\mathop{\mathbb{E}}_{\{\mathcal{X}_t^i \sim \mathcal{D}_m^i\}_{i=1}^N} \dot{\mu}(\boldsymbol{x}_{t,\star}^\top \boldsymbol{\theta}^\star)} = \sqrt{\mathop{\mathbb{E}}_{\{\mathcal{X}_t^i \sim \mathcal{D}^i\}_{i=1}^N} \dot{\mu}(\boldsymbol{x}_{t,\star}^\top \boldsymbol{\theta}^\star)} = \sqrt{\mathop{\mathbb{E}}_{\{\mathcal{X}^i \sim \mathcal{D}^i\}_{i=1}^N} \dot{\mu}(\boldsymbol{x}_{\star}^\top \boldsymbol{\theta}^\star)}$$

where $\boldsymbol{x}_\star = \arg\max_{\boldsymbol{x} \in \mathcal{X}} \boldsymbol{x}^\top \boldsymbol{\theta}^\star$. Hence, this quantity is independent of $t$. We thus get

$$\sum_{t \in \mathcal{T}_m} \mathop{\mathbb{E}}_{t,m} \sqrt{\dot{\mu}(\boldsymbol{x}_{t,\star}^\top \boldsymbol{\theta}^\star)} \sum_{i=1}^{N} \max_{\boldsymbol{z} \in \mathcal{X}_t^i} \|\check{\boldsymbol{z}}\|_{(\boldsymbol{H}_{m-1}^i)^{-1}} \leq \sqrt{\mathop{\mathbb{E}}_{\{\mathcal{X}^i \sim \mathcal{D}^i\}_{i=1}^N} \dot{\mu}(\boldsymbol{x}_{\star}^\top \boldsymbol{\theta}^\star)} \underbrace{\sum_{t \in \mathcal{T}_m} \sqrt{\mathop{\mathbb{E}}_{t,m} \left( \sum_{i=1}^{N} \max_{\boldsymbol{z} \in \mathcal{X}_t^i} \|\check{\boldsymbol{z}}\|_{(\boldsymbol{H}_{m-1}^i)^{-1}} \right)^2}}_{\text{Term A}}.$$

Expanding the square in Term A gives us

$$\text{Term A} \leq \sum_{t \in \mathcal{T}_m} \sqrt{\underbrace{\sum_{i=1}^{N} \mathop{\mathbb{E}}_{\mathcal{X}_t^i \sim \mathcal{D}_m^i} \max_{\boldsymbol{z} \in \mathcal{X}_t^i} \|\check{\boldsymbol{z}}\|_{(\boldsymbol{H}_{m-1}^i)^{-1}}^2}_{\text{Term A1}} + 2 \underbrace{\mathop{\mathbb{E}}_{t,m} \sum_{i=1}^{N} \sum_{\substack{j=1 \\ j \neq i}}^{N} \max_{\substack{\boldsymbol{z} \in \mathcal{X}_t^i \\ \boldsymbol{u} \in \mathcal{X}_t^j}} \|\check{\boldsymbol{z}}\|_{(\boldsymbol{H}_{m-1}^i)^{-1}} \|\check{\boldsymbol{u}}\|_{(\boldsymbol{H}_{m-1}^j)^{-1}}}_{\text{Term A2}}}.$$

Upper-bounding Term A1, we get

$$\text{Term A1} = \sum_{i=1}^{N} \mathop{\mathbb{E}}_{\mathcal{X}_t^i \sim \mathcal{D}_m^i} \max_{\boldsymbol{z} \in \mathcal{X}_t^i} \|\check{\boldsymbol{z}}\|_{(\boldsymbol{H}_{m-1}^i)^{-1}}^2 \leq \sum_{i=1}^{N} \mathop{\mathbb{E}}_{\mathcal{X}_t^i \sim \mathcal{D}_{m-1}^i} \max_{\boldsymbol{z} \in \mathcal{X}_t^i} \|\check{\boldsymbol{z}}\|_{(\boldsymbol{H}_{m-1}^i)^{-1}}^2$$
$$\leq \frac{8 d^2 N}{|\mathcal{T}_{m-1}|}$$

where the first inequality follows from Lemma A.8, while the second inequality follows from using Lemma A.11 and Lemma A.12 in tandem as shown above.

We now upper bound Term A2 as follows:

$$\mathop{\mathbb{E}}_{t,m} \sum_{i=1}^{N} \sum_{\substack{j=1 \\ j \neq i}}^{N} \max_{\substack{\boldsymbol{z} \in \mathcal{X}_t^i \\ \boldsymbol{u} \in \mathcal{X}_t^j}} \|\check{\boldsymbol{z}}\|_{(\boldsymbol{H}_{m-1}^i)^{-1}} \|\check{\boldsymbol{u}}\|_{(\boldsymbol{H}_{m-1}^j)^{-1}} \leq 2 \mathop{\mathbb{E}}_{t,m} \sum_{i=1}^{N} \sum_{\substack{j=1 \\ j \neq i}}^{N} \max_{\substack{\boldsymbol{z} \in \mathcal{X}_t^i \\ \boldsymbol{u} \in \mathcal{X}_t^j}} \frac{\|\check{\boldsymbol{z}}\| \|\check{\boldsymbol{u}}\|}{\sqrt{\lambda_{\min}(\boldsymbol{H}_{m-1}^i) \lambda_{\min}(\boldsymbol{H}_{m-1}^j)}}$$
$$\leq 2 \mathop{\mathbb{E}}_{t,m} \sum_{i=1}^{N} \sum_{\substack{j=1 \\ j \neq i}}^{N} \frac{L_\mu}{N(\lambda + 0.5\rho |\mathcal{T}_m|)}$$

where the first inequality follows from Rayleigh's quotient, while the second inequality follows from the fact that $\|\check{z}\| \leq \sqrt{L_\mu N^{-1}}$. The second inequality also uses the linear growth of eigenvalues of the slot-level matrices, shown in Lemma A.13. Further simplification gives us a bound on Term A2 as

$$\text{Term A2} = 2 \underset{t,m}{\mathbb{E}} \sum_{i=1}^{N} \sum_{\substack{j=1 \\ j \neq i}}^{N} \max_{\substack{z \in \mathcal{X}_t^i \\ u \in \mathcal{X}_t^j}} \|\check{z}\|^2_{(H_{m-1}^i)^{-1}} \|\check{u}\|^2_{(H_{m-1}^j)^{-1}} \leq \frac{4L_\mu N}{\rho |\mathcal{T}_m|}.$$

Putting these bounds together, we get a bound on TermA as

$$\sum_{t \in \mathcal{T}_m} \sqrt{\underset{t,m}{\mathbb{E}} \left( \sum_{i=1}^{N} \max_{z \in \mathcal{X}_t^i} \|\check{z}\|_{(H_{m-1}^i)^{-1}} \right)^2} \leq \sqrt{\frac{8d^2 N |\mathcal{T}_m|^2}{|\mathcal{T}_{m-1}|} + \frac{4L_\mu N |\mathcal{T}_m|}{\rho}}.$$

Using Lemma A.9 and assembling all the bounds, we get that

$$\sum_{t \in \mathcal{T}_m} \underset{t,m}{\mathbb{E}} \left| \mu(\boldsymbol{x}_{t,\star}^\top \boldsymbol{\theta}^\star) - \mu(\boldsymbol{x}_t^\top \boldsymbol{\theta}^\star) \right|$$

$$\leq \frac{320 e^{3S} \gamma^2 N^2 d^2 \sqrt{\kappa L_\mu}}{S} \frac{|\mathcal{T}_m|}{\sqrt{|\mathcal{T}_{m-1}||\mathcal{T}_0|}} + 8\gamma \sqrt{\underset{\{\mathcal{X}^i \sim \mathcal{D}^i\}_{i=1}^N}{\mathbb{E}} \dot{\mu}(\boldsymbol{x}_\star^\top \boldsymbol{\theta}^\star)} \sqrt{\frac{8d^2 N |\mathcal{T}_m|^2}{|\mathcal{T}_{m-1}|} + \frac{4L_\mu N |\mathcal{T}_m|}{\rho}}.$$

$\square$

## A.4. Results on Optimal Designs

**Lemma A.11.** *(Corollary A.16, Sawarni et al. (2024)) Define* $\boldsymbol{A} = \lambda \boldsymbol{I} + \sum_{i=1}^{N} \boldsymbol{x}_i \boldsymbol{x}_i^\top$*. Then, for* $\lambda = \mathcal{O}(\log(Td))$*, we have*

$$\boldsymbol{A} \succeq \frac{N}{8} \underset{\mathcal{X} \sim \mathcal{D}}{\mathbb{E}} \underset{\boldsymbol{x} \sim \pi_G(\mathcal{X})}{\mathbb{E}} \left[ \boldsymbol{x} \boldsymbol{x}^\top \mid \mathcal{X} \right].$$

**Lemma A.12.** *(Lemma 4, Ruan et al. (2021)) Let*

$$\boldsymbol{W}_G = \underset{\mathcal{X} \sim \mathcal{D}}{\mathbb{E}} \underset{\boldsymbol{x} \sim \pi_G(\mathcal{X})}{\mathbb{E}} \left[ \boldsymbol{x} \boldsymbol{x}^\top \mid \mathcal{X} \right].$$

*Then, we have*

$$\underset{\mathcal{X} \sim \mathcal{D}}{\mathbb{E}} \max_{\boldsymbol{x} \in \mathcal{X}} \|\boldsymbol{x}\|^2_{\boldsymbol{W}_G^{-1}} \leq d^2.$$

## A.5. Showing Multiplicative Equivalence for `B-SlateGLinCB`

**Lemma A.13.** *Let* $\boldsymbol{H}_m$ *and* $\boldsymbol{H}_m^i$ *be defined as in Section A.1. Let* $|\mathcal{T}_m| \geq T(\boldsymbol{H}) := \frac{48L_\mu^2 + 8L_\mu N\rho}{3\rho^2}(N-1)^2 \log\left(\frac{2dNT}{\delta}\right)$*. Then, assuming the diversity assumptions (Section 2) hold, with high probability, we have that*

$$\frac{1}{4} diag(\boldsymbol{H}_m^1, \ldots, \boldsymbol{H}_m^N) \preceq \boldsymbol{H}_m \preceq \frac{7}{4} diag(\boldsymbol{H}_m^1, \ldots, \boldsymbol{H}_m^N).$$

*Consequently, for any* $\boldsymbol{x} = (\boldsymbol{x}^1, \ldots, \boldsymbol{x}^N)$*, we have that*

$$\|\boldsymbol{x}\|_{\boldsymbol{H}_m^{-1}} \leq 2 \sum_{i=1}^{N} \|\boldsymbol{x}^i\|_{(\boldsymbol{H}_m^i)^{-1}}.$$

*Proof.* Define $\overline{\boldsymbol{x}}_t := \sqrt{\dot{\mu}(\boldsymbol{b}_t^\top \widehat{\boldsymbol{\theta}}_0)\beta_t^{-1}} \boldsymbol{x}_t$ and $\overline{\boldsymbol{x}}_t^i := \sqrt{\dot{\mu}(\boldsymbol{b}_t^\top \widehat{\boldsymbol{\theta}}_0)\beta_t^{-1}} \boldsymbol{x}_t^i$. Then, note that $\|\overline{\boldsymbol{x}}_t^i\| \le \sqrt{L_\mu N^{-1}}$. Also, we can write

$$\boldsymbol{H}_m = \lambda \boldsymbol{I}_{Nd} + \sum_{t \in \mathcal{T}_m} \overline{\boldsymbol{x}}_t \overline{\boldsymbol{x}}_t^\top \quad \text{and} \quad \boldsymbol{H}_m^i = \lambda \boldsymbol{I} + \sum_{t \in \mathcal{T}_m} \overline{\boldsymbol{x}}_t^i \overline{\boldsymbol{x}}_t^{i\top}.$$

Also, for the sake of the proof, define $\boldsymbol{U}_m := \text{diag}(\boldsymbol{H}_m^1, \ldots, \boldsymbol{H}_m^N)$. Then, using Lemma B.1 from Goyal & Sinha (2025), we have that

$$\boldsymbol{U}_m^{-1/2} \boldsymbol{H}_m \boldsymbol{U}_m^{-1/2} = \boldsymbol{I}_{Nd} + \boldsymbol{G}_m.$$

where $(\boldsymbol{G}_m)_{ij} = \mathbb{1}\{i \ne j\}(\boldsymbol{H}_m^i)^{-1/2}\boldsymbol{H}_m^{(i,j)}(\boldsymbol{H}_m^j)^{-1/2}$ and $\boldsymbol{H}_m^{(i,j)} = \sum_{t \in \mathcal{T}_m} \overline{\boldsymbol{x}}_t^i \overline{\boldsymbol{x}}_t^{j\top}$. We now bound the norm of $\boldsymbol{H}_m^{(i,j)} \ \forall i \in [N]$ and $j \in [i+1, N]$. A straightforward application of Lemma D.2 from Goyal & Sinha (2025) shows that for fixed $i \in [N]$ and $j > i$ with the quantities $m_1 = m_2 = \sqrt{L_\mu N^{-1}}$, $d_1 = d_2 = d$ and $\delta = \frac{2\delta}{N(N-1)}$,

$$\mathbb{P}\left\{\exists t \ge 1 : \left\|\sum_{s \in [t]} \overline{\boldsymbol{x}}_s^i \overline{\boldsymbol{x}}_s^{j\top}\right\| \ge \sqrt{\frac{8L_\mu^2}{N^2}t \log\left(\frac{dN(N-1)}{\delta}\right)}\right\} \le \frac{2\delta}{N(N-1)}.$$

Taking a union bound over all $i \in [N]$ and $j \in [i+1, N]$ gives us the result for all pairs of $(i, j)$ where $j > i$. In particular, setting $t = |\mathcal{T}_m|$ gives us the result that for all pairs of $(i, j)$ where $j > i$, with high probability

$$\|\boldsymbol{H}_m^{(i,j)}\| \le \sqrt{\frac{8L_\mu^2}{N^2}|\mathcal{T}_m| \log\left(\frac{dN(N-1)}{\delta}\right)}.$$

Now, using the diversity conditions, we know that

$$\mathbb{E}[\boldsymbol{x}_t^i \boldsymbol{x}_t^{i\top} \mid \mathcal{F}_{t-1}] \succeq \rho \boldsymbol{I}.$$

Using the definition of $\kappa$, we can say that

$$\mathbb{E}[\overline{\boldsymbol{x}}_t^i \overline{\boldsymbol{x}}_t^{i\top} \mid \mathcal{F}_{t-1}] = \mathbb{E}[\dot{\mu}(\boldsymbol{b}_t^\top \widehat{\boldsymbol{\theta}}_0)\beta_t^{-1}\boldsymbol{x}_t^i \boldsymbol{x}_t^{i\top} \mid \mathcal{F}_{t-1}] \succeq \rho\kappa^{-1}\beta_t^{-1}\boldsymbol{I}.$$

Applying Lemma D.1 using the quantities $\alpha = \kappa^{-1}\beta_t^{-1}(\le 1)$, $m = \sqrt{L_\mu N^{-1}}$, $\gamma = \lambda$, $\delta = \frac{\delta}{N}$, and $c = 0.5$, for some fixed $i \in [N]$, with probability $1 - \frac{\delta}{N}$,

$$\lambda_{\min}\left(\lambda \boldsymbol{I} + \sum_{s \in [t]} \overline{\boldsymbol{x}}_s^i \overline{\boldsymbol{x}}_s^{i\top}\right) \ge \lambda + \frac{\rho t}{2} \ \forall t \ge \frac{48L_\mu^2 + 8L_\mu N\rho}{3\rho^2 N^2} \log\left(\frac{2dNT}{\delta}\right).$$

Using the fact that $(N-1)^2 \ge N^{-2}$, we also have that, with probability $1 - \frac{\delta}{N}$,

$$\lambda_{\min}\left(\lambda \boldsymbol{I} + \sum_{s \in [t]} \overline{\boldsymbol{x}}_s^i \overline{\boldsymbol{x}}_s^{i\top}\right) \ge \lambda + \frac{\rho t}{2} \ \forall t \ge T(\boldsymbol{H}) := \frac{48L_\mu^2 + 8L_\mu N\rho}{3\rho^2}(N-1)^2 \log\left(\frac{2dNT}{\delta}\right).$$

A union bound over all slots gives us this result for all $i \in [N]$. In particular, let $|\mathcal{T}_m| \ge T(\boldsymbol{H})$. Then, setting $t = |\mathcal{T}_m|$ gives us the result that for all $i \in [N]$, with high probability,

$$\lambda_{\min}\left(\boldsymbol{H}_m^i\right) \ge \lambda + \frac{\rho|\mathcal{T}_m|}{2}.$$

Now, for $i \in [N-1]$, define $\boldsymbol{Z}_m^i \in \mathbb{R}^{d \times id}$ as the following matrix: for $j \in [i]$, the $j^{th}$ $d \times d$ block of $\boldsymbol{Z}_m^i$ is given by $(\boldsymbol{H}_m^{N-i})^{-1/2}\boldsymbol{H}_m^{(N-i, N-i+j)}(\boldsymbol{H}_m^{N-i+j})^{-1/2}$.

Then, using Lemma B.7 from Goyal & Sinha (2025), we have that,

$$\|\boldsymbol{Z}_m^i\| \leq \sum_{j \in [i]} \frac{\|\boldsymbol{H}_m^{N-i,N-i+j}\|}{\sqrt{\lambda_{\min}(\boldsymbol{H}_m^{N-i})\lambda_{\min}(\boldsymbol{H}_m^{N-i+j})}}$$

$$\leq \sum_{j \in [i]} \frac{\sqrt{\frac{8L_\mu^2}{N^2}|\mathcal{T}_m|\log\left(\frac{dN(N-1)}{\delta}\right)}}{\lambda + 0.5\rho|\mathcal{T}_m|}$$

$$\leq \sum_{j \in [i]} \sqrt{\frac{32L_\mu^2 \log\left(\frac{dN(N-1)}{\delta}\right)}{N^2\rho^2|\mathcal{T}_m|}}.$$

Using the fact that $|\mathcal{T}_m| \geq T(\boldsymbol{H})$, we get that

$$\|\boldsymbol{Z}_m^i\| \leq \sum_{j \in [i]} \sqrt{\frac{96L_\mu^2 \log\left(\frac{dN(N-1)}{\delta}\right)}{N^2(48L_\mu^2 + 8L_\mu N\rho)(N-1)^2 \log\left(\frac{2dNT}{\delta}\right)}} \leq \frac{3i}{2N(N-1)}$$

where the last inequality follows from the fact that $\sqrt{2} \leq \frac{3}{2}$.

Finally, recall the definition of the matrix $\boldsymbol{G}_m$:

$$(\boldsymbol{G}_m)_{ij} = \mathbb{1}\{i \neq j\}(\boldsymbol{H}_m^i)^{-1/2}\boldsymbol{H}_m^{i,j}(\boldsymbol{H}_m^j)^{-1/2}.$$

It is easy to see that we can write $\boldsymbol{G}_m$ as the following matrix recurrence relation: for $i \in [1, N-1]$, define

$$\boldsymbol{G}_m^1 = \begin{bmatrix} \boldsymbol{0} & \boldsymbol{Z}_m^1 \\ (\boldsymbol{Z}_m^1)^\top & \boldsymbol{0} \end{bmatrix}, \quad \boldsymbol{G}_m^i = \begin{bmatrix} \boldsymbol{0} & \boldsymbol{Z}_m^i \\ (\boldsymbol{Z}_m^i)^\top & \boldsymbol{G}_m^{i-1} \end{bmatrix}.$$

Then, $\boldsymbol{G}_m = \boldsymbol{G}_m^{N-1}$. Using Lemma B.2 from Goyal & Sinha (2025) gives us:

$$\lambda_{\max}(\boldsymbol{G}_m) \leq \sum_{i \in [N-1]} \|\boldsymbol{Z}_m^i\| = \frac{3}{2} \sum_{i \in [N-1]} \frac{i}{N(N-1)} = \frac{3}{4},$$

$$\lambda_{\min}(\boldsymbol{G}_m) \geq -\sum_{i \in [N-1]} \|\boldsymbol{Z}_m^i\| = -\frac{3}{2} \sum_{i \in [N-1]} \frac{i}{N(N-1)} = -\frac{3}{4}.$$

Substituting $\boldsymbol{G}_m = \boldsymbol{U}_m^{-1/2}\boldsymbol{H}_m\boldsymbol{U}_m^{-1/2} - \boldsymbol{I}_{Nd}$, we get that

$$\frac{1}{4}\boldsymbol{U}_m \preceq \boldsymbol{H}_m \preceq \frac{7}{4}\boldsymbol{U}_m.$$

This finishes the first part of the proof. For the second part, notice that,

$$\boldsymbol{H}_m^{-1} \preceq 4\,\text{diag}((\boldsymbol{H}_m^1)^{-1}, \ldots, (\boldsymbol{H}_m^N)^{-1}).$$

Also, note that $\boldsymbol{x} = \sum_{i=1}^N (\boldsymbol{x}^i \otimes \boldsymbol{e}_i)$ (Section A.1) and hence, an application of the triangle inequality gives us:

$$\|\boldsymbol{x}\|_{\boldsymbol{H}_m^{-1}} \leq 2 \sum_{i \in [N]} \|\boldsymbol{x}^i \otimes \boldsymbol{e}_i\|_{\text{diag}((\boldsymbol{H}_m^1)^{-1}, \ldots, (\boldsymbol{H}_m^N)^{-1})} \leq 2 \sum_{i=1}^N \|\boldsymbol{x}^i\|_{(\boldsymbol{H}_m^i)^{-1}}.$$

which finishes the proof for the second part. □

**Lemma A.14.** *Let $V_0$ and $V_0^i$ be defined as in Section A.1. Let $|\mathcal{T}_0| \geq T(V) := \frac{48+8N\rho}{3\rho^2}(N-1)^2 \log\left(\frac{2dNT}{\delta}\right)$. Then, assuming the diversity assumptions (Section 2) hold, with high probability, we have that*

$$\frac{1}{4} diag(V_0^1, \ldots, V_0^N) \preceq V_0 \preceq \frac{7}{4} diag(V_0^1, \ldots, V_0^N).$$

*Consequently, for any $x = (x^1, \ldots, x^N)$, we have that*

$$\|x\|_{V_0^{-1}} \leq 2 \sum_{i=1}^{N} \|x^i\|_{(V_0^i)^{-1}}.$$

*Proof.* The proof follows on the same lines as that of Lemma A.13. First, we have that $\|x_t^i\| \leq N^{-1/2}$. Also, for the sake of the proof, define $W_0 := diag(V_0^1, \ldots, V_0^N)$. Then, using Lemma B.1 from Goyal & Sinha (2025), we have that

$$W_0^{-1/2} V_0 W_0^{-1/2} = I_{Nd} + G_0.$$

where $(G_0)_{ij} = \mathbb{1}\{i \neq j\}(V_0^i)^{-1/2} V_0^{(i,j)} (V_0^j)^{-1/2}$ and $V_0^{(i,j)} = \sum_{t \in \mathcal{T}_0} x_t^i x_t^{j\top}$. A straightforward application of Lemma D.2 from Goyal & Sinha (2025) shows that for fixed $i \in [N]$ and $j > i$ with the quantities $m_1 = m_2 = N^{-1/2}$, $d_1 = d_2 = d$ and $\delta = \frac{2\delta}{N(N-1)}$,

$$\mathbb{P}\left\{\exists t \geq 1 : \left\|\sum_{s \in [t]} x_s^i x_s^{j\top}\right\| \geq \sqrt{\frac{8t}{N^2} \log\left(\frac{dN(N-1)}{\delta}\right)}\right\} \leq \frac{2\delta}{N(N-1)}.$$

Taking a union bound over all $i \in [N]$ and $j \in [i+1, N]$ gives us the result for all pairs of $(i,j)$ where $j > i$. In particular, setting $T = |\mathcal{T}_0|$ gives us the result that for all pairs of $(i,j)$ where $j > i$, with high probability

$$\|V_0^{(i,j)}\| \leq \sqrt{\frac{8|\mathcal{T}_0|}{N^2} \log\left(\frac{dN(N-1)}{\delta}\right)}.$$

Now, using the diversity conditions, we know that

$$\mathbb{E}[x_t^i x_t^{i\top} \mid \mathcal{F}_{t-1}] \succeq \rho I.$$

Similar to Lemma A.13, an application of Lemma D.1 for a fixed $i \in [N]$ using the quantities $\alpha = 1$, $m = N^{-1/2}$, $\gamma = \lambda$, $\delta = \frac{\delta}{N}$ and $c = 0.5$, followed by the utilization of the fact that $(N-1)^2 \geq N^{-2}$, and finishing with a union bound over all $i \in [N]$ gives us that for all $i \in [N]$, with high probability,

$$\lambda_{\min}\left(\lambda I + \sum_{s \in [t]} x_s^i x_s^{i\top}\right) \geq \lambda + \frac{\rho t}{2} \quad \forall t \geq T(V) := \frac{48+8N\rho}{3\rho^2}(N-1)^2 \log\left(\frac{2dNT}{\delta}\right).$$

In particular, let $|\mathcal{T}_0| \geq T(V)$. Then, setting $t = |\mathcal{T}_0|$ gives us the result that for all $i \in [N]$, with high probability,

$$\lambda_{\min}\left(V_0^i\right) \geq \lambda + \frac{\rho|\mathcal{T}_0|}{2}.$$

Now, for $i \in [N-1]$, define $Z_0^i \in \mathbb{R}^{d \times id}$ as the following matrix: for $j \in [i]$, the $j^{th}$ $d \times d$ block of $Z_0^i$ is given by $(V_0^{N-i})^{-1/2} V_0^{(N-i, N-i+j)} (V_0^{N-i+j})^{-1/2}$.

Then, using Lemma B.7 from Goyal & Sinha (2025) and following a similar approach to that shown in Lemma A.13, we have that,

$$\|Z_0^i\| \leq \sum_{j \in [i]} \sqrt{\frac{96 \log\left(\frac{dN(N-1)}{\delta}\right)}{N^2(48+8N\rho)(N-1)^2 \log\left(\frac{2dNT}{\delta}\right)}} \leq \frac{3i}{2N(N-1)}.$$

Writing $\boldsymbol{G}_0$ as a matrix recurrence relation as in Lemma A.13 and using Lemma B.2 from Goyal & Sinha (2025) gives:

$$\lambda_{\max}(\boldsymbol{G}_0) \leq \frac{3}{4} \quad \text{and} \quad \lambda_{\min}(\boldsymbol{G}_0) \geq -\frac{3}{4}.$$

Substituting $\boldsymbol{G}_0 = \boldsymbol{W}_0^{-1/2} \boldsymbol{V}_0 \boldsymbol{W}_0^{-1/2} - \boldsymbol{I}_{Nd}$, we get that

$$\frac{1}{4} \boldsymbol{W}_0 \preceq \boldsymbol{V}_0 \preceq \frac{7}{4} \boldsymbol{W}_0.$$

This finishes the proof for the first part. The second part of the proof is exactly the same as in Lemma A.13. $\qquad\square$

### A.6. Other Relevant Lemmas

**Lemma A.15.** *Let $X^\alpha \geq k' \log(kX)$, where $k, k' > 0$, $k^\alpha k' \geq \alpha e$, and $\alpha \in (0, 1]$. Also, assume $X \geq k^{-1} \exp(\alpha^{-1})$. Then, we have that*

$$X \geq \frac{1}{k} \exp\left(-\frac{1}{\alpha} W_{-1}\left(-\frac{\alpha}{k^\alpha k'}\right)\right)$$

*where $W_{-1}(.)$ denotes the decreasing branch of the Lambert W function, defined as*

$$W_{-1} : [-1/e, 0) \mapsto (-\infty, -1], \quad W_{-1}(xe^x) = x \ \forall \ x \leq -1.$$

*Proof.* Define the function $f(x) = xe^x$. Note that $f'(x) = e^x(x+1)$, and hence, $f'$ is non-negative for $x \geq -1$. In other words, $f$ is increasing in the domain $[-1, \infty)$ and decreasing in the domain $(-\infty, -1]$.

Now, let us consider the function $f^{-1}(x)$. $f^{-1}$ is increasing in the domain $[f(-1), \lim_{x \to \infty} f(x)) = [-e^{-1}, \infty)$; we denote this branch of $f^{-1}$ as $W_0$. The other branch $W_{-1}$ is decreasing in the domain $[-e^{-1}, \lim_{x \to -\infty} f(x)) = (-e^{-1}, 0)$.

Now, rearranging the terms of $X^\alpha \geq k' \log(kX)$ and dividing both sides by $k^\alpha$ gives us,

$$\frac{1}{k^\alpha k'} \geq \exp(-\alpha \log(kX)) \log(kX).$$

Setting $Y = -\alpha \log(kX)$, we get

$$-\frac{\alpha}{k^\alpha k'} \leq Y \cdot \exp(Y).$$

Now, to apply the function $W_{-1}$ to both sides of the inequality, we require $-\alpha(k^\alpha k')^{-1} \in [-e^{-1}, 0)$ and $Y \exp(Y) \in [-e^{-1}, 0)$, and more particularly, $Y \leq -1$.

First, since $k, k' > 0$, we have that $-\alpha(k^\alpha k')^{-1} < 0$. Also, since $k^\alpha k' \geq \alpha e$, we have that $-\alpha(k^\alpha k')^{-1} \geq -e^{-1}$. Thus, we have that $-\alpha(k^\alpha k')^{-1} \in [-e^{-1}, 0)$.

For the second requirement, note that $Y \exp(Y) \in [-e^{-1}, 0)$ is satisfied for all $Y < 0$. However, since the range of $W_{-1}$ is $(-\infty, -1]$, we have that $W_{-1}(Y \exp(Y)) = Y$ if and only if $Y \leq -1$. The condition $X \geq k^{-1} \exp(\alpha^{-1})$ ensures $Y \leq -1$, thus, satisfying the second requirement.

Thus, applying $W_{-1}$ to both sides of the inequality, and using the fact that $W_{-1}$ is decreasing gives us:

$$W_{-1}\left(-\frac{\alpha}{k^\alpha k'}\right) \geq Y = -\alpha \log(kX).$$

Rearranging once again results in

$$X \geq \frac{1}{k} \exp\left(-\frac{1}{\alpha} W_{-1}\left(-\frac{\alpha}{k^\alpha k'}\right)\right).$$

$\qquad\square$

# B. `B-SlateGLinCB` with Distributional Optimal Designs

In this section, we first present an alternate version of `B-SlateGLinCB`, where we utilize the Distributional Optimal Design (Ruan et al., 2021) instead of the G-Optimal design. We then present the regret guarantees for this algorithm and a proof for the same.

---

**Algorithm 3** `B-SlateGLinCB` with Distributional Optimal Designs

---

1: **Inputs:** Number of Slots $N$, Number of batches $M$, Horizon $T$, Parameter norm bound $S$, Failure Level $\delta$, and non-linearity $\kappa$.
2: Initialize $\boldsymbol{\theta}_0 = \mathbf{0}_{Nd}$ and $\lambda = \mathcal{O}\left(NdR^2 \log(T/\delta)\right)$ and define batches $\mathcal{T}_0, \mathcal{T}_1, \dots, \mathcal{T}_M$ as per (4).
3: {Warmup Batch}
4: **for** $t \in \mathcal{T}_0$ **do**
5:    Receive the set of items $\mathcal{X}_t^i$ for all slots $i \in [N]$.
6:    Play the slate $\boldsymbol{x}_t = (\boldsymbol{x}_t^1, \dots, \boldsymbol{x}_t^N)$, where $\boldsymbol{x}_t^i \sim \pi_G(\mathcal{X}_t^i)$ (defined in Section (2.7)), and obtain $r_t$.
7: **end for**
8: Compute $\widehat{\boldsymbol{\theta}}_0 = \arg\min_{\boldsymbol{\theta}} \sum_{t \in \mathcal{T}_0} \ell(\boldsymbol{x}_t, r_t, \boldsymbol{\theta})$ as per (3) and $\boldsymbol{V}_0^i = \lambda \boldsymbol{I}_d + \sum_{t \in \mathcal{T}_0} \boldsymbol{x}_t^i {\boldsymbol{x}_t^i}^\top$ for all slots $i \in [N]$.
   //Other batches
9: **for** $m \in [M]$ **do**
10:    **for** $t \in \mathcal{T}_m$ **do**
11:       Receive the set of items $\mathcal{X}_t^i$ for all slots $i \in [N]$.
12:       **for** $i \in [N]$ **do**
13:          **for** $k \in [0, m-1]$ **do**
14:             $\mathcal{X}_t^i \leftarrow \{\boldsymbol{z} \in \mathcal{X}_t^i : \text{UCB}^{i,k}(\boldsymbol{z}) \geq \max_{\boldsymbol{y} \in \mathcal{X}_t^i} \text{LCB}^{i,k}(\boldsymbol{y})\}$ {Perform elimination}.
15:          **end for**
16:       **end for**
17:       Play the slate $\boldsymbol{x}_t = (\boldsymbol{x}_t^1, \dots, \boldsymbol{x}_t^N)$, where $\boldsymbol{x}_t^i \sim \pi_m^i(\mathcal{X}_t^i)$ and obtain $r_t$. Construct the slate $\boldsymbol{b}_t = (\boldsymbol{b}_t^1, \dots, \boldsymbol{b}_t^N)$, where $\boldsymbol{b}_t^i = \arg\max_{\boldsymbol{z} \in \mathcal{X}_t^i} \|\boldsymbol{z}\|_{(\boldsymbol{V}_0^i)^{-1}}$.
18:    **end for**
19:    Divide $\mathcal{T}_m$ into two sets $\mathcal{T}_{m,A}$ and $\mathcal{T}_{m,B}$ such that $\mathcal{T}_{m,A} \cap \mathcal{T}_{m,B} = \emptyset$ and $\mathcal{T}_{m,A} \cup \mathcal{T}_{m,B} = \mathcal{T}_m$.
20:    Using (3) and (9), compute

$$\widehat{\boldsymbol{\theta}}_m = \arg\min_{\boldsymbol{\theta}} \sum_{t \in \mathcal{T}_{m,A}} \ell(\boldsymbol{x}_t, r_t, \boldsymbol{\theta}) \quad \text{and} \quad \boldsymbol{H}_m^i = \lambda \boldsymbol{I}_d + \sum_{t \in \mathcal{T}_{m,A}} \frac{\dot{\mu}(\boldsymbol{b}_t^\top \widehat{\boldsymbol{\theta}}_0)}{\beta_t} \boldsymbol{x}_t^i {\boldsymbol{x}_t^i}^\top \ \forall i \in [N].$$

21:    Compute $\pi_{m+1}^i$ (Algorithm 3, Ruan et al. (2021)) with the set $\{\mathcal{X}_t^i\}_{t \in \mathcal{T}_{m,\beta}}$ for all $i \in [N]$.
22: **end for**

---

**Theorem B.1.** *Let $R(T)$ denote the regret of Algorithm 3. If*

$$\sqrt{\frac{2dN}{\delta}} \frac{(48 + 8N\rho)(N-1)^2}{3\rho^2} \geq \frac{e}{2}$$

*and*

$$T \geq T_0 := \frac{\delta}{2dN} \exp\left(-2W_{-1}\left(\frac{-3\rho^2\sqrt{\delta}}{2\sqrt{2dN}(48L_\mu^2 + 8L_\mu N\rho)(N-1)^2}\right)\right)$$

*where $W_{-1}$ denotes the decreasing branch of the Lambert W function (see Lemma A.15), then,*

$$R(T) = \tilde{\mathcal{O}}\left(RSNd\sqrt{T} \cdot \min\left\{\sqrt{d \cdot \mathop{\mathbb{E}}_{\{\mathcal{X}^i \sim \mathcal{D}^i\}_{i=1}^N} \dot{\mu}(\boldsymbol{x}_\star^\top \boldsymbol{\theta}^\star)}, \sqrt{N \cdot \max_{\{\mathcal{X}^i \sim \mathcal{D}^i\}_{i=1}^N} \dot{\mu}(\boldsymbol{x}_\star^\top \boldsymbol{\theta}^\star)}\right\}\right).$$

*where $\boldsymbol{x}_\star = \arg\max_{\boldsymbol{x} \in \mathcal{X}} \boldsymbol{x}^\top \boldsymbol{\theta}^\star$ and $\mathcal{X} = \mathcal{X}^1 \times \dots \times \mathcal{X}^N$.*

*Proof.* The proof follows on the same lines as the proof for Theorem A.1. However, the use of distributional optimal designs

prompts a change in the way we bound the regret for batches $m \geq 2$ (Lemma B.3). Define the optimal slate $\boldsymbol{x}_{t,\star}$ as

$$\boldsymbol{x}_{t,\star} = \arg\max_{\boldsymbol{x} \in \mathcal{X}_t} \boldsymbol{x}^\top \boldsymbol{\theta}^\star.$$

The regret for Algorithm 3 can be written as:

$$R(T) \leq \mathop{\mathbb{E}}_{\{\mathcal{X}_t^i \sim \mathcal{D}^i\}_{i=1}^N} \left[ \sum_{m \in [M]} \sum_{t \in \mathcal{T}_m} |\mu(\boldsymbol{x}_{t,\star}^\top \boldsymbol{\theta}^\star) - \mu(\boldsymbol{x}_t^\top \boldsymbol{\theta}^\star)| \right].$$

Similar to the proof in Theorem A.1, we choose the batch lengths as

$$|\mathcal{T}_0| = \lfloor \sqrt{T} \rfloor \quad \text{and} \quad |\mathcal{T}_m| = \lfloor T^{1-2^{-m}} \rfloor, m \geq 1.$$

Also, for $T \geq T_0$, the conditions of Lemma B.3 are satisfied. Thus, using Lemma B.3 for batches $m \geq 2$ as well as a trivial regret bound of R for each round $t \in \{\mathcal{T}_0, \mathcal{T}_1\}$ to obtain

$$R(T) \leq R(|\mathcal{T}_0| + |\mathcal{T}_1|) + \sum_{m \in [2,M]} \frac{320 e^{3S} \gamma^2 N^2 d^2 \sqrt{\kappa L_\mu}}{S} \frac{|\mathcal{T}_m|}{\sqrt{|\mathcal{T}_{m-1}||\mathcal{T}_0|}} +$$

$$\sum_{m \in [2,M]} 8\gamma \min \left\{ \sqrt{\mathop{\mathbb{E}}_{\{\mathcal{X}^i \sim \mathcal{D}^i\}_{i=1}^N} \dot{\mu}(\boldsymbol{x}_\star^\top \boldsymbol{\theta}^\star)} \sqrt{\frac{16 d^2 N |\mathcal{T}_m|^2}{|\mathcal{T}_{m-1}|} + \frac{4 L_\mu N |\mathcal{T}_m|}{\rho}}, 4N \sqrt{\max_{\{\mathcal{X}^i \sim \mathcal{D}^i\}_{i=1}^N} \dot{\mu}(\boldsymbol{x}_\star^\top \boldsymbol{\theta}^\star) \cdot d \log d} \frac{|\mathcal{T}_m|}{\sqrt{|\mathcal{T}_{m-1}|}} \right\}.$$

Substituting the values of $|\mathcal{T}_m|, \frac{|\mathcal{T}_m|}{\sqrt{|\mathcal{T}_{m-1}|}}, \frac{|\mathcal{T}_m|^2}{|\mathcal{T}_{m-1}|}$ from Theorem A.1, and using $|\mathcal{T}_m| \leq T$ and $|\mathcal{T}_0| = \lfloor \sqrt{T} \rfloor \geq \sqrt{T}/2$, we get

$$R(T) \leq 320\sqrt{2} e^{3S} S^{-1} \gamma^2 N^2 d^2 \sqrt{\kappa L_\mu} T^{1/4} \log\log T + 2R\sqrt{T} +$$

$$8\gamma \min \left\{ \sqrt{\mathop{\mathbb{E}}_{\{\mathcal{X}^i \sim \mathcal{D}^i\}_{i=1}^N} \dot{\mu}(\boldsymbol{x}_\star^\top \boldsymbol{\theta}^\star)} \sqrt{16 d^2 N + 4 L_\mu N \rho^{-1}}, 4N \sqrt{\max_{\{\mathcal{X}^i \sim \mathcal{D}^i\}_{i=1}^N} \dot{\mu}(\boldsymbol{x}_\star^\top \boldsymbol{\theta}^\star) \cdot d \log d} \right\} \sqrt{T} \log\log T.$$

Substituting the value of $\gamma = \mathcal{O}\left(SR\sqrt{Nd\log(T\delta^{-1})}\right)$ from Lemma A.2 gives us

$$R(T) = \tilde{O}\left(RSNd\sqrt{T} \cdot \min\left\{ \sqrt{d \cdot \mathop{\mathbb{E}}_{\{\mathcal{X}^i \sim \mathcal{D}^i\}_{i=1}^N} \dot{\mu}(\boldsymbol{x}_\star^\top \boldsymbol{\theta}^\star)}, \sqrt{N \cdot \max_{\{\mathcal{X}^i \sim \mathcal{D}^i\}_{i=1}^N} \dot{\mu}(\boldsymbol{x}_\star^\top \boldsymbol{\theta}^\star)} \right\}\right).$$

$\square$

*Remark* B.2. From the proof of Theorem B.1 and Lemma B.3 , we see that the dependence in $N$, $d$, and the reward sensitivity of the optimal slates is a result of bounding the following quantity for each batch:

$$\sum_{t \in \mathcal{T}_m} \mathop{\mathbb{E}}_{\{\boldsymbol{X}_t^i \sim \mathcal{D}_m^i\}_{i=1}^N} \sqrt{\dot{\mu}(\boldsymbol{x}_{t,\star}^\top \boldsymbol{\theta}^\star)} \left( \sum_{i=1}^N \max_{\boldsymbol{x}^i \in \mathcal{X}_t^i} \|\breve{\boldsymbol{x}}^i\|_{(\boldsymbol{H}_{m-1}^i)^{-1}} \right).$$

In the proof of Lemma B.3, we bound this quantity using two different methods. The first method involves using ideas similar to Lemma A.10, i.e, we use the Cauchy-Schwarz inequality to help us control the dependence on $N$. Using this method results in a dependence on the expected reward sensitivity of the optimal slates. Furthermore, the resulting dependence on $d$ is now $\mathcal{O}(d^{3/2})$. This is because, for a Distributional Optimal Design $\pi$, the best bound on the quantity

$$\mathop{\mathbb{E}}_{\mathcal{X} \sim \mathcal{D}} \max_{\boldsymbol{x} \in \mathcal{X}} \|\boldsymbol{x}\|_{\boldsymbol{W}^{-1}}^2, \quad \boldsymbol{W} = \mathop{\mathbb{E}}_{\mathcal{X} \sim \mathcal{D}} \mathop{\mathbb{E}}_{\boldsymbol{x} \sim \pi(\mathcal{X})} \boldsymbol{x}\boldsymbol{x}^\top$$

is known to be $\mathcal{O}(d^2)$, i.e, the best bound we can obtain asymptotically matches the bound obtained using a G-Optimal design. Ruan et al. (2021) are unable to provide a bound for this quantity (see Theorem 5, Ruan et al. (2021)), and Sawarni

et al. (2024) naively bound this quantity using the fact that the Distributional Optimal Design samples from a G-Optimal Design with half probability (see Lemma A.14, Sawarni et al. (2024)).

Now, the second method allows us to leverage the improved optimal design bound provided by Distributional Optimal Designs, which is

$$\mathbb{E}_{\mathcal{X}\sim\mathcal{D}}\max_{\boldsymbol{x}\in\mathcal{X}}\|\boldsymbol{x}\|_{\boldsymbol{W}^{-1}} = \mathcal{O}(d\log d), \quad \boldsymbol{W} = \mathbb{E}_{\mathcal{X}\sim\mathcal{D}}\mathbb{E}_{\boldsymbol{x}\sim\pi(\mathcal{X})}\boldsymbol{x}\boldsymbol{x}^{\top}.$$

Thus, to use this method, we avoid using the Cauchy-Schwartz inequality, which ensures that we do not have to deal with the square of the normed terms. This method improves the dependence on $d$ by a factor of $\sqrt{d}$. Simultaneously, this method worsens the dependence on $N$ by a factor of $\sqrt{N}$. Also, the regret bound now depends on the maximum reward sensitivity of the optimal slates as compared to the expected reward sensitivity (this gap can be significant for several distributions, see Section 2.1, Sawarni et al. (2024)). Thus, in the ideal scenario, optimal dependence on $N$ and the reward sensitivity can be obtained using our first method, however, obtaining optimal dependence on $d$ requires a tighter bound on the quantity

$$\mathbb{E}_{\mathcal{X}\sim\mathcal{D}}\max_{\boldsymbol{x}\in\mathcal{X}}\|\boldsymbol{x}\|_{\boldsymbol{W}^{-1}}^2, \quad \boldsymbol{W} = \mathbb{E}_{\mathcal{X}\sim\mathcal{D}}\mathbb{E}_{\boldsymbol{x}\sim\pi(\mathcal{X})}\boldsymbol{x}\boldsymbol{x}^{\top}.$$

Whether we can obtain a tighter bound on this quantity using Distributional Optimal Designs remains unclear.

## B.1. Supporting Lemmas for Theorem B.1

**Lemma B.3.** *At round $t \in \mathcal{T}_m$, let $\boldsymbol{x}_{t,\star}$ be the optimal slate, i.e,*

$$\boldsymbol{x}_{t,\star} = \arg\max_{\boldsymbol{x}\in\mathcal{X}_t} \boldsymbol{x}^{\top}\boldsymbol{\theta}^{\star}.$$

*If $|\mathcal{T}_0| \geq T(\boldsymbol{V})$ and $|\mathcal{T}_k| \geq T(\boldsymbol{H})$ for all $1 \leq k \leq m-1$, then, we have*

$$\sum_{t\in\mathcal{T}_m}\mathbb{E}_{\{\mathcal{X}_t^i\sim\mathcal{D}_m^i\}_{i=1}^N}\left|\mu(\boldsymbol{x}_{t,\star}^{\top}\boldsymbol{\theta}^{\star}) - \mu(\boldsymbol{x}_t^{\top}\boldsymbol{\theta}^{\star})\right| \leq \frac{320e^{3S}\gamma^2 N^2 d^2\sqrt{\kappa L_\mu}}{S}\frac{|\mathcal{T}_m|}{\sqrt{|\mathcal{T}_{m-1}||\mathcal{T}_0|}} +$$

$$8\gamma\min\left\{\sqrt{\mathbb{E}_{\{\mathcal{X}^i\sim\mathcal{D}^i\}_{i=1}^N}\dot{\mu}(\boldsymbol{x}_\star^{\top}\boldsymbol{\theta}^{\star})}\sqrt{\frac{16d^2 N|\mathcal{T}_m|^2}{|\mathcal{T}_{m-1}|} + \frac{4L_\mu N|\mathcal{T}_m|}{\rho}}, 4N\sqrt{\max_{\{\mathcal{X}^i\sim\mathcal{D}^i\}_{i=1}^N}\dot{\mu}(\boldsymbol{x}_\star^{\top}\boldsymbol{\theta}^{\star})\cdot d\log d}\frac{|\mathcal{T}_m|}{\sqrt{|\mathcal{T}_{m-1}|}}\right\}$$

*where $\boldsymbol{x}_\star = \arg\max_{\boldsymbol{x}\in\mathcal{X}} \boldsymbol{x}^{\top}\boldsymbol{\theta}^{\star}$ and $\mathcal{X} = \mathcal{X}^1 \times \ldots \times \mathcal{X}^N$.*

*Proof.* The proof follows on the lines of Lemma A.10. Define $\breve{\boldsymbol{x}} := \sqrt{\dot{\mu}(\boldsymbol{b}_t^{\top}\widehat{\boldsymbol{\theta}}_0)\beta_t^{-1}}\boldsymbol{x}$ and $\mathbb{E}_{\{\mathcal{X}_t^i\sim\mathcal{D}_m^i\}_{i=1}^N}[.]$ as $\mathbb{E}_{t,m}[.]$. Then, using Lemma A.9, we wish to bound the following two terms (excluding constants):

$$1. \mathbb{E}_{t,m}\sqrt{\dot{\mu}(\boldsymbol{x}_{t,\star}^{\top}\boldsymbol{\theta}^{\star})}\left(\sum_{i=1}^N\max_{\boldsymbol{z}\in\mathcal{X}_t^i}\|\breve{\boldsymbol{z}}\|_{(\boldsymbol{H}_{m-1}^i)^{-1}}\right)\left(\sum_{i=1}^N\|\boldsymbol{b}_t^i\|_{(\boldsymbol{V}_0^i)^{-1}}\right)$$

$$2. \mathbb{E}_{t,m}\sqrt{\dot{\mu}(\boldsymbol{x}_{t,\star}^{\top}\boldsymbol{\theta}^{\star})}\left(\sum_{i=1}^N\max_{\boldsymbol{z}\in\mathcal{X}_t^i}\|\breve{\boldsymbol{z}}\|_{(\boldsymbol{H}_{m-1}^i)^{-1}}\right)$$

Using Lemma A.11 and Lemma B.4 in tandem, we get

$$\mathbb{E}_{\mathcal{X}^i\sim\mathcal{D}^i}\max_{\boldsymbol{z}\in\mathcal{X}^i}\|\boldsymbol{z}\|_{(\boldsymbol{V}_0^i)^{-1}}^2 \leq \frac{8d^2}{|\mathcal{T}_0|},$$

$$\mathbb{E}_{\mathcal{X}^i\sim\mathcal{D}_m^i}\max_{\boldsymbol{z}\in\mathcal{X}^i}\|\breve{\boldsymbol{z}}\|_{(\boldsymbol{H}_m^i)^{-1}} \leq \sqrt{\frac{8d\log d}{|\mathcal{T}_m|}}.$$

Also, since the Distributional Optimal Design samples according to a G-Optimal Design with half probability (see Lemma A.14, Sawarni et al. (2024)), we have

$$\mathbb{E}_{\mathcal{X}^i\sim\mathcal{D}_m^i}\max_{\boldsymbol{z}\in\mathcal{X}^i}\|\breve{\boldsymbol{z}}\|_{(\boldsymbol{H}_m^i)^{-1}}^2 \leq \frac{16d^2}{|\mathcal{T}_m|}.$$

Using the same steps as Lemma A.10 to bound the first term, and substituting the optimal design bounds above results in:

$$\sum_{t \in \mathcal{T}_m} \mathop{\mathbb{E}}_{t,m} \sqrt{\dot{\mu}(\boldsymbol{x}_{t,\star}^\top \boldsymbol{\theta}^\star)} \left( \sum_{i=1}^N \max_{\boldsymbol{z} \in \mathcal{X}_t^i} \|\check{\boldsymbol{z}}\|_{(\boldsymbol{H}_{m-1}^i)^{-1}} \sum_{i=1}^N \|\boldsymbol{b}_t^i\|_{(\boldsymbol{V}_0^i)^{-1}} \right) \le \frac{12 N^2 d^2 \sqrt{L_\mu} |\mathcal{T}_m|}{\sqrt{|\mathcal{T}_{m-1}||\mathcal{T}_0|}}.$$

The second term can be upper-bounded in two different ways. First, using the Cauchy-Schwarz inequality allows us to bound the second term with respect to the average reward sensitivity of the optimal slates, similar to Lemma A.10, i.e,

$$\sum_{t \in \mathcal{T}_m} \mathop{\mathbb{E}}_{t,m} \sqrt{\dot{\mu}(\boldsymbol{x}_{t,\star}^\top \boldsymbol{\theta}^\star)} \sum_{i=1}^N \max_{\boldsymbol{z} \in \mathcal{X}_t^i} \|\check{\boldsymbol{z}}\|_{(\boldsymbol{H}_{m-1}^i)^{-1}} \le \sqrt{\mathop{\mathbb{E}}_{\{\mathcal{X}^i \sim \mathcal{D}^i\}_{i=1}^N} \dot{\mu}(\boldsymbol{x}_\star^\top \boldsymbol{\theta}^\star)} \sum_{t \in \mathcal{T}_m} \sqrt{\mathop{\mathbb{E}}_{t,m} \left( \sum_{i=1}^N \max_{\boldsymbol{z} \in \mathcal{X}_t^i} \|\check{\boldsymbol{z}}\|_{(\boldsymbol{H}_{m-1}^i)^{-1}} \right)^2}$$

$$\le \sqrt{\mathop{\mathbb{E}}_{\{\mathcal{X}^i \sim \mathcal{D}^i\}_{i=1}^N} \dot{\mu}(\boldsymbol{x}_\star^\top \boldsymbol{\theta}^\star)} \sqrt{\frac{16 d^2 N |\mathcal{T}_m|^2}{|\mathcal{T}_{m-1}|} + \frac{4 L_\mu N |\mathcal{T}_m|}{\rho}}.$$

where $\boldsymbol{x}_\star = \arg\max_{\boldsymbol{x} \in \mathcal{X}} \boldsymbol{x}^\top \boldsymbol{\theta}^\star$ and $\mathcal{X} = \mathcal{X}^1 \times \ldots \times \mathcal{X}^N$. Here, the second inequality follows from Lemma A.10.

On the other hand, to leverage the advantage of Distributional Optimal design, we can avoid using the Cauchy-Schwartz inequality, resulting in

$$\sum_{t \in \mathcal{T}_m} \mathop{\mathbb{E}}_{t,m} \sqrt{\dot{\mu}(\boldsymbol{x}_{t,\star}^\top \boldsymbol{\theta}^\star)} \sum_{i=1}^N \max_{\boldsymbol{z} \in \mathcal{X}_t^i} \|\check{\boldsymbol{z}}\|_{(\boldsymbol{H}_{m-1}^i)^{-1}} \le \sqrt{\max_{\{\mathcal{X}^i \sim \mathcal{D}^i\}_{i=1}^N} \dot{\mu}(\boldsymbol{x}_\star^\top \boldsymbol{\theta}^\star)} \sum_{t \in \mathcal{T}_m} \sum_{i=1}^N \mathop{\mathbb{E}}_{\mathcal{X}_t^i \sim \mathcal{D}_m^i} \max_{\boldsymbol{z} \in \mathcal{X}_t^i} \|\check{\boldsymbol{z}}\|_{(\boldsymbol{H}_{m-1}^i)^{-1}}$$

$$\le \sqrt{\max_{\{\mathcal{X}^i \sim \mathcal{D}^i\}_{i=1}^N} \dot{\mu}(\boldsymbol{x}_\star^\top \boldsymbol{\theta}^\star)} \sum_{t \in \mathcal{T}_m} \sum_{i=1}^N \mathop{\mathbb{E}}_{\mathcal{X}_t^i \sim \mathcal{D}_{m-1}^i} \max_{\boldsymbol{z} \in \mathcal{X}_t^i} \|\check{\boldsymbol{z}}\|_{(\boldsymbol{H}_{m-1}^i)^{-1}}$$

$$\le 4N \sqrt{\max_{\{\mathcal{X}^i \sim \mathcal{D}^i\}_{i=1}^N} \dot{\mu}(\boldsymbol{x}_\star^\top \boldsymbol{\theta}^\star) \cdot d \log d} \frac{|\mathcal{T}_m|}{\sqrt{|\mathcal{T}_{m-1}|}}.$$

where the second inequality follows from Lemma A.8 and the final inequality follows from the optimal design bound given above.

Using Lemma A.9 and assembling all the bounds, we get that

$$\sum_{t \in \mathcal{T}_m} \mathop{\mathbb{E}}_{t,m} |\mu(\boldsymbol{x}_{t,\star}^\top \boldsymbol{\theta}^\star) - \mu(\boldsymbol{x}_t^\top \boldsymbol{\theta}^\star)| \le \frac{320 e^{3S} \gamma^2 N^2 d^2 \sqrt{\kappa L_\mu}}{S} \frac{|\mathcal{T}_m|}{\sqrt{|\mathcal{T}_{m-1}||\mathcal{T}_0|}} +$$

$$8\gamma \min \left\{ \sqrt{\mathop{\mathbb{E}}_{\{\mathcal{X}^i \sim \mathcal{D}^i\}_{i=1}^N} \dot{\mu}(\boldsymbol{x}_\star^\top \boldsymbol{\theta}^\star)} \sqrt{\frac{16 d^2 N |\mathcal{T}_m|^2}{|\mathcal{T}_{m-1}|} + \frac{4 L_\mu N |\mathcal{T}_m|}{\rho}}, 4N \sqrt{\max_{\{\mathcal{X}^i \sim \mathcal{D}^i\}_{i=1}^N} \dot{\mu}(\boldsymbol{x}_\star^\top \boldsymbol{\theta}^\star) \cdot d \log d} \frac{|\mathcal{T}_m|}{\sqrt{|\mathcal{T}_{m-1}|}} \right\}.$$

$\square$

**Lemma B.4.** *(Theorem 5, Ruan et al. (2021)) Let $\pi$ denote the distributional optimal design that has been learnt using $N$ i.i.d samples and let*

$$\boldsymbol{W} = \mathop{\mathbb{E}}_{\mathcal{X} \sim \mathcal{D}} \mathop{\mathbb{E}}_{\boldsymbol{x} \sim \pi(\mathcal{X})} \boldsymbol{x}\boldsymbol{x}^\top.$$

*Then, we have that*

$$\Pr \left[ \mathop{\mathbb{E}}_{\mathcal{X} \sim \mathcal{D}} \max_{\boldsymbol{x} \in \mathcal{X}} \|\boldsymbol{x}\|_{\boldsymbol{W}^{-1}} \le O(d \log d) \right] \ge 1 - \delta,$$

*where $\delta = \exp(O(d^4 \log^2 d) - N d^{-12} \cdot 2^{-16})$.*

## C. Regret Analysis for `RS-SlateGLinCB`

In this section, we state and prove the regret bound for `RS-SlateGLinCB` (Algorithm 2).

### C.1. Notations

First, define the following scalars:

$$\lambda = \mathcal{O}\left(R\sqrt{NdS\log(ST\delta^{-1})}\right),$$

$$\gamma = \mathcal{O}\left(R^2 S^{3/2}\sqrt{Nd\log(ST\delta^{-1})}\right),$$

$$\beta = \mathcal{O}\left(R\sqrt{NdS\log(ST\delta^{-1})}\right).$$

Similar to Section A.1, define $\tilde{\boldsymbol{x}}^i = \boldsymbol{x}^i \otimes \boldsymbol{e}_i$ so that any slate $\boldsymbol{x} = (\boldsymbol{x}^1, \dots, \boldsymbol{x}^N)$ can be written as

$$\boldsymbol{x} = \sum_{i=1}^{N} \tilde{\boldsymbol{x}}^i.$$

Define the set of warm-up rounds as $\mathcal{T}_0$. Then, we can define the warm-up matrix $\boldsymbol{V}$ and the corresponding slot-level warm-up matrices $\boldsymbol{V}^i$ for all $i \in [N]$ as

1. $\boldsymbol{V} = \lambda \boldsymbol{I}_{Nd} + \sum_{t \in \mathcal{T}_0} \boldsymbol{x}_t \boldsymbol{x}_t^\top$.

2. $\boldsymbol{V}^i = \lambda \boldsymbol{I}_d + \sum_{t \in \mathcal{T}_0} \boldsymbol{x}_t^i \boldsymbol{x}_t^{i\top}$.

Define the set of indices $\mathcal{T}_{\neg 0} := [|\mathcal{T}_0| + 1, T]$ to be the set of all time rounds post warm-up. In particular, define the set $\mathcal{T}_{\neg 0}^{<t} := [|\mathcal{T}_0| + 1, t-1]$. For round $t \in \mathcal{T}_{\neg 0}$, define the Hessian of the GLM-MLE loss $\boldsymbol{H}_t^\star$ as

$$\boldsymbol{H}_t^\star = \lambda \boldsymbol{I}_{Nd} + \sum_{k \in \mathcal{T}_{\neg 0}^{\leq t}} \dot{\mu}(\boldsymbol{x}_k^\top \boldsymbol{\theta}^\star) \boldsymbol{x}_k \boldsymbol{x}_k^\top.$$

Since $\boldsymbol{\theta}^\star$ is unknown, we estimate the Hessian using a scaled design matrix $\boldsymbol{H}_t$, defined as

$$\boldsymbol{H}_t = \lambda \boldsymbol{I}_{Nd} + \sum_{k \in \mathcal{T}_{\neg 0}^{\leq t}} \dot{\mu}(\boldsymbol{x}_k^\top \widehat{\boldsymbol{\theta}}_0) e^{-1} \boldsymbol{x}_k \boldsymbol{x}_k^\top.$$

Also, for all $i \in [N]$, we define the slot-level scaled matrices $\boldsymbol{H}_t^i$ as

$$\boldsymbol{H}_t^i = \lambda \boldsymbol{I}_d + \sum_{k \in \mathcal{T}_{\neg 0}^{\leq t}} \dot{\mu}(\boldsymbol{x}_k^\top \widehat{\boldsymbol{\theta}}_0) e^{-1} \boldsymbol{x}_k^i \boldsymbol{x}_k^{i\top}.$$

For any time round $t \in \mathcal{T}_{\neg 0}$, define $s(t) \leq t$ to be the last time round where an estimate $\widehat{\boldsymbol{\theta}}_{s(t)}$ was computed. Hence, we have

$$\det \boldsymbol{H}_t < 2 \det \boldsymbol{H}_{s(t)}.$$

Equality is obtained if $s(t) = t$, which happens if $\det \boldsymbol{H}_t \geq 2 \det \boldsymbol{H}_{s(t-1)}$, leading to an update at round $t$.

Finally, similar to Section A.1, for any slot $i \in [N]$, item $\boldsymbol{z} \in \mathcal{X}_t^i$ and a prior batch $l \in [M]$, we define the scores $\text{UCB}^{i,l}(\boldsymbol{z})$ and $\text{LCB}^{i,l}(\boldsymbol{z})$ as

$$\text{UCB}^{i,l}(\boldsymbol{z}) = \begin{cases} \boldsymbol{z}^\top \widehat{\boldsymbol{\theta}}_0^i + 2\sqrt{\kappa}\gamma \|\boldsymbol{z}\|_{(\boldsymbol{V}_0^i)^{-1}} & l = 0, \\ \boldsymbol{z}^\top \widehat{\boldsymbol{\theta}}_l^i + 2\gamma \|\boldsymbol{z}\|_{(\boldsymbol{H}_l^i)^{-1}} & l \neq 0. \end{cases}$$

$$\text{LCB}^{i,l}(\boldsymbol{z}) = \begin{cases} \boldsymbol{z}^\top \widehat{\boldsymbol{\theta}}_0^i - 2\sqrt{\kappa}\gamma\|\boldsymbol{z}\|_{(\boldsymbol{V}_0^i)^{-1}} & l = 0, \\ \boldsymbol{z}^\top \widehat{\boldsymbol{\theta}}_l^i - 2\gamma\|\boldsymbol{z}\|_{(\boldsymbol{H}_l^i)^{-1}} & l \neq 0. \end{cases}$$

Finally, define the following quantities:

$$T(\neg 0) := \frac{(48L_\mu^2 + 8L_\mu N\rho)(N-1)^2}{3\rho^2} \log\left(\frac{2dNT}{\delta}\right) \quad \text{and} \quad T(0) := \frac{(48 + 8N\rho)(N-1)^2}{3\rho^2} \log\left(\frac{2dNT}{\delta}\right).$$

Unless otherwise mentioned, without loss of generality, we assume that all constants such as $S, T, R, N, d, \kappa$ and $L_\mu$ are greater than $1$.

## C.2. Regret Guarantee for `RS-SlateGLinCB`

Now, we restate the regret guarantee for `RS-SlateGLinCB`, given in Theorem 4.1), and provide a proof for the same.

**Theorem C.1.** *Let $R(T)$ denote the regret of `RS-SlateGLinCB` (Algorithm 2). If*

$$\sqrt{\frac{2dN}{\delta}} \frac{(48 + 8N\rho)(N-1)^2}{3\rho^2} \geq \frac{e}{2} \quad \text{and} \quad \frac{16R^6 S^{7/2} N^2 \kappa d}{\sqrt{\delta}\rho} \geq e$$

*and*

$$T \geq$$

$$T_0 := \max\left\{\frac{\delta}{2dN} \exp\left(-2W_{-1}\left(\frac{-3\rho^2\sqrt{\delta}}{2\sqrt{2dN}(48L_\mu^2 + 8NL_\mu\rho)(N-1)^2}\right)\right), \frac{\delta}{S} \exp\left(-2W_{-1}\left(\frac{-\sqrt{\delta}\rho}{32R^6 S^{7/2} N^2 \kappa d}\right)\right)\right\},$$

*where $W_{-1}$ is the decreasing branch of Lambert W function (see Lemma A.15), then,*

$$R(T) = \tilde{\mathcal{O}}\left(R\sqrt{T} + RS^{1/2}Nd\sqrt{\sum_{t \in T_{\neg 0}} \dot{\mu}(\boldsymbol{x}_{t,\star}^\top \boldsymbol{\theta}^\star)}\right).$$

*Proof.* At round $t \in [T]$, let $\boldsymbol{x}_{t,\star}$ be the optimal slate, i.e,

$$\boldsymbol{x}_{t,\star} = \arg\max_{\boldsymbol{x} \in \mathcal{X}_t} \boldsymbol{x}^\top \boldsymbol{\theta}^\star.$$

Then, the regret of Algorithm 2 can be written as

$$R(T) = \sum_{t \in [T]} \mu(\boldsymbol{x}_{t,\star}^\top \boldsymbol{\theta}^\star) - \mu(\boldsymbol{x}_t^\top \boldsymbol{\theta}^\star).$$

In Lemma C.8, we bound this exact quantity. However, we require that $|\mathcal{T}_0| \geq \max\{T(0), 8\gamma^2 R^2 \rho^{-1} \kappa N\}$. We set $|\mathcal{T}_0| = \lfloor \sqrt{T} \rfloor$ resulting in the inequality:

$$\sqrt{T} \geq \max\left\{\frac{(48 + 8N\rho)(N-1)^2}{3\rho^2} \log\left(\frac{2dNT}{\delta}\right), 8\gamma^2 R^2 \rho^{-1} \kappa N\right\}.$$

Since,

$$\sqrt{\frac{2dN}{\delta}} \frac{(48 + 8N\rho)(N-1)^2}{3\rho^2} \geq \frac{e}{2} \quad \text{and} \quad \frac{16R^6 S^{7/2} N^2 \kappa d}{\sqrt{\delta}\rho} \geq e,$$

and also

$$T \geq 2 \geq \frac{\delta}{S} \exp\left(\frac{1}{2}\right), \quad \text{and} \quad T \geq 2 \geq \frac{\delta}{2dN} \exp\left(\frac{1}{2}\right),$$

we can use Lemma A.15 and the definition of $\gamma$ (Section C.1) to get

$$T \geq \max \left\{ \frac{\delta}{2dN} \exp\left( -2W_{-1}\left( \frac{-3\rho^2\sqrt{\delta}}{2\sqrt{2dN}(48 + 8N\rho)(N-1)^2} \right) \right), \frac{\delta}{S} \exp\left( -2W_{-1}\left( \frac{-\sqrt{\delta}\rho}{32R^6 S^{7/2} N^2 \kappa d} \right) \right) \right\}.$$

From Lemma C.8, we also have $|\mathcal{T}_{\neg 0}^{\leq t}| \geq T(\neg 0)$. Now, let $t'$ be such that $|\mathcal{T}_{\neg 0}^{\leq t'}| = \lfloor\sqrt{T}\rfloor$. Such a $t'$ exists because $T \geq 2$ and hence, $T \geq |\mathcal{T}_0| + |\mathcal{T}_{\neg 0}^{\leq t'}| = 2\lfloor\sqrt{T}\rfloor$. Let $|\mathcal{T}_{\neg 0}^{\leq t'}| \geq T(\neg 0)$, then, we have

$$\sqrt{T} \geq \frac{(48L_\mu^2 + 8L_\mu N\rho)(N-1)^2}{3\rho^2} \log\left( \frac{2dNT}{\delta} \right)$$

resulting in the bound (using Lemma A.15)

$$T \geq \frac{\delta}{2dN} \exp\left( -2W\left( \frac{-3\rho^2\sqrt{\delta}}{2\sqrt{2dN}(48L_\mu^2 + 8NL_\mu\rho)(N-1)^2} \right) \right).$$

Using the fact that both $\exp(-x)$ and $W_{-1}(x)$ are decreasing (in the domain $(-e^{-1}, 0)$ Lemma A.15) and $L_\mu \geq 1$, we get the final bound on $T$ as

$$T \geq$$

$$T_0 := \max \left\{ \frac{\delta}{2dN} \exp\left( -2W_{-1}\left( \frac{-3\rho^2\sqrt{\delta}}{2\sqrt{2dN}(48L_\mu^2 + 8NL_\mu\rho)(N-1)^2} \right) \right), \frac{\delta}{S} \exp\left( -2W_{-1}\left( \frac{-\sqrt{\delta}\rho}{32R^6 S^{7/2} N^2 \kappa d} \right) \right) \right\}.$$

Now, assuming $T \geq T_0$, we can split $R(T)$ as

$$R(T) = \sum_{t \in \mathcal{T}_0} \mu(\boldsymbol{x}_{t,\star}^\top \boldsymbol{\theta}^\star) - \mu(\boldsymbol{x}_t^\top \boldsymbol{\theta}^\star) + \sum_{t \in \mathcal{T}_{\neg 0}^{\leq t'}} \mu(\boldsymbol{x}_{t,\star}^\top \boldsymbol{\theta}^\star) - \mu(\boldsymbol{x}_t^\top \boldsymbol{\theta}^\star) + \sum_{t=t'}^T \mu(\boldsymbol{x}_{t,\star}^\top \boldsymbol{\theta}^\star) - \mu(\boldsymbol{x}_t^\top \boldsymbol{\theta}^\star)$$

$$\leq 2R\sqrt{T} + \sum_{t=t'}^T \mu(\boldsymbol{x}_{t,\star}^\top \boldsymbol{\theta}^\star) - \mu(\boldsymbol{x}_t^\top \boldsymbol{\theta}^\star)$$

where we use a trivial regret bound of $R$ for $t \in \mathcal{T}_0 \cup \mathcal{T}_{\neg 0}^{\leq t'}$ alongside the fact that $|\mathcal{T}_0| = |\mathcal{T}_{\neg 0}^{\leq t'}| = \lfloor\sqrt{T}\rfloor$.

Now, using Lemma C.8, we have

$$\sum_{t=t'}^T \mu(\boldsymbol{x}_{t,\star}^\top \boldsymbol{\theta}^\star) - \mu(\boldsymbol{x}_t^\top \boldsymbol{\theta}^\star) \leq 4\sqrt{2}e^5\beta \sum_{t=t'}^T \sqrt{\dot{\mu}(\boldsymbol{x}_{t,\star}^\top \boldsymbol{\theta}^\star) \cdot e^{-1}\dot{\mu}(\boldsymbol{x}_t^\top \widehat{\boldsymbol{\theta}}_0)} \sum_{i=1}^N \|\boldsymbol{x}_t^i\|_{(\boldsymbol{H}_t^i)^{-1}}$$

$$\leq 4\sqrt{2}e^5\beta \sqrt{ \sum_{t=t'}^T \dot{\mu}(\boldsymbol{x}_{t,\star}^\top \boldsymbol{\theta}^\star) \cdot \sum_{t=t'}^T \frac{\dot{\mu}(\boldsymbol{x}_t^\top \widehat{\boldsymbol{\theta}}_0)}{e} \left( \sum_{i=1}^N \|\boldsymbol{x}_t^i\|_{(\boldsymbol{H}_t^i)^{-1}} \right)^2 }$$

$$\leq 4\sqrt{2}e^5\beta \sqrt{ \sum_{t=t'}^T \dot{\mu}(\boldsymbol{x}_{t,\star}^\top \boldsymbol{\theta}^\star) \cdot \left( \underbrace{\sum_{i=1}^N \sum_{t=t'}^T \left\| \sqrt{\frac{\dot{\mu}(\boldsymbol{x}_t^\top \widehat{\boldsymbol{\theta}}_0)}{e}} \boldsymbol{x}_t^i \right\|_{(\boldsymbol{H}_t^i)^{-1}}^2}_{\text{Term A}} + \underbrace{\sum_{i=1}^N \sum_{\substack{j=1 \\ j\neq i}}^N \sum_{t=t'}^T \frac{\dot{\mu}(\boldsymbol{x}_t^\top \widehat{\boldsymbol{\theta}}_0)}{e} \|\boldsymbol{x}_t^i\|_{(\boldsymbol{H}_t^i)^{-1}} \|\boldsymbol{x}_t^j\|_{(\boldsymbol{H}_t^j)^{-1}}}_{\text{Term B}} \right) }$$

Bounding Term A, we get

$$\sum_{i=1}^N \sum_{t=t'}^T \left\| \sqrt{\frac{\dot{\mu}(\boldsymbol{x}_t^\top \widehat{\boldsymbol{\theta}}_0)}{e}} \boldsymbol{x}_t^i \right\|_{(\boldsymbol{H}_t^i)^{-1}}^2 \leq 2Nd \log\left( 1 + \frac{L_\mu \cdot t}{e \cdot N\lambda d} \right) \leq 2Nd \log T.$$

Here, we use Lemma C.12 with the vectors $\breve{\boldsymbol{x}}_t^i := \sqrt{\frac{\dot{\mu}(\boldsymbol{x}_t^\top \widehat{\boldsymbol{\theta}}_0)}{e}} \boldsymbol{x}_t^i$, resulting in $\|\breve{\boldsymbol{x}}_t^i\| \leq \sqrt{L_\mu e^{-1} \cdot N^{-1}}$.

Bounding Term B, we get

$$\sum_{i=1}^N \sum_{\substack{j=1 \\ j \neq i}}^N \sum_{t=t'}^T \frac{\dot{\mu}(\boldsymbol{x}_t^\top \widehat{\boldsymbol{\theta}}_0)}{e} \|\boldsymbol{x}_t^i\|_{(\boldsymbol{H}_t^i)^{-1}} \|\boldsymbol{x}_t^j\|_{(\boldsymbol{H}_t^j)^{-1}} \leq \sum_{i=1}^N \sum_{\substack{j=1 \\ j \neq i}}^N \sum_{t=t'}^T L_\mu \frac{\|\boldsymbol{x}_t^i\| \|\boldsymbol{x}_t^j\|}{\sqrt{\lambda_{\min}(\boldsymbol{H}_t^i) \lambda_{\min}(\boldsymbol{H}_t^j)}},$$

where the inequality uses Rayleigh's quotient. Since, for all $t \in [t', T]$, $|\mathcal{T}_{\neg 0}^{<t}| \geq |\mathcal{T}_{\neg 0}^{<t'}| \geq T(\neg 0)$, using Lemma C.10, for all $i \in [N]$

$$\lambda_{\min}(\boldsymbol{H}_t^i) \geq \lambda + \frac{\rho |\mathcal{T}_{\neg 0}^{<t}|}{2}.$$

Substituting this back, we get

$$\sum_{i=1}^N \sum_{\substack{j=1 \\ j \neq i}}^N \sum_{t=t'}^T \frac{\dot{\mu}(\boldsymbol{x}_t^\top \widehat{\boldsymbol{\theta}}_0)}{e} \|\boldsymbol{x}_t^i\|_{(\boldsymbol{H}_t^i)^{-1}} \|\boldsymbol{x}_t^j\|_{(\boldsymbol{H}_t^j)^{-1}} \leq \sum_{i=1}^N \sum_{\substack{j=1 \\ j \neq i}}^N \sum_{t=t'}^T L_\mu \frac{\|\boldsymbol{x}_t^i\| \|\boldsymbol{x}_t^j\|}{\lambda + 0.5\rho |\mathcal{T}_{\neg 0}^{<t}|}$$

$$\leq \sum_{i=1}^N \sum_{\substack{j=1 \\ j \neq i}}^N \frac{2L_\mu}{\rho} \|\boldsymbol{x}_t^i\| \|\boldsymbol{x}_t^j\| \left( \sum_{s \in [|\mathcal{T}_{\neg 0}|]} \frac{1}{s} \right).$$

Using the sum of the Harmonic series, alongside the fact that $\forall i \in [N], \|\boldsymbol{x}^i\| \leq \sqrt{N^{-1}}$, we get a bound on Term B as

$$\sum_{i=1}^N \sum_{\substack{j=1 \\ j \neq i}}^N \sum_{t=t'}^T \frac{\dot{\mu}(\boldsymbol{x}_t^\top \widehat{\boldsymbol{\theta}}_0)}{e} \|\boldsymbol{x}_t^i\|_{(\boldsymbol{H}_t^i)^{-1}} \|\boldsymbol{x}_t^j\|_{(\boldsymbol{H}_t^j)^{-1}} \leq 2N L_\mu \rho^{-1} \log T.$$

Combining all the bounds, we get

$$R(T) \leq 2R\sqrt{T} + 4\sqrt{2} e^5 \beta \sqrt{\left( \sum_{t=t'}^T \dot{\mu}(\boldsymbol{x}_{t,\star}^\top \theta^\star) \right) \cdot (2Nd + 2Nl_\mu \rho^{-1}) \log T}.$$

Substituting the value of $\beta = \mathcal{O}\left( R\sqrt{NdS \log(ST\delta^{-1})} \right)$ from Section C.1, we get

$$R(T) = \tilde{\mathcal{O}} \left( R\sqrt{T} + RS^{1/2} Nd \sqrt{\sum_{t \in T_{\neg 0}} \dot{\mu}(\boldsymbol{x}_{t,\star}^\top \theta^\star)} \right).$$

$\square$

## C.3. Supporting Lemmas for Theorem C.1

**Lemma C.2.** *(Lemma B.3 and B.7, (Sawarni et al., 2024)) Let $\widehat{\boldsymbol{\theta}}_0$ be the MLE estimate of $\theta^\star$ learned using $\{\boldsymbol{x}_t, r_t\}_{t \in \mathcal{T}_0}$, while $\widehat{\boldsymbol{\theta}}_{s(t)}$ be the MLE estimate of $\theta^\star$ learned using $\{\boldsymbol{x}_t, r_t\}_{t \in \mathcal{T}_{\neg 0}^{<s(t)}}$. Let $\boldsymbol{H}_t^\star$ be defined as in Section C.1. Also, define*

$$\boldsymbol{H}_0^\star = \lambda \boldsymbol{I}_{Nd} + \sum_{t \in \mathcal{T}_0} \dot{\mu}(\boldsymbol{x}_t^\top \theta^\star) \boldsymbol{x}_t \boldsymbol{x}_t^\top.$$

*Then, we have*

$$\|\theta^\star - \widehat{\boldsymbol{\theta}}_0\|_{\boldsymbol{H}_0^\star} \leq \gamma \quad and \quad \|\theta^\star - \widehat{\boldsymbol{\theta}}_{s(t)}\|_{\boldsymbol{H}_{s(t)}^\star} \leq \beta.$$

**Lemma C.3.** *Let $t \in \mathcal{T}_{\neg 0}$. Also, let $|\mathcal{T}_0| \geq \max\{T(0), 8\gamma^2 R^2 \rho^{-1} \kappa N\}$, then, we have*

$$\left| \boldsymbol{x}_t^\top (\boldsymbol{\theta}^\star - \widehat{\boldsymbol{\theta}}_0) \right| \leq \frac{1}{R}.$$

*Proof.* For the sake of this proof, define

$$\boldsymbol{H}_0^\star = \lambda \boldsymbol{I}_{Nd} + \sum_{t \in \mathcal{T}_0} \dot{\mu}(\boldsymbol{x}_t^\top \boldsymbol{\theta}^\star) \boldsymbol{x}_t \boldsymbol{x}_t^\top.$$

Then, we have

$$
\begin{aligned}
\left| \boldsymbol{x}^\top (\boldsymbol{\theta}^\star - \widehat{\boldsymbol{\theta}}_0) \right| &\leq \|\boldsymbol{x}\|_{(\boldsymbol{H}_0^\star)^{-1}} \|\boldsymbol{\theta}^\star - \widehat{\boldsymbol{\theta}}_0\|_{\boldsymbol{H}_0^\star} \\
&\leq \gamma \sqrt{\kappa} \|\boldsymbol{x}\|_{\boldsymbol{V}^{-1}} \\
&\leq 2\gamma \sqrt{\kappa} \sum_{i=1}^N \|\boldsymbol{x}^i\|_{(\boldsymbol{V}^i)^{-1}} \\
&\leq 2\gamma \sqrt{\kappa} \sum_{i=1}^N \frac{\|\boldsymbol{x}^i\|}{\sqrt{\lambda_{\min}(\boldsymbol{V}^i)}}
\end{aligned}
$$

where the first inequality uses the Cauchy-Schwarz inequality, the second inequality follows from Lemma C.2 and Lemma A.3, the third inequality follows from Lemma C.11, and the final inequality follows from Rayleigh's quotient.

Now, from Lemma C.11, we have that

$$\lambda_{\min}(\boldsymbol{V}^i) \geq \lambda + \frac{\rho |\mathcal{T}_0|}{2}.$$

Thus, using the fact that $\|\boldsymbol{x}^i\| \leq \sqrt{N^{-1}}$, we have

$$\left| \boldsymbol{x}^\top (\boldsymbol{\theta}^\star - \widehat{\boldsymbol{\theta}}_0) \right| \leq 2\gamma \sqrt{\frac{\kappa N}{\lambda + 0.5\rho |\mathcal{T}_0|}}.$$

Finally, using the fact that $|\mathcal{T}_0| \geq 8\gamma^2 R^2 \rho^{-1} \kappa N$, we get that

$$\left| \boldsymbol{x}^\top (\boldsymbol{\theta}^\star - \widehat{\boldsymbol{\theta}}_0) \right| \leq \frac{1}{R}.$$

$\square$

**Lemma C.4.** *For some round $t \in \mathcal{T}_{\neg 0}$, let $\boldsymbol{H}_t^\star$ and $\boldsymbol{H}_t$ be defined as in Section C.1. If $|\mathcal{T}_0| \geq \max\{T(0), 8\gamma^2 R^2 \rho^{-1} \kappa N\}$, then, we have*

$$\boldsymbol{H}_t \preceq \boldsymbol{H}_t^\star.$$

*Proof.* Using the self-concordance property of GLMs, we have that

$$\exp\left(-R\left|\boldsymbol{x}^\top(\boldsymbol{\theta}^\star - \widehat{\boldsymbol{\theta}}_0)\right|\right) \cdot \dot{\mu}(\boldsymbol{x}^\top \widehat{\boldsymbol{\theta}}_0) \preceq \dot{\mu}(\boldsymbol{x}^\top \boldsymbol{\theta}^\star) \preceq \exp\left(R\left|\boldsymbol{x}^\top(\boldsymbol{\theta}^\star - \widehat{\boldsymbol{\theta}}_0)\right|\right) \cdot \dot{\mu}(\boldsymbol{x}^\top \widehat{\boldsymbol{\theta}}_0).$$

From Lemma C.3, we get

$$e^{-1} \cdot \dot{\mu}(\boldsymbol{x}^\top \widehat{\boldsymbol{\theta}}_0) \preceq \dot{\mu}(\boldsymbol{x}^\top \boldsymbol{\theta}^\star) \preceq e \cdot \dot{\mu}(\boldsymbol{x}^\top \widehat{\boldsymbol{\theta}}_0).$$

Hence, we can write,

$$\boldsymbol{H}_t^\star = \lambda \boldsymbol{I}_{Nd} + \sum_{s \in \mathcal{T}_{\neg 0}^{<t}} \dot{\mu}(\boldsymbol{x}_s^\top \boldsymbol{\theta}^\star) \boldsymbol{x}_s \boldsymbol{x}_s^\top \succeq \lambda \boldsymbol{I}_{Nd} + \sum_{s \in \mathcal{T}_{\neg 0}^{<t}} \dot{\mu}(\boldsymbol{x}_s^\top \widehat{\boldsymbol{\theta}}_0) e^{-1} \boldsymbol{x}_s \boldsymbol{x}_s^\top = \boldsymbol{H}_t.$$

$\square$

**Lemma C.5.** *For some round $t \in \mathcal{T}_{\neg 0}$, let $s(t) \leq t$ be the most recent time round at which the policy was updated. Then, we have that*

$$\|x\|^2_{H_{s(t)}^{-1}} \leq 2\|x\|^2_{H_t^{-1}}.$$

*Proof.* First, from the definition of $s(t)$, we have that $s(t) \leq t$. Thus, we have that

$$|\mathcal{T}_{\neg 0}^{<s(t)}| \leq |\mathcal{T}_{\neg 0}^{\leq t}| \implies H_{s(t)} \preceq H_t \implies H_{s(t)}^{-1} \succeq H_t^{-1}.$$

Hence, applying Lemma C.13 with $A = H_{s(t)}^{-1}$ and $B = H_t^{-1}$, we get that

$$\|x\|^2_{H_{s(t)}^{-1}} \leq \|x\|^2_{H_t^{-1}} \cdot \frac{\det H_{s(t)}^{-1}}{\det H_t^{-1}}.$$

Using the fact that $\det H_t \leq 2 \det H_{s(t)}$ (Section C.1), we get that

$$\|x\|^2_{H_{s(t)}^{-1}} \leq \|x\|^2_{H_t^{-1}} \cdot \frac{\det H_t}{\det H_{s(t)}} \leq 2\|x\|^2_{H_t^{-1}}.$$

$\square$

**Lemma C.6.** *Let $t \in \mathcal{T}_{\neg 0}$. Define the optimal slate $x_{t,\star}$ as*

$$x_{t,\star} = \arg\max_{x \in \mathcal{X}_t} x^\top \theta^\star.$$

*If $|\mathcal{T}_0| \geq T(0)$, then the optimal slate never gets eliminated.*

*Proof.* First, it is easy to see that all the components of the optimal slate, i.e. $x_{t,\star}^i$ are also optimal w.r.t $\theta^{\star i}$. In other words, for all $i \in [N]$, we have

$$x_{t,\star}^i = \arg\max_{z \in \mathcal{X}_t^i} \tilde{z}^\top \theta^\star.$$

Fix $i \in [N]$. Then, for some arbitrary $z \in \mathcal{X}_t^i$, we have that

$$
\begin{aligned}
0 &\leq \left(\tilde{x}_{t,\star}^i - \tilde{z}\right)^\top \theta^\star \\
&= \left(\tilde{x}_{t,\star}^i - \tilde{z}\right)^\top \left(\theta^\star - \widehat{\theta}_0\right) + \left(\tilde{x}_{t,\star}^i - \tilde{z}\right)^\top \widehat{\theta}_0 \\
&\leq 2\sqrt{\kappa}\gamma\|x_{t,\star}^i\|_{(V^i)^{-1}} + 2\sqrt{\kappa}\gamma\|z\|_{(V^i)^{-1}} + \left(\tilde{x}_{t,\star}^i - \tilde{z}\right)^\top \widehat{\theta}_0 \\
&= \mathrm{UCB}^{i,0}(x_{t,\star}^i) - \mathrm{LCB}^{i,0}(z)
\end{aligned}
$$

where the second inequality follows from Lemma A.3. Since this is true $\forall z \in \mathcal{X}_t^i$, we get

$$\mathrm{UCB}^{i,0}(x_{t,\star}^i) \geq \max_{z \in \mathcal{X}_t^i} \mathrm{LCB}^{i,0}(z)$$

or in other words,

$$\mathrm{UCB}^{i,0}(x_{t,\star}^i) - \max_{z \in \mathcal{X}_t^i} \mathrm{LCB}^{i,0}(z) \geq 0.$$

Since this holds for a fixed but arbitrary $i \in [N]$, the above inequality holds for all $i \in [N]$. Thus, the components of the optimal slate, and hence, the optimal slate never get eliminated. $\square$

**Lemma C.7.** *Let $t \in \mathcal{T}_{\neg 0}$. Let $x, y \in \mathcal{X}_t$ be two slates which do not get eliminated. If $|\mathcal{T}_0| \geq \max\{T(0), 8\gamma^2 R^2 \rho^{-1} \kappa N\}$, then, we have*

$$\left|(x - y)^\top \widehat{\theta}_0\right| \leq \frac{2}{R}.$$

*Proof.* Using the triangle inequality, we can write

$$\left|(\boldsymbol{x} - \boldsymbol{y})^{\top}\widehat{\boldsymbol{\theta}}_0\right| \leq \sum_{i=1}^{N}\left|(\boldsymbol{x}^i - \boldsymbol{y}^i)^{\top}\widehat{\boldsymbol{\theta}}_0^i\right|.$$

Now, since both $\boldsymbol{x}$ and $\boldsymbol{y}$ survive the elimination, their respective components $\boldsymbol{x}^i$ and $\boldsymbol{y}^i$ for all $i \in [N]$ also do not get eliminated. Thus, for a fixed $i \in [N]$, we have

$$\mathrm{UCB}^{i,0}(\boldsymbol{x}^i) \geq \max_{\boldsymbol{z} \in \mathcal{X}_t^i} \mathrm{LCB}^{i,0}(\boldsymbol{z}^i) \geq \mathrm{LCB}^{i,0}(\boldsymbol{y}^i).$$

Using the definitions of $\mathrm{UCB}^{i,0}(\boldsymbol{z})$ and $\mathrm{LCB}^{i,0}(\boldsymbol{z})$ (Section C.1), we get

$$(\boldsymbol{y}^i - \boldsymbol{x}^i)^{\top}\widehat{\boldsymbol{\theta}}_0^i \leq 2\sqrt{\kappa}\gamma\|\boldsymbol{x}^i\|_{(\boldsymbol{V}_0^i)^{-1}} + 2\sqrt{\kappa}\gamma\|\boldsymbol{y}^i\|_{(\boldsymbol{V}_0^i)^{-1}}.$$

A symmetric argument gives us

$$(\boldsymbol{x}^i - \boldsymbol{y}^i)^{\top}\widehat{\boldsymbol{\theta}}_0^i \leq 2\sqrt{\kappa}\gamma\|\boldsymbol{x}^i\|_{(\boldsymbol{V}_0^i)^{-1}} + 2\sqrt{\kappa}\gamma\|\boldsymbol{y}^i\|_{(\boldsymbol{V}_0^i)^{-1}}.$$

Thus, combining both the inequalities, we get

$$\left|(\boldsymbol{x}^i - \boldsymbol{y}^i)^{\top}\widehat{\boldsymbol{\theta}}_0^i\right| \leq 4\sqrt{\kappa}\gamma\max_{\boldsymbol{z} \in \mathcal{X}_t^i}\|\boldsymbol{z}\|_{(\boldsymbol{V}_0^i)^{-1}}$$

$$\leq 4\gamma\sqrt{\kappa}\sum_{i=1}^{N}\max_{z \in \mathcal{X}_t^i}\frac{\|\boldsymbol{z}\|}{\sqrt{\lambda_{\min}(\boldsymbol{V}^i)}}$$

where the final inequality follows from Rayleigh's quotient. Now, from Lemma C.11, we have that

$$\lambda_{\min}(\boldsymbol{V}^i) \geq \lambda + \frac{\rho|\mathcal{T}_0|}{2}.$$

Thus, using the fact that $\|\boldsymbol{z}\| \leq \sqrt{N^{-1}}$, we have

$$\left|(\boldsymbol{x}^i - \boldsymbol{y}^i)^{\top}\widehat{\boldsymbol{\theta}}_0^i\right| \leq 4\gamma\sqrt{\frac{\kappa N}{\lambda + 0.5\rho|\mathcal{T}_0|}}.$$

Finally, using the fact that $|\mathcal{T}_0| \geq 8\gamma^2 R^2 \rho^{-1}\kappa N$, we get that

$$\left|(\boldsymbol{x}^i - \boldsymbol{y}^i)^{\top}\widehat{\boldsymbol{\theta}}_0^i\right| \leq \frac{2}{R}.$$

$\square$

**Lemma C.8.** *Let $t \in \mathcal{T}_{\neg 0}$. Define the optimal slate $\boldsymbol{x}_{t,\star}$ as*

$$\boldsymbol{x}_{t,\star} = \arg\max_{\boldsymbol{x} \in \mathcal{X}_t} \boldsymbol{x}^{\top}\boldsymbol{\theta}^{\star}.$$

*Let $|\mathcal{T}_0| \geq \max\{T(0), 8\gamma^2 R^2 \rho^{-1}\kappa N\}$ and $|\mathcal{T}_{\neg 0}^{<t}| \geq T(\neg 0)$, then*

$$\mu(\boldsymbol{x}_{t,\star}^{\top}\boldsymbol{\theta}^{\star}) - \mu(\boldsymbol{x}_t^{\top}\boldsymbol{\theta}^{\star}) \leq 4\sqrt{2}e^5\beta\sqrt{\dot{\mu}(\boldsymbol{x}_{t,\star}^{\top}\boldsymbol{\theta}^{\star}) \cdot e^{-1}\dot{\mu}(\boldsymbol{x}_t^{\top}\widehat{\boldsymbol{\theta}}_0)}\sum_{i=1}^{N}\|\boldsymbol{x}_t^i\|_{(\boldsymbol{H}_t^i)^{-1}}.$$

*Proof.* Using a first-order Taylor Series expansion, for some $z_t \in [\boldsymbol{x}_t^\top \boldsymbol{\theta}^\star, \boldsymbol{x}_{t,\star}^\top \boldsymbol{\theta}^\star]$, we get

$$
\begin{aligned}
\mu(\boldsymbol{x}_{t,\star}^\top \boldsymbol{\theta}^\star) - \mu(\boldsymbol{x}_t^\top \boldsymbol{\theta}^\star) &\leq \dot{\mu}(z_t) \left( \boldsymbol{x}_{t,\star}^\top \boldsymbol{\theta}^\star - \boldsymbol{x}_t^\top \boldsymbol{\theta}^\star \right) \\
&\leq \dot{\mu}(z_t) \left[ \left| \boldsymbol{x}_{t,\star}^\top (\boldsymbol{\theta}^\star - \widehat{\boldsymbol{\theta}}_{s(t)}) \right| + \left| \boldsymbol{x}_t^\top (\boldsymbol{\theta}^\star - \widehat{\boldsymbol{\theta}}_{s(t)}) \right| + (\boldsymbol{x}_{t,\star} - \boldsymbol{x}_t)^\top \widehat{\boldsymbol{\theta}}_{s(t)} \right] \\
&\leq \dot{\mu}(z_t) \left[ \beta \|\boldsymbol{x}_{t,\star}\|_{\boldsymbol{H}_{s(t)}^{-1}} + \beta \|\boldsymbol{x}_t\|_{\boldsymbol{H}_{s(t)}^{-1}} + (\boldsymbol{x}_{t,\star} - \boldsymbol{x}_t)^\top \widehat{\boldsymbol{\theta}}_{s(t)} \right] \\
&\leq \dot{\mu}(z_t) \left[ \sqrt{2}\beta \|\boldsymbol{x}_{t,\star}\|_{\boldsymbol{H}_t^{-1}} + \sqrt{2}\beta \|\boldsymbol{x}_t\|_{\boldsymbol{H}_t^{-1}} + (\boldsymbol{x}_{t,\star} - \boldsymbol{x}_t)^\top \widehat{\boldsymbol{\theta}}_{s(t)} \right] \\
&\leq \dot{\mu}(z_t) \sum_{i=1}^N \left[ 2\sqrt{2}\beta \|\boldsymbol{x}_{t,\star}^i\|_{(\boldsymbol{H}_t^i)^{-1}} + 2\sqrt{2}\beta \|\boldsymbol{x}_t^i\|_{(\boldsymbol{H}_t^i)^{-1}} + (\boldsymbol{x}_{t,\star}^i - \boldsymbol{x}_t^i)^\top \widehat{\boldsymbol{\theta}}_{s(t)}^i \right]
\end{aligned}
$$

where the third inequality follows by applying the Cauchy-Schwarz inequality and Lemma C.2 in tandem, the second-to-last inequality follows from Lemma C.5, and the final inequality follows from Lemma C.10.

Now, since $\boldsymbol{x}_t$ is chosen, for some fixed slot $i \in [N]$, from *Step 17* of Algorithm 2, we have

$$
(\boldsymbol{x}_t^i)^\top \widehat{\boldsymbol{\theta}}_{s(t)}^i + 2\sqrt{2}\beta \|\boldsymbol{x}_t^i\|_{(\boldsymbol{H}_t^i)^{-1}} \geq (\boldsymbol{x}_{t,\star}^i)^\top \widehat{\boldsymbol{\theta}}_{s(t)}^i + 2\sqrt{2}\beta \|\boldsymbol{x}_{t,\star}^i\|_{(\boldsymbol{H}_t^i)^{-1}}.
$$

Rearranging and summing over all $i \in [N]$, we get

$$
\sum_{i=1}^N (\boldsymbol{x}_{t,\star}^i - \boldsymbol{x}_t^i)^\top \widehat{\boldsymbol{\theta}}_{s(t)}^i \leq 2\sqrt{2}\beta \sum_{i=1}^N \|\boldsymbol{x}_t^i\|_{(\boldsymbol{H}_t^i)^{-1}} - 2\sqrt{2}\beta \sum_{i=1}^N \|\boldsymbol{x}_{t,\star}^i\|_{(\boldsymbol{H}_t^i)^{-1}}.
$$

Substituting this back, we get

$$
\mu(\boldsymbol{x}_{t,\star}^\top \boldsymbol{\theta}^\star) - \mu(\boldsymbol{x}_t^\top \boldsymbol{\theta}^\star) \leq 4\sqrt{2}\beta \dot{\mu}(z_t) \sum_{i=1}^N \|\boldsymbol{x}_t^i\|_{(\boldsymbol{H}_t^i)^{-1}}.
$$

Now, using the self-concordance property of GLMs, we have that

$$
\dot{\mu}(z_t) \leq \dot{\mu}(\boldsymbol{x}_t^\top \widehat{\boldsymbol{\theta}}_0) \exp \left( R \left| z_t - \boldsymbol{x}_t^\top \widehat{\boldsymbol{\theta}}_0 \right| \right).
$$

We now bound $\left| z_t - \boldsymbol{x}_t^\top \widehat{\boldsymbol{\theta}}_0 \right|$ as follows:

$$
\begin{aligned}
\left| z_t - \boldsymbol{x}_t^\top \widehat{\boldsymbol{\theta}}_0 \right| &\leq \left| \boldsymbol{x}_t^\top \boldsymbol{\theta}^\star - \boldsymbol{x}_t^\top \widehat{\boldsymbol{\theta}}_0 \right| + \left| \boldsymbol{x}_t^\top \boldsymbol{\theta}^\star - z_t \right| \\
&\leq \left| \boldsymbol{x}_t^\top \boldsymbol{\theta}^\star - \boldsymbol{x}_t^\top \widehat{\boldsymbol{\theta}}_0 \right| + \left| \boldsymbol{x}_t^\top \boldsymbol{\theta}^\star - \boldsymbol{x}_{t,\star}^\top \boldsymbol{\theta}^\star \right| \\
&\leq \left| \boldsymbol{x}_t^\top \boldsymbol{\theta}^\star - \boldsymbol{x}_t^\top \widehat{\boldsymbol{\theta}}_0 \right| + \left| \boldsymbol{x}_t^\top \boldsymbol{\theta}^\star - \boldsymbol{x}_t^\top \widehat{\boldsymbol{\theta}}_0 \right| + \left| \boldsymbol{x}_{t,\star}^\top \boldsymbol{\theta}^\star - \boldsymbol{x}_{t,\star}^\top \widehat{\boldsymbol{\theta}}_0 \right| + \left| \boldsymbol{x}_t^\top \widehat{\boldsymbol{\theta}}_0 - \boldsymbol{x}_{t,\star}^\top \widehat{\boldsymbol{\theta}}_0 \right| \\
&\leq \frac{5}{R}
\end{aligned}
$$

where the second inequality follows from the fact that $z_t \in [\boldsymbol{x}_t^\top \boldsymbol{\theta}^\star, \boldsymbol{x}_{t,\star}^\top \boldsymbol{\theta}^\star]$ and the final inequality follows from Lemma C.3, Lemma C.6, and Lemma C.7.

Similarly, we have

$$
\begin{aligned}
\left| z_t - \boldsymbol{x}_{t,\star}^\top \boldsymbol{\theta}^\star \right| &\leq \left| \boldsymbol{x}_t^\top \boldsymbol{\theta}^\star - \boldsymbol{x}_{t,\star}^\top \boldsymbol{\theta}^\star \right| \\
&\leq \left| \boldsymbol{x}_t^\top \boldsymbol{\theta}^\star - \boldsymbol{x}_t^\top \widehat{\boldsymbol{\theta}}_0 \right| + \left| \boldsymbol{x}_{t,\star}^\top \boldsymbol{\theta}^\star - \boldsymbol{x}_{t,\star}^\top \widehat{\boldsymbol{\theta}}_0 \right| + \left| \boldsymbol{x}_t^\top \widehat{\boldsymbol{\theta}}_0 - \boldsymbol{x}_{t,\star}^\top \widehat{\boldsymbol{\theta}}_0 \right| \\
&\leq \frac{4}{R}.
\end{aligned}
$$

Thus, we get

$$
\dot{\mu}(z_t) \leq \sqrt{e^5 \dot{\mu}(\boldsymbol{x}_t^\top \widehat{\boldsymbol{\theta}}_0)} \sqrt{e^4 \dot{\mu}(\boldsymbol{x}_{t,\star}^\top \boldsymbol{\theta}^\star)} = \sqrt{e^9 \dot{\mu}(\boldsymbol{x}_t^\top \widehat{\boldsymbol{\theta}}_0) \dot{\mu}(\boldsymbol{x}_{t,\star}^\top \boldsymbol{\theta}^\star)}
$$

and hence,

$$\mu(\boldsymbol{x}_{t,\star}^\top \boldsymbol{\theta}^\star) - \mu(\boldsymbol{x}_t^\top \boldsymbol{\theta}^\star) \le 4\sqrt{2}e^5 \beta \sqrt{\dot{\mu}(\boldsymbol{x}_{t,\star}^\top \boldsymbol{\theta}^\star) \cdot e^{-1} \dot{\mu}(\boldsymbol{x}_t^\top \widehat{\boldsymbol{\theta}}_0)} \sum_{i=1}^N \|\boldsymbol{x}_t^i\|_{(\boldsymbol{H}_t^i)^{-1}}.$$

$\square$

**Lemma C.9.** *(Lemma B.17, Sawarni et al. (2024)) The total number of policy switches executed by Algorithm 2 is* $\mathcal{O}(Nd\log T)$.

*Proof.* The proof follows on the same lines as Lemma B.17 from Sawarni et al. (2024). However, all the matrices $\boldsymbol{H}$ are now $Nd$-dimensional, resulting in a dependence on $N$. $\square$

### C.4. Showing Multiplicative Equivalence for `RS-SlateGLinCB`

**Lemma C.10.** *Let* $\boldsymbol{H}_t$ *and* $\boldsymbol{H}_t^i$ *be defined as in Section C.1. Define* $T(\neg 0) := \frac{48L_\mu^2 + 8L_\mu N\rho}{3\rho^2}(N-1)^2 \log\left(\frac{2dNT}{\delta}\right)$. *Then, assuming the diversity assumptions (Section 2) hold, with high probability, for all* $t$ *such that* $|\mathcal{T}_{\neg 0}^{<t}| \ge T(\neg 0)$, *we have that*

$$\frac{1}{4}diag(\boldsymbol{H}_t^1, \ldots, \boldsymbol{H}_t^N) \preceq \boldsymbol{H}_t \preceq \frac{7}{4}diag(\boldsymbol{H}_t^1, \ldots, \boldsymbol{H}_t^N).$$

*Consequently, for any* $\boldsymbol{x} = (\boldsymbol{x}^1, \ldots, \boldsymbol{x}^N)$, *we have that*

$$\|\boldsymbol{x}\|_{\boldsymbol{H}_t^{-1}} \le 2\sum_{i=1}^N \|\boldsymbol{x}^i\|_{(\boldsymbol{H}_t^i)^{-1}}.$$

*Proof.* Define $\overline{\boldsymbol{x}}_t := \sqrt{\dot{\mu}(\boldsymbol{x}_t^\top \widehat{\boldsymbol{\theta}}_0)e^{-1}}\boldsymbol{x}_t$ and $\overline{\boldsymbol{x}}_t^i := \sqrt{\dot{\mu}(\boldsymbol{x}_t^\top \widehat{\boldsymbol{\theta}}_0)e^{-1}}\boldsymbol{x}_t^i$. Then, note that $\|\overline{\boldsymbol{x}}_t^i\| \le \sqrt{L_\mu N^{-1}}$.

Also, for the sake of the proof, define $\boldsymbol{U}_t := \text{diag}(\boldsymbol{H}_t^1, \ldots, \boldsymbol{H}_t^N)$. Thus, we can write

$$\boldsymbol{H}_t = \lambda \boldsymbol{I}_{Nd} + \sum_{s \in \mathcal{T}_{\neg 0}^{<t}} \overline{\boldsymbol{x}}_s \overline{\boldsymbol{x}}_s^\top \quad \text{and} \quad \boldsymbol{H}_t^i = \lambda \boldsymbol{I} + \sum_{s \in \mathcal{T}_{\neg 0}^{<t}} \overline{\boldsymbol{x}}_s^i \overline{\boldsymbol{x}}_s^{i\top}.$$

and using Lemma B.1 from Goyal & Sinha (2025), we have that

$$\boldsymbol{U}_t^{-1/2} \boldsymbol{H}_t \boldsymbol{U}_t^{-1/2} = \boldsymbol{I}_{Nd} + \boldsymbol{G}_t.$$

where $(\boldsymbol{G}_t)_{ij} = \mathbb{1}\{i \ne j\}(\boldsymbol{H}_t^i)^{-1/2}\boldsymbol{H}_t^{(i,j)}(\boldsymbol{H}_t^j)^{-1/2}$ and $\boldsymbol{H}_t^{(i,j)} = \sum_{t \in \mathcal{T}_{\neg 0}^{<t}} \overline{\boldsymbol{x}}_t^i \overline{\boldsymbol{x}}_t^{j\top}$. A straightforward application of Lemma D.2 from Goyal & Sinha (2025) with the quantities $m_1 = m_2 = \sqrt{L_\mu N^{-1}}$, $d_1 = d_2 = d$ and $\delta = \frac{2\delta}{N(N-1)}$, followed by a union bound over all $i \in [N]$ and $j \in [i+1, N]$ like in Lemma A.13, gives us, with high probability, for all pairs of $(i,j)$ where $i < j$,

$$\|\boldsymbol{H}_t^{(i,j)}\| \le \sqrt{\frac{8L_\mu^2}{N^2} |\mathcal{T}_{\neg 0}^{<t}| \log\left(\frac{dN(N-1)}{\delta}\right)}.$$

Now, using the diversity conditions and the definition of $\kappa$, we have that

$$\mathbb{E}[\overline{\boldsymbol{x}}_t^i \overline{\boldsymbol{x}}_t^{i\top} \mid \mathcal{F}_{t-1}] = \mathbb{E}[\dot{\mu}(\boldsymbol{x}_t^\top \widehat{\boldsymbol{\theta}}_0)e^{-1}\boldsymbol{x}_t^i \boldsymbol{x}_t^{i\top} \mid \mathcal{F}_{t-1}] \succeq \rho\kappa^{-1}e^{-1}\boldsymbol{I}.$$

Similar to Lemma A.13, applying Lemma D.1 using the quantities $\alpha = \kappa^{-1}e^{-1} \le 1$, $m = \sqrt{L_\mu N^{-1}}$, $\gamma = \lambda$, $\delta = \frac{\delta}{N}$ and $c = 0.5$, alongside the fact that $(N-1)^2 \ge N^{-2}$, and followed by a union bound over all $i \in [N]$ gives us with high probability,

$$\lambda_{\min}\left(\boldsymbol{H}_t^i\right) \ge \lambda + \frac{\rho|\mathcal{T}_{\neg 0}^{<t}|}{2} \,\forall t \text{ such that } |\mathcal{T}_{\neg 0}^{<t}| \ge T(\neg 0) := \frac{48L_\mu^2 + 8L_\mu N\rho}{3\rho^2}(N-1)^2 \log\left(\frac{2dNT}{\delta}\right).$$

Now, once again, for $i \in [N-1]$, define $\boldsymbol{Z}_t^i \in \mathbb{R}^{d \times id}$ as the following matrix: for $j \in [i]$, the $j^{th}$ $d \times d$ block of $\boldsymbol{Z}_t^i$ is given by $(\boldsymbol{H}_t^{N-i})^{-1/2} \boldsymbol{H}_t^{(N-i,N-i+j)} (\boldsymbol{H}_t^{N-i+j})^{-1/2}$.

Then, using Lemma B.7 from Goyal & Sinha (2025) and following a similar approach to that shown in Lemma A.13, we have that,

$$\|\boldsymbol{Z}_t^i\| \leq \sum_{j \in [i]} \sqrt{\frac{96 \log\left(\frac{dN(N-1)}{\delta}\right)}{N^2(48L_\mu^2 + 8L_\mu N\rho)(N-1)^2 \log\left(\frac{2dNT}{\delta}\right)}} \leq \frac{3i}{2N(N-1)}.$$

Writing $\boldsymbol{G}_t$ as a matrix recurrence relation as in Lemma A.13 and using Lemma B.2 from Goyal & Sinha (2025) gives:

$$\lambda_{\max}(\boldsymbol{G}_t) \leq \frac{3}{4} \quad \text{and} \quad \lambda_{\min}(\boldsymbol{G}_t) \geq -\frac{3}{4}.$$

Substituting $\boldsymbol{G}_t = \boldsymbol{U}_t^{-1/2} \boldsymbol{H}_t \boldsymbol{U}_t^{-1/2} - \boldsymbol{I}_{Nd}$, we get that

$$\frac{1}{4}\boldsymbol{U}_t \preceq \boldsymbol{H}_t \preceq \frac{7}{4}\boldsymbol{U}_t.$$

This finishes the proof for the first part. The second part is exactly the same as in Lemma A.13. $\qquad\square$

**Lemma C.11.** *Let $\boldsymbol{V}$ and $\boldsymbol{V}^i$ be defined as in Section C.1. Let $|\mathcal{T}_0| \geq T(0) := \frac{48+8N\rho}{3\rho^2}(N-1)^2 \log\left(\frac{2dNT}{\delta}\right)$. Then, assuming the diversity assumptions (Section 2) hold, with high probability, we have that*

$$\frac{1}{4}diag(\boldsymbol{V}^1, \ldots, \boldsymbol{V}^N) \preceq \boldsymbol{V} \preceq \frac{7}{4}diag(\boldsymbol{V}^1, \ldots, \boldsymbol{V}^N).$$

*Consequently, for any $\boldsymbol{x} = (\boldsymbol{x}^1, \ldots, \boldsymbol{x}^N)$, we have that*

$$\|\boldsymbol{x}\|_{\boldsymbol{V}^{-1}} \leq 2 \sum_{i=1}^{N} \|\boldsymbol{x}^i\|_{(\boldsymbol{V}^i)^{-1}}.$$

*Proof.* The proof is the same as that of Lemma A.14. $\qquad\square$

### C.5. Other Relevant Lemmas

**Lemma C.12.** *(Elliptical Potential Lemma, Lemma 10 Abbasi-Yadkori et al. (2011)) Let $\{\boldsymbol{x}_s\}_{s \in [t]}$ be a sequence of vectors in $\mathbb{R}^d$ such that $\|\boldsymbol{x}\| \leq L$ for all $s \in [t]$. Define*

$$\boldsymbol{V}_s = \lambda \boldsymbol{I} + \sum_{m=1}^{s-1} \boldsymbol{x}_m \boldsymbol{x}_m^\top$$

*where $\lambda \geq L^2$. Then, we have*

$$\det \boldsymbol{V}_t \leq \left(\lambda + \frac{tL^2}{d}\right)^d \quad \text{and} \quad \sum_{s \in [t]} \|\boldsymbol{x}_s\|_{\boldsymbol{V}_s^{-1}}^2 \leq 2d \log\left(1 + \frac{L^2 t}{\lambda d}\right).$$

**Lemma C.13.** *(Lemma 12, Abbasi-Yadkori et al. (2011)) Let $\boldsymbol{A} \succeq \boldsymbol{B} \succ \boldsymbol{0}$. Then, we have that*

$$\sup_{\boldsymbol{x} \neq \boldsymbol{0}} \frac{\boldsymbol{x}^\top \boldsymbol{A} \boldsymbol{x}}{\boldsymbol{x}^\top \boldsymbol{B} \boldsymbol{x}} \leq \frac{\det \boldsymbol{A}}{\det \boldsymbol{B}}.$$

# D. Demonstrating Linear Growth of Eigenvalues of the Design Matrices

We first recall the diversity assumptions from Section 2:

Define $\mathcal{F}_{t-1} = \sigma(\boldsymbol{x}_1, r_1, \ldots \boldsymbol{x}_{t-1}, r_{t-1})$. For all slots $i \in [N]$ and some $\rho > 0$, we have that

$$E[\boldsymbol{x}_t^i \mid \mathcal{F}_{t-1}] = \boldsymbol{0} \quad \text{and} \quad \mathbb{E}[\boldsymbol{x}_t^i {\boldsymbol{x}_t^i}^\top \mid \mathcal{F}_{t-1}] \succeq \rho \boldsymbol{I}.$$

Next, we state a generalization of Lemma D.1 from Goyal & Sinha (2025) and give a proof for the same:

**Lemma D.1.** *(Generalization of Lemma D.1, Goyal & Sinha (2025)) Let $\{\boldsymbol{x}_s\}_{s \in [T]}$ be a stochastic process in $\mathbb{R}^d$ such that $\mathbb{E}[\boldsymbol{x}_s \mid \mathcal{F}_{s-1}] = \boldsymbol{0}$ and $\mathbb{E}[\boldsymbol{x}_s \boldsymbol{x}_s^\top \mid \mathcal{F}_{s-1}] \succeq \alpha \rho I$, where $\alpha \in (0,1]$. Let $\|\boldsymbol{x}_s\|_2 \leq m \ \forall s \in [T]$. Also, define the matrix*

$$\boldsymbol{Q}_t = \gamma \boldsymbol{I} + \sum_{s=1}^t \boldsymbol{x}_s \boldsymbol{x}_s^\top.$$

*Then, with probability at least $1 - \delta$, we have that*

$$\lambda_{\min}(\boldsymbol{Q}_t) \geq \gamma + c\rho t$$

*for $c \in (0,1)$ and $t \in \left( \frac{12m^4 + 4(1-c)m^2\rho}{3(1-c)^2\rho^2} \log\left(\frac{2dT}{\delta}\right), T \right)$.*

*Proof.* Define $\Sigma_C := \mathbb{E}[\boldsymbol{x}_s \boldsymbol{x}_s^\top \mid \mathcal{F}_{s-1}] \succeq \alpha \rho \boldsymbol{I}$. Also, define the matrix martingale $\boldsymbol{Z}_t = \sum_{s \in [t]} \boldsymbol{x}_s \boldsymbol{x}_s^\top - t\Sigma_C$ and $\boldsymbol{Z}_0 = 0$. Finally, define the martingale difference sequence $\boldsymbol{X}_s = \boldsymbol{Z}_s - \boldsymbol{Z}_{s-1}$ for all $s \geq 1$.

Then, we have that $\|\Sigma_C\| \leq m^2$, and $\|\boldsymbol{X}_s\| = \|\boldsymbol{x}_s \boldsymbol{x}_s^\top + \Sigma_C\| \leq 2m^2$. A calculation similar to Lemma D.1 from Goyal & Sinha (2025) shows that

$$\sum_{s \in [t]} \|\mathbb{E}[\boldsymbol{X}_s^\top \boldsymbol{X}_s \mid \mathcal{F}_{s-1}]\| = \sum_{s \in [t]} \|\mathbb{E}[\boldsymbol{X}_s \boldsymbol{X}_s^\top \mid \mathcal{F}_{s-1}]\| \leq 2m^4 t.$$

Thus, an application of Lemma D.2 with the quantities $d_1 = d_2 = d$, $R = 2m^2$, $w = m^2\sqrt{2t}$, and $u = (1-c)\rho t$ for some $c \in (0,1)$ results in

$$\mathbb{P}\left\{ \left\| \sum_{s \in [t]} \boldsymbol{x}_s \boldsymbol{x}_s^\top - t\Sigma_C \right\| \geq (1-c)\rho t \right\} \leq 2d \exp\left( -\frac{3(1-c)^2\rho^2 t^2}{12m^4 t + 4(1-c)m^2\rho t} \right)$$

Thus, for all $t \geq T_0 := \frac{12m^4 + 4(1-c)m^2\rho}{3(1-c)^2\rho^2} \log\left(\frac{2dT}{\delta}\right)$, we have that with probability at least $1 - \frac{\delta}{T}$,

$$\left\| \sum_{s \in [t]} \boldsymbol{x}_s \boldsymbol{x}_s^\top - t\Sigma_C \right\| \leq (1-c)\rho t.$$

Now, using the fact that $\|\boldsymbol{A}\| = \lambda_{\max}(\boldsymbol{A})$ and $\lambda_{\max}(-\boldsymbol{A}) = -\lambda_{\min}(\boldsymbol{A})$, we have that

$$\|\boldsymbol{A} - \boldsymbol{B}\| = \|(-\boldsymbol{A}) - (-\boldsymbol{B})\| \geq |\lambda_{\max}(-\boldsymbol{A}) - \lambda_{\max}(-\boldsymbol{B})| = |\lambda_{\min}(\boldsymbol{A}) - \lambda_{\min}(\boldsymbol{B})|$$

and thus, we can write that with probability $1 - \frac{\delta}{T}$,

$$(1-c)\rho t \geq t\lambda_{\min}(\Sigma_C) - \lambda_{\min}\left( \sum_{s \in [t]} \boldsymbol{x}_s \boldsymbol{x}_s^\top \right).$$

Rearranging results in

$$\lambda_{\min}\left( \sum_{s \in [t]} \boldsymbol{x}_s \boldsymbol{x}_s^\top \right) \geq t\lambda_{\min}(\Sigma_C) - (1-c)\rho t \geq \alpha\rho t - (1-c)\rho t \geq c\rho t$$

where the last inequality uses the fact that $\alpha \in (0,1]$. A union bound over all $t \in [T_0, T]$ finishes the proof. $\square$

**Lemma D.2.** *(Matrix Freedman Inequality) Define a matrix martingale $\boldsymbol{Z}_s \in \mathbb{R}^{d_1 \times d_2}$ with respect to the filtration $\mathcal{F}_s$ and the corresponding martingale difference sequence $\boldsymbol{X}_s = \boldsymbol{Z}_s - \boldsymbol{Z}_{s-1}$. Assume that the difference sequence is uniformly bounded a.s., i.e, $\|\boldsymbol{X}_s\| \leq R$. Define*

$$\boldsymbol{W}_{row,t} := \sum_{s \in [t]} \mathbb{E}[\boldsymbol{X}_s \boldsymbol{X}_s^\top \mid \mathcal{F}_{s-1}].$$

$$\boldsymbol{W}_{col,t} := \sum_{s \in [t]} \mathbb{E}[\boldsymbol{X}_s^\top \boldsymbol{X}_s \mid \mathcal{F}_{s-1}].$$

*Then, for all $u \geq 0$ and $w^2 > 0$, we have that*

$$\mathbb{P}\{\exists t \geq 0 : \|\boldsymbol{Z}_t\| \geq u \text{ and } \max\{\|\boldsymbol{W}_{row,t}\|, \|\boldsymbol{W}_{col,t}\|\} \leq w^2\} \leq (d_1 + d_2) \exp\left(-\frac{3u^2}{6w^2 + 2Ru}\right).$$

# E. Additional Experiments and Experimental Setup

In this section, we detail the implementation of all our algorithms and baselines. Then, we provide some additional experiments, building upon the setup in Section 5.

While the implementations of `RS-GLinCB` and `RS-MNL` are publicly available, at each time round, they iterate over the set of slates. Hence, we speed up the implementation using `np.einsum`. Also, for `RS-MNL`, we set the number of outcomes to 1, corresponding to the logistic setting (Midigeshi et al., 2025). We also implement a version of `SoftBatch`, and to lift it to the GLM setting, we replace the least squares estimate of the parameter with an MLE estimate. Also, since $q = O(T^{-\log T})$, we set $q$ as the machine epsilon[8]. For `B-SlateGLinCB`, we double the batch lengths and the number of batches, i.e, we calculate the batch lengths as

$$\mathcal{T}_m = \lfloor 2T^{1-2^{-m//2}} \rfloor$$

where $a//b$ represents integer division. Note that the number of batches is still $\mathcal{O}(\log \log T)$ and the regret only scales by a constant factor. This allows for a better estimate of the parameter in the initial batches, while increasing the number and speed of updates during the later stages. Finally, for the higher-dimensional experiments (Experiment **E2**; explained next), we limit the number of optimization steps for `RS-SlateGLinCB` to 25, with the hope that limiting the convergence of policy updates is offset by the higher number of updates.

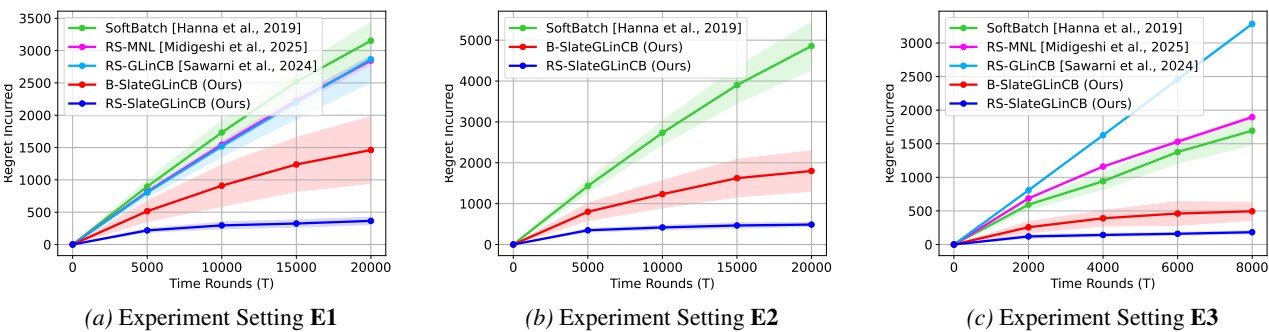

*(a)* Experiment Setting **E1**      *(b)* Experiment Setting **E2**      *(c)* Experiment Setting **E3**

*Figure 4.* Comparison with limited adaptivity algorithms, `SoftBatch`, `RS-MNL`, and `RS-GLinCB`

We now explain the experimental setup. At each $t \in [T]$, for each slot $i \in [N]$, the set of items $\mathcal{X}_t^i \subset \mathbb{R}^d$, is chosen such that $|\mathcal{X}_t^i| = K = 5$ and $d = 5$. Each item in $\mathcal{X}_t^i$ is sampled from $[-1, 1]^5$ and is normalized to have $\ell_2$-norm $1/\sqrt{N}$ where $N$ varies depending on the experiment setting. We randomly select $\theta^\star$ from $[-1, 1]^{Nd}$ and normalize it to have $\ell_2$-norm $S$. For our algorithms, we set $\delta = 1/N^2$, which puts it in the range $[0.004, 0.04]$ for the values of $N$ used. For the baselines, we use the default values of $\delta$ provided in the corresponding implementation, which are of the same order as ours. We now explain the choice of the experimental settings:

**E1**: We set $S = 2$ and $N = 5$, resulting in $K^N = 3125$ slates with dimension $Nd = 25$ and $\kappa \le e^S \approx 7.38$. We run our algorithms for $T \in \{5000, 10000, 15000, 20000\}$ rounds and display the results in Figure 4a.

**E2**: We set $S = 5$ and $N = 10$, resulting in $K^N = 9765725$ slates with dimension $Nd = 50$ and $\kappa \le e^S \approx 150$. We run our algorithms for $T \in \{5000, 10000, 15000, 20000\}$ rounds. In this experiment, we do not compare to `RS-GLinCB` and `RS-MNL` since these algorithms are not well-suited for large action spaces. We display the results in Figure 4b.

**E3**: We set $S = 5$ and $N = 5$, resulting in $K^N = 3125$ slates with dimension $Nd = 25$ and $\kappa \le e^S \approx 150$. We run our algorithms for $T \in \{2000, 4000, 6000, 8000\}$ rounds and display the results in Figure 4c.

We average all the results over 25 different seeds for sampling rewards and display the results in Figure 4. In all three settings, we see that our algorithms achieve sublinear regret and outperform the other baselines by a significant margin. These results also provide strong empirical support for our $\kappa$-free regret guarantees in Theorem A.1 and Theorem C.1. Also, we see that the regret of `RS-SlateGLinCB` is better than `B-SlateGLinCB` in all the settinfs, which can possibly be attributed to better constants, as well as, the $\sqrt{d}$ gap between the bounds in our theorems.

---

[8]For our device, the value of $q$ is set to be $1.1920929 \times 10^{-7}$.

# F. `B-SlateGLinCB+`: Additional Observations and Insights

In this section, we build upon the observations and insights for `B-SlateGLinCB+`, which was first introduced in Section 5. We first explain some empirical observations, which motivated the design of this algorithm. Then, we highlight the major differences between `B-SlateGLinCB` and `B-SlateGLinCB+`.

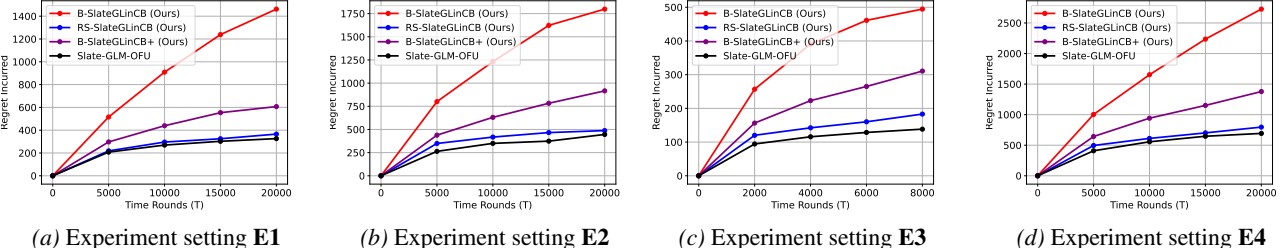

*(a)* Experiment setting **E1**    *(b)* Experiment setting **E2**    *(c)* Experiment setting **E3**    *(d)* Experiment setting **E4**

*Figure 5.* Comparison with fully sequential slate bandit algorithm `Slate-GLM-OFU`

In Figure 5, we compare our algorithms `B-SlateGLinCB` and `RS-SlateGLinCB` to the fully sequential `Slate-GLM-OFU` (Algorithm 1, Goyal & Sinha (2025)). We retain the same experimental setup as in Appendix E, and add an experiment described below:

**E4**: We set $S = 5$ and $N = 15$, resulting in $K^N = 30517578125$ slates with dimension $Nd = 75$ and $\kappa \le e^S \approx 150$. We run the algorithms for $T \in \{5000, 10000, 15000, 20000\}$ and display the results in Figure 5d.

In Figure 5, we see that the gap between `RS-SlateGLinCB` and `Slate-GLM-OFU` is very small, even though we would expect `Slate-GLM-OFU` to have much better regret because of its fully sequential nature. However, there remains a significant gap between the regrets incurred by `B-SlateGLinCB` and `Slate-GLM-OFU`. This raises the question of whether we can develop an algorithm that performs $\mathcal{O}(\log \log T)$ updates and can compete with the likes of `RS-SlateGLinCB` and `Slate-GLM-OFU`. Based on our empirical observations, we modify `B-SlateGLinCB` to obtain `B-SlateGLinCB+`, which is also a batched algorithm that performs $\mathcal{O}(\log \log T)$ updates and incurs regret similar to that of `Slate-GLM-OFU`. The modifications made are completely based on empirical observations and heuristics, which we explain in the next paragraph, and leave the theoretical analysis of this algorithm as an interesting future direction.

In `B-SlateGLinCB`, we notice that the estimate of the parameter $\boldsymbol{\theta}^\star$ improves throughout the course of the algorithm, that is, $\|\boldsymbol{\theta}^\star - \widehat{\boldsymbol{\theta}}_m\|_2$ strictly decreases as the algorithm progresses. We also observe that often, the optimal items in each slot get eliminated, especially in the initial rounds of pruning. Thus, we hypothesize that the optimal items get eliminated in the initial rounds of pruning because the estimates of the parameter are often suboptimal. Clearly, the estimates of $\boldsymbol{\theta}^\star$ learned during the later batches are better representative of $\boldsymbol{\theta}^\star$. Hence, the later estimates should carry more weight in deciding the set of items retained after elimination. This opens the gate to several algorithms with different elimination techniques to achieve the exploration-exploitation tradeoff, such as weighted majority-like strategies with a higher weight given to the more recent batches, or eliminating items from the item-set with respect to only the last $m' < m$ batches.

Thus, in `B-SlateGLinCB+`, we set $m' = 1$, i.e, we eliminate items with respect to only the most recent estimate, and the scaling slate $\boldsymbol{b}_t$ is chosen from this set of remaining items. We do not prune with respect to the warmup estimate $\widehat{\boldsymbol{\theta}}_0$ anymore. Further, at each policy update, we allow the algorithm to use all the previous data seen during the course of the algorithm, unlike `B-SlateGLinCB`, where the algorithm only uses the data from the corresponding batch. This improves the quality of the estimation of the parameter. We present the empirical performance of `B-SlateGLinCB+` in Figure 5, and see that the algorithm incurs sublinear regret, and closely matches the regrets incurred by `RS-SlateGLinCB` and `Slate-GLM-OFU`. An interesting future direction is to study the constraints under which we can prove strong regret bounds for such heuristics.

# G. Prompt Tuning Experiments

In this section, we explain the experimental set up of our prompt tuning experiments.

We use RoBERTa-large (Liu et al., 2021) as the base model and Nomic-Embed-Text-v1.5 (Nussbaum et al., 2024) as the embedding model for all our experiments. At each round $t$, the algorithm is presented with a query $q_t$ and $N$ (different) sets consisting of $K$ candidate examples each. Each set of candidate examples corresponds to one exemplar (*slot*) in the prompt (*slate*). At round $t$, we denote the $j^{th}$ candidate example for slot $i$ as $(e_t^{ij}, l_t^{ij})$, where $e$ denotes the example, and $l \in \{0, 1\}$ denotes the true label for the example.

We now describe the construction of the arm-sets $\{\mathcal{X}_t^i\}_{i \in [N]}$ at each round $t$. For a slot $i$, the feature vector for the $k^{th}$ candidate example $(e_t^{ik}, l_t^{ik})$ is denoted as $(j, l, c)$, where $j, l$ and $c$ are the three components, as described in Section 5. We describe them in greater detail here. $j$ denotes the joint embedding of the query $q_t$ and the candidate example $e_t^{ik}$, $l$ is the example's label, i.e, $l = l_t^{ik}$, and $c = (c_1, c_2)$ represents a pair of scores that measure the similarity between the query and the example. Here, $c_1$ measures the N-gram similarity between the query and the example sentence. This is done by calculating the cosine similarity between the bag-of-character-N-gram vectors of the query and the example. $c_2$ represents the similarity score between the query and the example in the embedding space, which is calculated as the cosine similarity between the embeddings of the two.

We choose $N$, the number of exemplars per prompt, to be 6, and $K$, the number of candidate examples provided to each slot at each round as 9. The prompt instruction and format are fixed apriori and are given below:

> **Prompt Template**
>
> In this task, you are given sentences from movie reviews. The task is to classify a sentence as "great" if the sentiment of the sentence is positive or as "terrible" if the sentiment of the sentence is negative. Following are some examples to help you:
>
> Review1:
> Sentiment1:
> Review2:
> Sentiment2:
> Review3:
> Sentiment3:
> Review4:
> Sentiment4:
> Review5:
> Sentiment5:
> Review6:
> Sentiment6:
>
> Query:
> Sentiment: $<$mask$>$

To demonstrate the long-horizon capability of our algorithm, we augment our test set by including an additional 4000 queries sampled from the training set, resulting in a total of 4870 queries. These additional queries are sampled prior to the construction of the candidate example sets, and hence, we ensure that none of these 4000 queries appear in any of the candidate example sets provided to the algorithm across the horizon. We also ensure there is no further training and instead report the cumulative average accuracy over the entire time horizon.

We compare our results to the following baselines: (i) the base model with no exemplars, (ii) the base model, where the exemplars are chosen randomly at each round, and (iii) `Slate-GLM-OFU`. The random allocation baseline chooses 6 examples from the pool of exemplars with equal probability. We note that all the hyperparameters including embedding size, $N$, $K$, embedding dimensions, as well as, the methodology to select queries, exemplar pools, and the construction of arm-sets is fixed across all baselines. We plot the cumulative average accuracy of all algorithms and display the results in Figure 3.

# H. Empirical Verification of Linear Growth of Eigenvalues for `B-SlateGLinCB`, `RS-SlateGLinCB`, and `B-SlateGLinCB+`

In this section, we empirically validate that the minimum eigenvalue for the design matrices indeed grow (near-) linearly over the course of the algorithms. We choose the number of slots $N = 4$, the dimension of items in each slot $d = 5$, and the number of items per slot $K = 5$, resulting in a total of $K^N = 625$ slates with dimension $Nd = 20$. At each round $t \in [T]$ and each slot $i \in [N]$, the item-sets $\mathcal{X}_t^i$ are chosen in a manner similar to the one described in Section 5. Similarly, the optimal parameter $\boldsymbol{\theta}^\star$ is also sampled in a manner similar to the one described in Section 5. We choose the $\ell_2-$norm of $\boldsymbol{\theta}^\star$ to be $S = 2$. All the algorithms are run for $T = 10000$ for 50 different seeds for sampling rewards.

We display the results for `B-SlateGLinCB` in Figure 6, for `RS-SlateGLinCB` in Figure 7, and for `B-SlateGLinCB+` in Figure 8. Throughout, the black dotted lines represent the transition between batches (or in the case of `RS-SlateGLinCB`, the end of the warmup phase). The blue graphs represent the minimum eigenvalues of the design matrices $\boldsymbol{V}^i$ during the warmup phases (since the $\boldsymbol{V}$ matrices are only updated during this phase), while the red graphs represent the minimum eigenvalues of the Hessian matrices $\boldsymbol{H}^i$ during the corresponding phase of the algorithm. For `RS-SlateGLinCB`, in Figure 7, the first row represents a zoomed-in version of the warmup phase (since the slope of the growth of the blue graph is low), while the second row represents the growth of both $\boldsymbol{V}$ and $\boldsymbol{H}$ during the course of the algorithm. From all the graphs, we see that the growth of the minimum eigenvalues appears to be (near)-linear, thus, validating the conclusion we draw from the Diversity Assumptions (Definition 2.1).

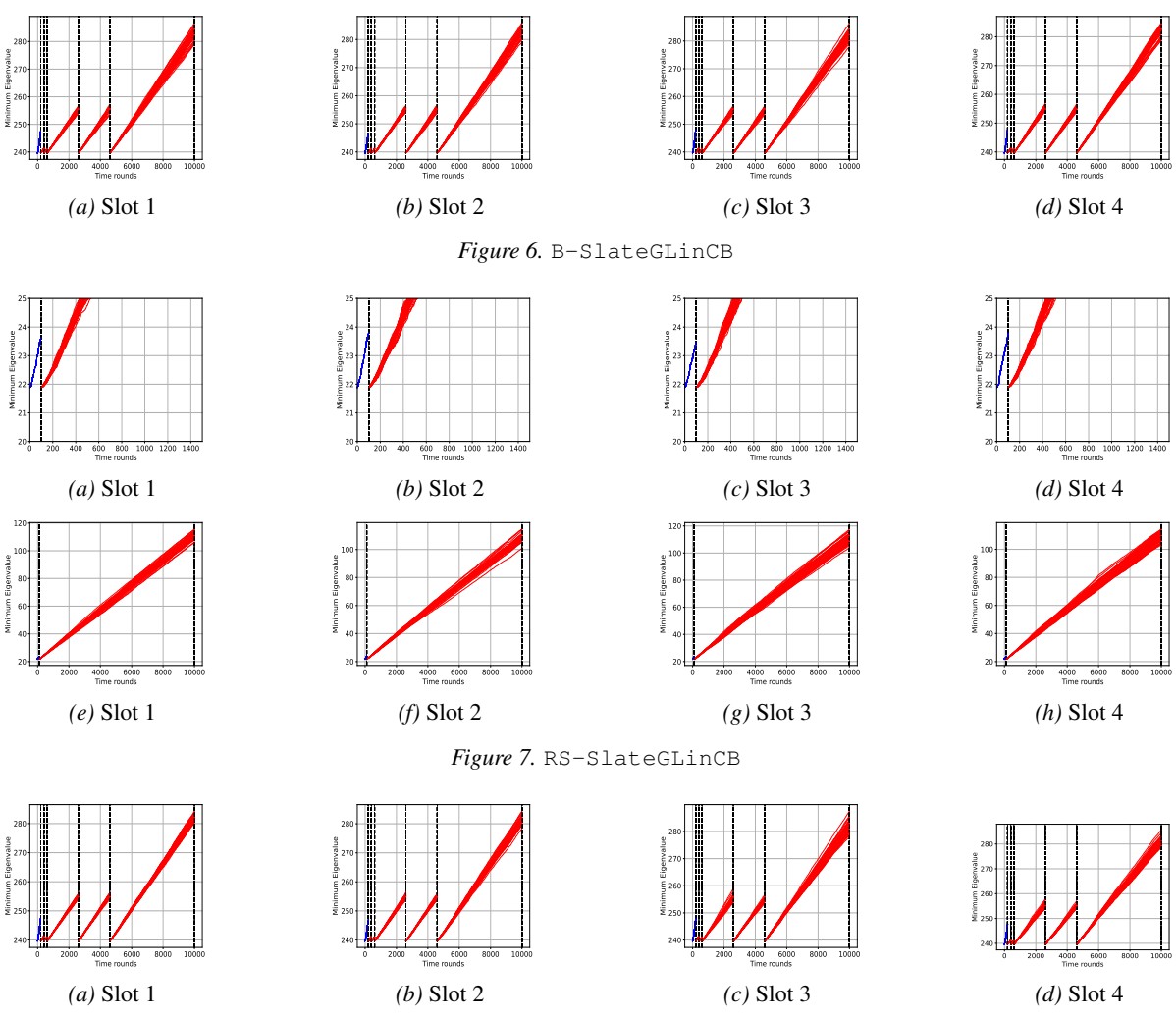

*(a)* Slot 1        *(b)* Slot 2        *(c)* Slot 3        *(d)* Slot 4

*Figure 6.* `B-SlateGLinCB`

*(a)* Slot 1        *(b)* Slot 2        *(c)* Slot 3        *(d)* Slot 4

*(e)* Slot 1        *(f)* Slot 2        *(g)* Slot 3        *(h)* Slot 4

*Figure 7.* `RS-SlateGLinCB`

*(a)* Slot 1        *(b)* Slot 2        *(c)* Slot 3        *(d)* Slot 4

*Figure 8.* `B-SlateGLinCB+`

