# OpenReview forum: "Contextual Slate GLM Bandits with Limited Adaptivity"
_ICML.cc/2026/Conference — ICML 2026 regular_

### Official Review · Reviewer_df9D · 2026-03-04

**Soundness:** 3
**Presentation:** 2
**Significance:** 3
**Originality:** 3
**Overall Recommendation:** 4
**Confidence:** 3

**Summary:**

This paper studies contextual slate GLM bandits and proposes two computationally efficient algorithms: (a) a batched algorithm and (b) a rarely-switching algorithm. The main contribution seems to be avoiding the need to enumerate all possible slates—thereby preventing exponential dependence on N.

**Compliance With Llm Reviewing Policy:**

Affirmed.

**Final Justification:**

My major concerns have been adequately addressed; therefore, I am raising my score to 4.

**Key Questions For Authors:**

What is the lower bound for this setting?

**Limitations:**

yes

**Strengths And Weaknesses:**

### Strengths:
This paper considers an interesting setting, called slate GLM bandits, and proposes computationally efficient algorithms.


### Weaknesses:
1. **Technical novelty**: The main weakness of this paper is the limited technical novelty. The efficient slate selection strategy has already been proposed by Goyal & Sinha (2025), and the batched update (or rare-switching) scheme was introduced in Sawarni et al. (2024). As a result, this work appears to be largely a combination of these prior ideas. In general, extending the logistic bandit setting to the GLM bandit setting seems incremental, and many of the techniques used in the paper appear to be very similar to those in existing work.

2. **Lack of motivation for the setting**: The paper should provide a clearer motivation for the slate GLM bandit setting. For example, in landing page optimization, conversions often exhibit strong interactions between page components. However, the proposed model does not account for such slot interaction effects. Furthermore, the paper assumes that the context set for each slot is independent, whereas in practice the context space is often shared across slots. This setting also appears to be related to the cascading bandit setting [1], but the paper does not clearly explain the advantages of the proposed framework over existing formulations.

3. **Presentation**:

      a) $x$ is defined as a $dN$-dimensional vector. However, in Line 117, the notation $R^{d \times N}$ typically denotes a d by N matrix. This could be confusing.

      b) In Lines 136 and 144, the vectors $x$ and $\theta^*$ should be defined as $dN$-dimensional vectors rather than $d$-dimensional vectors.

      c) Definition 2.1 should be stated as an assumption rather than a definition.

      d) Is the MLE constrained or unconstrained? This should be clearly stated.

      e) The authors should more clearly explain the technical novelties of the paper and more explicitly acknowledge related prior work, such as Sawarni et al., 2024 and Goyal & Sinha, 2025.



[1] Li, Shuai, et al. "Contextual combinatorial cascading bandits." International conference on machine learning. PMLR, 2016.

---

> ### Author Rebuttal · Authors · 2026-03-31
>
> Thanks, we address your concerns below:
>
> *W1. “The main weakness of this paper is the limited technical novelty… existing work.”*
>
> We disagree with the reviewer here. Our algos and their analyses have significant differences from those in [1] and [2]. We highlight the main ones below:
>
> **Comparison with [1]**: The slate selection strategy in [1] is based on UCB applied at the slot level. This entails computing UCB scores for each eligible item and deterministically selecting the item with the highest score. In B-SlateGLinCB, we instead sample items from optimal designs constructed separately for each slot. These designs are a distribution over the item-set such that sampling according to them leads to optimal parameter estimation. The only thing common is the idea of building a slate slot-by-slot, which we expect in a slate bandit scenario. In RS-SlateGLinCB, we use UCB as well but only a subset of good items are scored.
>
> **Comparison with [2]**: [2] does not have a notion of “slots”. Applying their algorithm would require constructing an optimal design over the set of all slates (leading to per-round complexity $K^N$). As mentioned above, our algo B-SlateGLinCB constructs these designs for each slot separately (with other modifications), thereby taking only $poly(K,N)$ time per-round. While this change clearly makes our algorithm efficient, proving optimal regret guarantees is significantly hard and novel (see Lemma A.9 and A.10 in our paper).
>
>
>
> *W2. “In general, extending the logistic bandit setting to the GLM bandit setting .. existing work.”*
>
> We think the reviewer has misunderstood our objective. Our goal is not to just extend the logistic to the GLM setting, but to design limited adaptivity algos (batched, rarely-switching) that are efficient for the slate bandit problem. Limited adaptivity algos for GLM bandits already exist [2], but they are inefficient for slate bandits (as explained above). Addressing this is our main objective.
>
> *W3. "Lack of motivation for the setting … However, the proposed model does not account for such slot interaction effects."*
>
> We agree that practical situations might require strong interactions between slots. Please notice that GLMs also model interactions between slots via the link function and so it is incorrect to say that our proposed model does not account for such interactions. One can definitely go further to models with even stronger interactions captured (say neural networks) but then analyzing such algorithms becomes super hard. GLMs provide a nice sweet spot between the practical requirement of modelling non-trivial interactions while being amenable to theoretical analysis. Going beyond GLMs is definitely an interesting open question for future work.
>
> *W4. "Furthermore, the paper assumes that the context set for each slot is independent, whereas in practice the context space is often shared across slots."*
>
> The setting where the context space is shared across slots is better modelled via combinatorial bandits. Slate bandits, however, address situations where different slots correspond to different component types. For example, a product-ad can be thought of as a slate with two slots; one containing the product-title and the other containing the product-image, thereby justifying the requirement of independent context sets.
>
> *W5. "This setting also appears to be related to the cascading bandit setting… formulations."*
>
> It seems the reviewer might have misunderstood our setting. Our setting is significantly different from cascading bandits [3]. We mention some of the main differences below:
>
>  a. **Feedback type**: In [3], the learner receives linear feedback (with the same parameters $\theta^\star$ and additive subgaussian noise) for each item (semi-bandit feedback) within a subset $O_t$ of the set of items $A_t$ that were played. On the other hand, we assume a single feedback for the entire slate (bandit feedback).
>
> b. **Reward Model**: The reward in [3] is a Lipschitz function of the subgaussian feedback whereas GLMs capture much more general noise models than additive subgaussian noise.
>
> c. **Slate selection**: In [3], valid slates can only be selected via the oracle ($O_s$, defined at the end of Section 3) completely offloading the per-round time complexity to the oracle. Our setting directly enables us to choose valid slates.
>
> d. There is no notion of limited adaptivity in [3].
>
> *W6. Presentation (a-e)*:
>
> Thanks, we will correct the issues raised in (a)-(c). Our MLE is unconstrained. Projecting it to the parameter space worsens the regret by only $poly(R,S)$ (see footnote 10,11, Appendix E in [2]). We will add this explanation. Please see the above discussion for technical novelty with respect to [1,2].
>
> *Q1. Lower bounds*:
>
> Please see our answer to the reviewer c2UG.
>
> **References**:
>
> [1] Goyal and Sinha
>
> [2] Sawarni et al.
>
> [3] Li, Shuai, et al. "Contextual combinatorial cascading bandits." International conference on machine learning. PMLR, 2016.

---

> > ### Author Rebuttal · Reviewer_df9D · 2026-04-02
> >
> > Thank you for the detailed response.
> >
> > Could the authors clarify more precisely why the present results are not obtainable by a relatively direct combination of Goyal & Sinha (2025) and Sawarni et al. (2024)? In particular, beyond the difference in problem setting, please explicitly identify what new technical obstacle arises when transferring those ingredients to the slate GLM limited-adaptivity setting, and which part of your analysis resolves it. As written, it appears that the slate decomposition is inherited largely from Goyal & Sinha (2025), while the limited-adaptivity GLM machinery is inherited largely from Sawarni et al. (2024), so it would be helpful if the authors could state clearly what genuinely new lemma, argument, or coupling step is required here and why it is nontrivial.
> >
> > ---
> > After the authors’ second rebuttal, I now better understand the contributions of this work, and I have raised my score accordingly. Thank you again for the detailed response. For the final version, I encourage the authors to present the technical novelties relative to prior work more clearly.

---

> > > ### Author Response · Authors · 2026-04-03
> > >
> > > Thank you for the follow-up. Please find our answer below:
> > >
> > > **Novelty in Batch-SlateGLinCB wrt prior work**
> > >
> > > 1. **Why direct application of [2] fails**: Applying the methodology of [2] straightaway to our problem, albeit, in a slot-by-slot fashion, immediately results in $\kappa$-dependent regret. In particular, [2] uses a warm-up batch to calculate a parameter estimate $\theta_0$ at the end. For all subsequent batches, they then use $\theta_0$ to scale each arm before constructing an optimal design, i.e, they scale an arm $x$ with $\dot\mu(x^T\theta_0)$. Thus, the natural extension would be: in slot $i$, scale $x^i$ using $\dot\mu((x^i)^T\theta_0^i)$, i.e, use the $i^{th}$ component of the parameter vector. However, this would result in terms of the form $\dot\mu((x^i)^T\theta_0^i)/\dot\mu(x^\top\theta^\star)$, which can be arbitrarily large ($\propto e^S \sim \kappa$) in case of multiple slots. Thus, the natural extension of [2] fails. Note that without such a scaling factor, the regret will scale linearly with $\kappa$ which is a critical issue in GLM bandits. It is unclear what should be the analogous scaling factor for our setting.
> > >
> > > 2. **How we get around this problem**: Another key step in adapting [2] is elimination which now needs to be done at the slot level. This is where we introduce a novel idea that resolves the scaling factor issue described above. First, we construct a scaling slate $b_t$ with items that survive the elimination. This slate is used to scale the Hessian (design matrix $H$) by the factor $\dot\mu(b_t^T\theta_0)$. This new scaled matrix plays a critical role during further elimination, as can be seen in steps 18-21 of our algorithm Batch-SlateGLinCB. Then, we select items by sampling from optimal designs (over the items that survive this elimination). In our analysis (Lemma A.4,A.5), this procedure of scaling the matrix with the new scaling factor leads to terms proportional to $\dot\mu(b_t^T\theta_0)/\dot\mu(x_t^T\theta_0)$, where $x_t$ is the chosen slate. Now, since each item $b^1_t,\ldots, b^N_t$ of $b_t$ itself survived such an elimination, they are already "good" items for their slots. At this point, we leverage Assumption 2.1 (in a way similar to [1]) to show that the slate $b_t$ is itself "good" and $\dot\mu(b_t^T\theta_0)/\dot\mu(x_t^T\theta_0) = O(1)$, thereby, getting rid of a potential
> > > $\kappa$-dependency.
> > >
> > > 3. **Additional novelty- Weakening of Assumption made in [1]**:  The Assumption 2.1 in our paper is much weaker than the corresponding assumption made in [1]. In particular, while [1] assumes eigenvalues of the expected design matrix to be $\Omega(\kappa)$, we only assume it to be $\Omega(1)$. This is critical because the assumption is completely free of the non-linearity of the problem. Our assumption now aligns with other past works [3] which have utilized a similar assumption. While our proof for this weaker assumption builds up on the proof in [1], it also contains more careful and newer insights.
> > >
> > > 4. **Proving optimal regret for slot-by-slot optimal design algorithm**: While the above discussion provides the intuition of how we manage the "scaling challenge" to remove $\kappa$-dependence, proving that the regret of our algorithm grows as $O(\sqrt{T})$ involves further novel ideas (Lemma A.9, A.10) to account for the slot-by-slot selection. These ideas differ at multiple points from the analysis in [1] (which also did slot-by-slot) due to the usage of optimal designs instead of UCB scores in selection.
> > >
> > > **Novelty in RS-SlateGLinCB**
> > >
> > > In the rarely-switching setting, Assumption 2.1 (which we verify in practice in Appendix H) allows us to show that the eigenvalue of the design matrix grows linearly (Lemma C.10), which we then use cleverly to get rid of Switching-Criterion 1 of RS-GLinCB [2] (analysis in Lemma C.3 of our paper). Thus, we obtain an algorithm with $O(d\log T)$ updates, significantly better than [2], which uses $O(\kappa d^2\log^2 T)$ updates. This also means we end up matching the lower bound shown in [4]. In practice, this switching criterion also causes the algorithm of [2] to explore extensively (as also noted by the authors of [5], see section 5), and hence, the removal of this criterion significantly improves performance in practice (see Appendix E).
> > >
> > > **References**
> > >
> > > [1] Goyal and Sinha. Efficient Algorithms for Logistic Contextual Slate Bandits with Bandit Feedback. UAI 2025
> > >
> > > [2] Sawarni et al. Generalized Linear Bandits with Limited Adaptivity. NeurIPS 2024.
> > >
> > > [3] Das and Sinha. Linear Contextual Bandits with Hybrid Payoff: Revisited. ECML PKDD 2024.
> > >
> > > [4] Ruan et al. Linear Bandits with Limited Adaptivity and Learning Distributional Optimal Design, STOC, 2021
> > >
> > > [5] Lee et al. A Unified Confidence Sequence for Generalized Linear Models, with Applications to Bandits. NeurIPS 2024.

---

### Official Review · Reviewer_c2UG · 2026-03-05

**Soundness:** 3
**Presentation:** 3
**Significance:** 2
**Originality:** 2
**Overall Recommendation:** 4
**Confidence:** 3

**Summary:**

This work studies the contextual slate bandit problem under a generalized linear reward model with infrequent policy updates, where at each round the learner is given $N$ sets of items and needs to construct a slate by choosing one item from each set. The authors consider two settings for how the item sets are generated: (a) stochastic and (b) adversarial. In the stochastic setting, it develops B-SlateGLinCB policy that uses $O(\log\log T)$ batches to achieve a regret of order $\tilde{O}(Nd^{3/2}\sqrt{T})$. In the adversarial setting, it introduces RS-SlateGLinCB policy that achieves $\tilde{O}(Nd\sqrt{T})$ regret with $O(d\log T)$ updates. Numerical experiments are provided to evaluate the empirical performance of the proposed methods.

**Compliance With Llm Reviewing Policy:**

Affirmed.

**Final Justification:**

I remain positive about the paper.

**Key Questions For Authors:**

See above.

**Limitations:**

Yes.

**Strengths And Weaknesses:**

Strength:
1. It develops theoretical guarantees for rarely-switching Slate GLM Bandit under both the stochastic and adversarial settings. The technical results appear to be sound.

2. The proposed algorithms have poly($N$) per round time complexity, avoiding enumeration over the exponentially large $2^{\Omega(N)}$ set of possible slates.


Weakness:
1. Can you comment on the tightness of the regret bounds in Theorem 3.1 and 4.1?

2. Both algorithms assume prior knowledge of the norm bound $S$ and the non-linear parameter $\kappa$ as inputs. What would happen if these parameters were unknown?

---

> ### Author Rebuttal · Authors · 2026-03-31
>
> Thanks for your review. We answer your questions below:
>
> *W1. "Can you comment on the tightness of the regret bounds in Theorem 3.1 and 4.1?"*
>
> The lower bounds known in literature for the expected regret of contextual bandits is $\Omega(d\sqrt{T})$ in the linear [2,4] and the logistic case [3]. This bound would translate to $\Omega(Nd\sqrt{T})$ in our setting. However, we would like to note that slate bandits have more structure, i.e, a slate comprises $N$ slots and items are available separately for each slot, and thus, it is not clear whether this lower bound holds in our case. Resolving this and/or obtaining strong lower bounds seems like an interesting question.
>
> a. We would like to note that the regret of our rarely-switching algorithm exactly matches this bound.
>
> b. Our batched algorithm achieves a regret bound of $O(Nd^{3/2}\sqrt{T})$ (using G-optimal designs) and $O(N^{3/2}d\sqrt{T})$ (using distributional optimal designs).
>
> If the $\Omega(Nd\sqrt{T})$ lower bound also holds for our setting, then, the rarely-switching algorithm is tight and the batched algorithm is off by a factor of $\sqrt{d}$ or $\sqrt{N}$ depending on the use of G-Optimal designs or Distributional Optimal designs respectively.
>
> *W2. "Both algorithms assume prior knowledge of the norm bound S and the non-linear parameter $\kappa$  as inputs. What would happen if these parameters were unknown?"*
>
> This question has been partially addressed in [1] (see, Footnote 6 and 7). It suffices to have an upper bound on these parameters. One approach to dealing with not knowing $S$ completely is to construct confidence sets and therefore, algorithms, that are $S$-free. There is some recent work in this direction [5], however, how it can be applied to our setting requires further thinking. We will include this as interesting future work in our paper.
>
> **References**
>
> [1] Sawarni et al. Generalized Linear Bandits with Limited Adaptivity. NeurIPS 2024.
>
> [2] Dani et al. Stochastic Linear Optimization under Bandit Feedback. COLT 2008
>
> [3] Abeille et al. Instance-Wise Minimax-Optimal Algorithms for Logistic Bandits. AISTATS 2021
>
> [4] Cesa-Bianchi et al. Delay and Cooperation in Nonstochastic Bandits. COLT 2016
>
> [5] Lee et al. A Unified Confidence Sequence for Generalized Linear Models, with Applications to Bandits. NeurIPS 2024.

---

> > ### Author Rebuttal · Reviewer_c2UG · 2026-04-03
> >
> > Thank you for your responses. I remain positive about the paper. It would be interesting to explore how the approach can be extended to handle unknown parameters.

---

### Official Review · Reviewer_k9Wc · 2026-03-06

**Soundness:** 3
**Presentation:** 3
**Significance:** 2
**Originality:** 2
**Overall Recommendation:** 4
**Confidence:** 4

**Summary:**

Thie paper studies the contextual slate bandit problem with generalized linear rewards (GLM) under two "limited adaptivity" settings: (a) Batched: The algorithm is required to partition the horizon $[T]$ into $M = O(\log \log T)$ disjoint batches, and the policy is only updated between subsequent batches. (b) Rarely Switching: The algorithm is required to do only $O(\log T)$ estimations of latent reward parameter.

For the batched setting (a), the authors introduce algorithm B-SlateLinCB, achieving $O(N d^{2/3} \sqrt{T})$ regret.

For the rarely switching setting (b), the authors introduce algorithm RS-SlateLinCB, achieving $O(N d \sqrt{T}) regret.

Finally, numerical simulations demonstrate and validate the designed algorithm performance.

**Compliance With Llm Reviewing Policy:**

Affirmed.

**Final Justification:**

I thank the reviewer for addressing my concern.

**Key Questions For Authors:**

See Strengths And Weaknesses

**Limitations:**

See Strengths And Weaknesses

**Strengths And Weaknesses:**

**Strengths**

- Overall, this paper studies an interesting problem. Although I think the results are a bit limited, it is interesting to see some advance in the topic of "contextual slate bandits with limited adaptivity", which has high value in many applications. This paper provides many new insights in this research line.

- The paper studies two very specific settings. The presentation is very clear and well structured, and easy to follow. All claims are well supported. It is a very theoretical paper. I briefly go through the theoretical proofs. Although I do not check them line by line, they appear to be generally correct to the best of my understanding.

**Weaknesses and Questions**

- Both regret bounds appear $O(N)$ term (linear-in-$N$). Is it possible to reduce to sublinear-in-$N$ (e.g. $\sqrt{N}$)?

- The assumptions in both settings are overly specific, restrictive and strong, making them unrealistic in practice. For example, in the batched setting, the assumption on $M=O(\log\log T)$ is overly restrictive. What about other $M$ (e.g. fixed constant, $M = O(\sqrt{T})$)? How could we motivate this $O(\log\log T)$ value? In the rarely switching setting, what is the motivation of "can only do limited times of estimations"? In real applications, it is reasonable that the policy cannot be updated too frequently. However, when the feedback is not delayed, the algorithm itself can keep updating the latent reward parameter estimations. Also, what is the motivation that "can only update for $O(\log T)$ times"? Is it possible to generalized $O(\log T)$ to other value?

- Given the above discussions, I think the contributions of this paper is a bit restrictive and cannot be easily generalized and extended. I acknowledge that I may miss something important. If the authors can provide more clarification on my concern, I would be happy to increase my score.

---

> ### Author Rebuttal · Authors · 2026-03-31
>
> Thanks for the thoughtful review. We answer your questions below:
>
> *W.1 "Both regret bounds appear $O(N)$  term (linear-in-$N$). Is it possible to reduce to sublinear-in-$N$ (e.g. $\sqrt{N}$ )?*
>
> Applying the known lower bounds for linear/logistic bandits [3,4,6] would imply that regret is $\Omega(Nd\sqrt{T})$. Just going by this bound, we are tempted to believe that the regret has to scale linearly with $N$. However, slate bandits have more structure, i.e, a slate comprises $N$ slots and items are available separately for each slot. Whether this extra structure can enable regret sublinear in $N$ is an interesting open question, which we will mention in our paper.
>
> *"W2. In the batched setting, the assumption on $M$ is overly restrictive. What about other $M$ (e.g. fixed constant, $O(\sqrt{T})$ )? How could we motivate this $O(\log \log T)$ value?"*
>
> There might be a small misunderstanding here. What we are showing is that we can achieve optimal $O(\sqrt{T})$ regret with just $O(\log \log T)$ batches. If $M$ is larger (say $\sqrt{T}$), we can continue to enjoy this same optimal regret by using only $O(\log \log T)$ out of this $\sqrt{T}$ batches, and the remaining $\sqrt{T} - O(\log \log T)$ batches can be empty. So, our result remains the same for any $M = \Omega(\log \log T)$. When $M$ is smaller than $O(\log \log T)$, prior work [1] (non-slate bandit) proves a $O(T^\frac{1}{2(1-2^{1-M})})$ regret bound. A very similar bound can be shown for our slate bandit setting as well. For example, when $M = 2$, this becomes $O(T)$. But note, $M=2$ means we have just two batches, making it almost non-adaptive and uninteresting. In different bandit settings [5], it has been shown that $\log \log T$ batches are necessary to achieve $O(\sqrt{T})$ regret. Therefore, the literature on batched bandits has focused on achieving optimal regret for $M = \log \log T$ batches only. Our work is an addition to this literature in the contextual slate bandit setting.
>
> *"W3. In the rarely switching setting, what is the motivation of "can only do limited times of estimations"? In real applications, it is reasonable that the policy cannot be updated too frequently. However, when the feedback is not delayed, the algorithm itself can keep updating the latent reward parameter estimations."*
>
> Estimating the reward parameters is an optimization problem (convex, in our case) that will utilize $t$ action-reward pairs at the end of round $t$. In the convex case, to get high accuracy, one would need to spend $poly(t)$ time to solve this problem. As $t$ grows, this becomes infeasible. Therefore, for practical applications, even when the feedback is not delayed, solving this optimization problem rarely is desirable.
>
> *W.4 "Also, what is the motivation that "can only update for $O(\log T)$ times"? Is it possible to generalized $O(\log T)$ to other value?"*
>
> The answer is similar to our answer for W2. Basically, if you allow more than $O(\log T)$ updates, we can still do only $O(\log T)$ updates and achieve optimal regret. Moreover, [2] shows a lower bound of $\tilde{\Omega}(d\log T)$ updates to obtain optimal $O(\sqrt{T})$ regret, which our algorithm matches.
>
> *W5. "Given the above discussions, I think the contributions of this paper is a bit restrictive and cannot be easily generalized and extended. I acknowledge that I may miss something important. If the authors can provide more clarification on my concern, I would be happy to increase my score."*
>
> We hope our answers to W2, W3, and W4 have clarified that the values we chose for $M$ are general and not restrictive. We will be happy to take any further questions from the reviewer and clarify it further.
>
> **References**
>
>
> [1] Sawarni et al. Generalized Linear Bandits with Limited Adaptivity. NeurIPS 2024.
>
> [2] Ruan et al. Linear Bandits with Limited Adaptivity and Learning Distributional Optimal Design, STOC, 2021
>
> [3] Dani et al. Stochastic Linear Optimization under Bandit Feedback. COLT 2008
>
> [4] Abeille et al. Instance-Wise Minimax-Optimal Algorithms for Logistic Bandits. AISTATS 2021
>
> [5] Gao et al. Batched Multi-armed Bandits Problem. NeurIPS 2019.
>
> [6] Cesa-Bianchi et al. Delay and Cooperation in Nonstochastic Bandits. COLT 2016

---

> > ### Author Rebuttal · Reviewer_k9Wc · 2026-04-03
> >
> > I thank the reviewer for addressing my concern. I will increase the score to 4.

---

### Official Review · Reviewer_AKD2 · 2026-03-13

**Soundness:** 3
**Presentation:** 3
**Significance:** 3
**Originality:** 3
**Overall Recommendation:** 5
**Confidence:** 3

**Summary:**

This paper considers the contextual slate bandit problem with GLM reward. The authors applied batched and rarely switching algorithm design to this setting and proposed B-SlateGLinCB and RS-SlateGLinCB. The authors proved the regret bounds of the proposed algorithms and demonstrated that RS-SlateGLinCB achieved the state-of-the-art regret bound in this setup. The numerical experiments demonstrated that the proposed algorithm incurred controllable cumulative regret with limited adaptivity.

**Compliance With Llm Reviewing Policy:**

Affirmed.

**Final Justification:**

All concerns are addressed. Therefore, I keep my positive score.

**Key Questions For Authors:**

**1.** As presented in the paper, RS-SlateGLinCB achieves the state-of-the-art regret bound, are there any direct or indirect comparisons for the regret bound for B-SlateGLinCB with related works?

**Limitations:**

The limitations in this work are the same as listed in "Key Questions For Authors", addressing or clarifying these questions could strengthen this paper significantly.

**Strengths And Weaknesses:**

**Soundness**
*Strengths:* The claims in the paper are well supported by rigorous proofs. The efficiency of the proposed algorithms are supported by extensive experiments.

**Presentation**
*Strengths:* The structure of the paper is clear, the concepts and algorithms are well explained.
*Weaknesses:* In Section 2.1, the slate $x$ is defined on $N$ slots while the GLM is defined on one slot ($\theta \in \mathbb{R}^{d}$), it is not clear how to calculate the expected reward for the slate.

**Significance and Originality**
This work applied batched and rarely switching design into the contextual slate GLM bandit setting, extending the applicability of these design principles from traditional bandit problem into slate bandit problem with state-of-the-art theoretical guarantee. Overall, this paper is novel and significantly contributes to this field.

---

> ### Author Rebuttal · Authors · 2026-03-31
>
> Thanks for the review. We address your questions and concerns below:
>
> *W1 "In Section 2.1, the slate $x$ is defined on $N$ slots while the GLM is defined on one slot ($\theta \in \mathbb{R}^{d}$), it is not clear how to calculate the expected reward for the slate."*
>
> While defining GLMs, we mentioned $\theta^\star \in R^d$, however, the GLM corresponding to our reward in our problem will have $\theta^\star \in R^{Nd}$. Thanks for pointing this out; we will make this clear in our paper.
>
> *Q1 "As presented in the paper, RS-SlateGLinCB achieves the state-of-the-art regret bound, are there any direct or indirect comparisons for the regret bound for B-SlateGLinCB with related works?"*
>
> The closest competitor of B-SlateGLinCB would be B-GLinCB (Algorithm 1, [2]), a batched algorithm for GLMs that obtains a regret bound of $\tilde{O}(d\sqrt{T})$ when actions are $d$-dimensional. Treating the slate bandit problem as a canonical bandit problem with $Nd$-dimensional feature vectors and bandit feedback, B-GLinCB would achieve a regret of $O(Nd\sqrt{T})$. However, the algorithm would be computationally inefficient, because of iteration over the set of $K^N$ slates while constructing the optimal designs.
>
> Our algorithm achieves a regret bound of $O(Nd^{3/2}\sqrt{T})$ (with G-Optimal designs) and $O(N^{3/2}d\sqrt{T})$ (with distributional optimal designs), at the cost of reducing the computational time at each round from exponential in $N$ to linear in $N$. An interesting future direction is obtaining lower bounds for the contextual slate bandit problem with bandit feedback (and limited adaptivity), with a constraint on computational efficiency.
>
> Another competitor of B-SlateGLinCB would be Slate-GLM-OFU (Algorithm 1, [1]), a sequential algorithm for logistic slate bandits, which obtains a regret bound of $O(Nd\sqrt{T})$. While Slate-GLM-OFU is computationally efficient, it is a completely sequential algorithm, whereas our algorithm adheres to limited adaptivity constraints. Hence, our algorithm lags by a factor of $O(\sqrt{d})$ (using G-optimal designs) or $O(\sqrt{N})$ (using distributional optimal designs).
>
> **References**
>
> [1] Goyal and Sinha. Efficient Algorithms for Logistic Contextual Slate Bandits with Bandit Feedback. UAI 2025
>
> [2] Sawarni et al. Generalized Linear Bandits with Limited Adaptivity. NeurIPS 2024.

---

> > ### Author Rebuttal · Reviewer_AKD2 · 2026-04-04
> >
> > All concerns are properly resolved in detail. Therefore, I keep my original score.

---

### Decision · Program_Chairs · 2026-04-30

**Decision:**

Accept (regular)

**Comment:**

This submission investigates the contextual slate bandit problem under a Generalized Linear Model (GLM) reward structure and limited adaptivity constraints. The authors propose two algorithms, B-SlateGLinCB for the batched setting and RS-SlateGLinCB for the rarely-switching setting, providing theoretical guarantees that match or improve upon existing bounds. A key technical contribution is a refined switching criterion that reduces the number of policy updates from polylogarithmic to logarithmic in $T$, effectively matching established lower bounds while improving practical exploration efficiency.

Reviewers were generally positive, noting that the paper addresses a well-motivated problem by combining the complexity of slate rewards with the practical need for low adaptivity. The technical refinement of the switching criterion was highlighted as a meaningful contribution that simplifies the algorithm compared to prior work while providing stronger theoretical results. However, some reviewers maintained concerns regarding the overall novelty and technical depth of the contribution. Consequently, the paper is recommended for a weak acceptance as a solid, though somewhat incremental, contribution to the literature on constrained online learning.

Recommendation: Weak Accept (low priority: accept if there is room in the program)